# Enhancer-associated H3K4 methylation safeguards in vitro germline competence

Tore Bleckwehl [1✉], Giuliano Crispatzu [1,2,3], Kaitlin Schaaf[1], Patricia Respuela[1,4], Michaela Bartusel [1,5], Laura Benson [6], Stephen J. Clark [6], Kristel M. Dorighi[7], Antonio Barral[8], Magdalena Laugsch[1,9], Wilfred F. J. van IJcken [10], Miguel Manzanares [8,11], Joanna Wysocka [7,12,13], Wolf Reik [6,14,15] & Álvaro Rada-Iglesias [1,3,4✉]

Germline specification in mammals occurs through an inductive process whereby competent cells in the post-implantation epiblast differentiate into primordial germ cells (PGC). The intrinsic factors that endow epiblast cells with the competence to respond to germline inductive signals remain unknown. Single-cell RNA sequencing across multiple stages of an in vitro PGC-like cells (PGCLC) differentiation system shows that PGCLC genes initially expressed in the naïve pluripotent stage become homogeneously dismantled in germline competent epiblast like-cells (EpiLC). In contrast, the decommissioning of enhancers associated with these germline genes is incomplete. Namely, a subset of these enhancers partly retain H3K4me1, accumulate less heterochromatic marks and remain accessible and responsive to transcriptional activators. Subsequently, as in vitro germline competence is lost, these enhancers get further decommissioned and lose their responsiveness to transcriptional activators. Importantly, using H3K4me1-deficient cells, we show that the loss of this histone modification reduces the germline competence of EpiLC and decreases PGCLC differentiation efficiency. Our work suggests that, although H3K4me1 might not be essential for enhancer function, it can facilitate the (re)activation of enhancers and the establishment of gene expression programs during specific developmental transitions.

---

[1] Center for Molecular Medicine Cologne (CMMC), University of Cologne, Cologne, Germany. [2] Department of Internal Medicine 2, University Hospital Cologne, Cologne, Germany. [3] Cluster of Excellence Cellular Stress Responses in Aging-Associated Diseases (CECAD), University of Cologne, Cologne, Germany. [4] Institute of Biomedicine and Biotechnology of Cantabria (IBBTEC), CSIC/University of Cantabria, Santander, Spain. [5] Department of Biology, Massachusetts Institute of Technology, Cambridge, MA, USA. [6] Epigenetics Programme, Babraham Institute, Cambridge, UK. [7] Department of Chemical and Systems Biology, Stanford University School of Medicine, Stanford, USA. [8] Centro Nacional de Investigaciones Cardiovasculares (CNIC), Madrid, Spain. [9] Institute of Human Genetics, Heidelberg University Hospital, Heidelberg, Germany. [10] Erasmus University Medical Center Rotterdam, Center for Biomics, Rotterdam, the Netherlands. [11] Centro de Biología Molecular Severo Ochoa (CBMSO), CSIC-UAM, Madrid, Spain. [12] Department of Developmental Biology, Stanford University School of Medicine, Stanford, USA. [13] Howard Hughes Medical Institute, Stanford University School of Medicine, Stanford, USA. [14] Centre for Trophoblast Research, University of Cambridge, Cambridge, UK. [15] Wellcome Trust Sanger Institute, Cambridge, UK. ✉email: tbleckwe@uni-koeln.de; alvaro.rada@unican.es

Competence can be defined as the ability of a cell to differentiate towards a specific cell fate in response to intrinsic and extrinsic signals[1]. While the extracellular signals involved in the induction of multiple cell fates have been described[2], the intrinsic factors that determine the cellular competence to respond to those signals remain elusive. One major example illustrating the dynamic and transient nature of competence occurs early during mammalian embryogenesis, as the primordial germ cells (PGC), the precursors of the gametes, become specified. In mice, following implantation and the exit of naïve pluripotency (E4.5–E5.5), PGC are induced from the epiblast around E6.0–E6.25 at the proximo-posterior end of the mouse embryo[3]. The induction of PGC occurs in response to signals emanating from the extraembryonic tissues surrounding the epiblast: BMP4 from the extraembryonic ectoderm and WNT3 from the visceral endoderm[4]. Furthermore, regardless of their position within the embryo, formative epiblast cells (~E5.5–6.25)[5–7] are germline competent when exposed to appropriate signals, but this ability gets lost as the epiblast progresses towards a primed pluripotency stage (>E6.5)[4]. However, the intrinsic factors that confer germline competence on the formative but not the primed epiblast cells remain obscure[8,9], partly due to the limited cell numbers that can be obtained in vivo from mouse peri-implantation embryos (E4.5–E6.5). These limitations were mitigated by a robust in vitro differentiation system whereby mouse embryonic stem cells (ESC) grown under 2i conditions (naïve pluripotency) can be sequentially differentiated into EpiLC and PGCLC that resemble the formative epiblast and E9.5 PGC, respectively[10]. This system facilitated the mechanistic and genomic characterization of transcription factors (TFs)[11–13] and epigenetic reprogramming events[14–16] previously shown to be involved in PGC specification in vivo[17,18].

Using the PGCLC in vitro differentiation system, two transcription factors, FOXD3 and OTX2, were found to promote the transition from naïve to formative pluripotency by coordinating the silencing of naïve genes and the activation of early post-implantation epiblast markers[9,19,20]. Subsequently, FOXD3 and OTX2 restrict the differentiation of EpiLC into PGCLC and, thus, the silencing of these TFs is required for in vitro germline specification[9,20]. The regulatory function of FOXD3 during these developmental transitions involves binding to and silencing of enhancers shared between naïve pluripotent cells and PGCLC[9]. Interestingly, during the transition from 2i ESC to EpiLC, FOXD3-bound enhancers lose TF and co-activator binding as well as H3K27ac but partly retain H3K4me1. This suggests that these enhancers do not become fully decommissioned, but transiently display a chromatin state similar, but not identical, to that of primed enhancers[21,22]. Enhancer priming typically involves binding of pioneer TFs and pre-marking by H3K4me1 that can precede and facilitate subsequent enhancer activation (i.e. marking by H3K27ac, recruitment of RNA Pol II, production of eRNAs)[23–26]. Interestingly, in differentiated macrophages, enhancers activated upon stimulation rapidly lose H3K27ac and TF binding, while retaining H3K4me1 for considerably longer. It was proposed that H3K4me1 persistence could facilitate a faster and stronger enhancer induction upon restimulation[27]. It is currently unknown whether, during development, H3K4me1 persistence once enhancers become decommissioned can similarly facilitate their eventual re-activation[21].

H3K4me1 is catalyzed by the SET domains of the histone methyltransferases MLL3 (KMT2C) and MLL4 (KMT2D), which are part of the COMPASS Complex[28,29]. The knockout (KO) of Mll3/4 or Mll4 alone impairs enhancer activation and results in differentiation defects in various lineages[30–37]. However, the importance of MLL3/4 for enhancer function might be independent of H3K4me1 deposition, since Mll3/4 catalytic mutant

ESC (Mll3/4 dCD) in which H3K4me1 was globally lost, only displayed minor defects in enhancer activation (i.e. H3K27ac, RNA Pol II and eRNA levels) in comparison to Mll3/4 KO ESC[38]. Similarly, work in Drosophila melanogaster showed that, while the KO of Trr, the homolog of Mll3/4 in flies, was embryonic lethal, an amino acid substitution in the SET domain of Trr that globally reduced H3K4me1 did not impair development or viability[39]. On the other hand, subsequent work with Mll3/4 dCD ESC showed that the recruitment of chromatin remodelers[40] and the establishment of long-range chromatin interactions[32] required H3K4me1. Hence, the functional relevance of H3K4me1 for enhancer function is still under debate[41].

Here we perform an extensive transcriptional and epigenetic characterization of the main stages of PGCLC differentiation to gain insights into the molecular basis of germline competence. Comparisons between germline competent EpiLC and non-competent EpiSC reveal that a notable fraction of PGCLC enhancers, which tend to be already active in the naïve stage (i.e. 2i ESC), partly retain H3K4me1 and remain accessible and responsive to TFs in EpiLC in comparison to EpiSC. Most importantly, the persistence of H3K4me1 within PGCLC enhancers seems to contribute to in vitro germline competence, as the absence of this histone mark reduces PGCLC differentiation efficiency.

## Results

### Characterization of the PGCLC in vitro differentiation system by single-cell RNA-seq.
To overcome the scarcity and transient nature of PGCs in vivo, we used an in vitro PGCLC differentiation system[10] whereby mouse ESC are differentiated into EpiLC from which PGCLC can be obtained within heterogeneous embryoid bodies (EB). In contrast, Epiblast stem cells (EpiSC), resembling the post-implantation gastrulating epiblast, cannot be efficiently differentiated into PGCLC and, thus, display limited germline competence (Fig. 1a)[7,10]. To better characterize this in vitro system, we performed single-cell RNA sequencing (scRNA-seq) across multiple stages of PGCLC differentiation (the scRNA-seq data can be easily explored with the cloupe file available through GEO: GSE155088). t-SNE analysis of the resulting single-cell transcriptomes (Supplementary Data 1) showed that cells tend to cluster within their corresponding differentiation stage (Fig. 1b). However, Day 2 and Day 4 EB showed cellular heterogeneity and formed distinct subclusters (Fig. 1b, c). One of these subclusters was identified as PGCLC based on the high expression of major PGC markers (e.g. Prdm14, Prdm1, Tfap2c, Dppa3), while the additional subpopulations within d4 EB were annotated based on the expression of cell identity markers identified by single-cell transcriptional profiling of E8.25 mouse embryos[42] (Fig. 1c, Supplementary Fig. 1a, b). Remarkably, these subclusters were similar to the extraembryonic tissues (i.e. extraembryonic ectoderm, extraembryonic mesoderm, and endothelium) that surround PGCs in the proximo-posterior end of the mouse embryo following germline specification in vivo. Furthermore, differential expression analysis between the PGCLC cluster and the remaining cells of the d2 and d4 EBs (see Methods) led to the identification of a set of 389 PGCLC genes (Supplementary Data 2), which included the PGC markers mentioned above as well as major naïve pluripotency regulators (e.g. Nanog, Esrrb) (Fig. 1d, e). In agreement with previous reports[10], we found that several PGCLC genes were more highly expressed in ESC, progressively silenced in EpiLC, and finally reactivated in PGCLC (Fig. 1d, e, Supplementary Fig. 1c). Despite the similar expression profile of ESC and PGCLC, major ESC (e.g. Klf4, Zfp57, Tbx3) and PGC (e.g Prdm1, Dppa3, Dnd1) markers were specifically expressed in the ESC and PGCLC clusters,

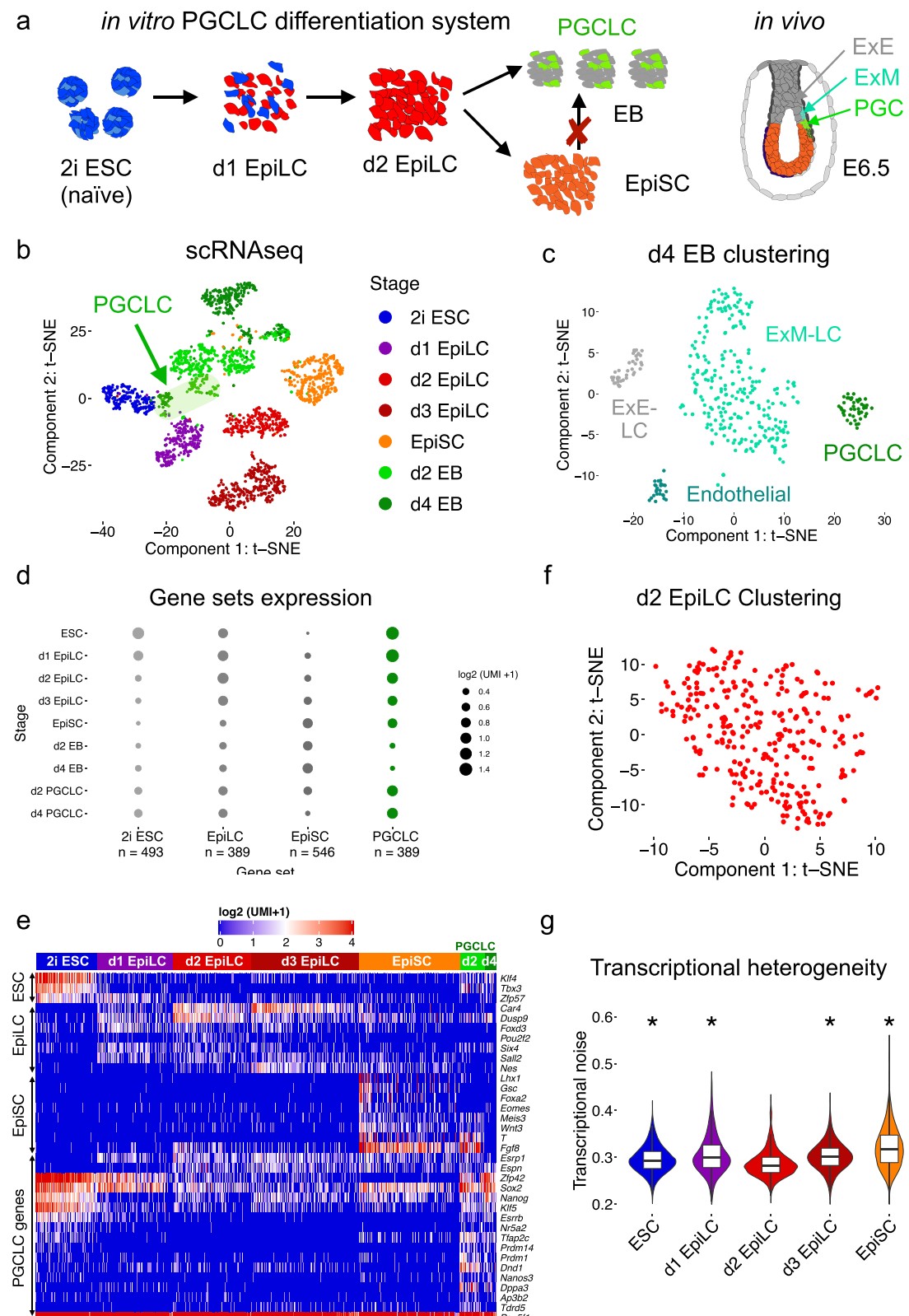

respectively (Fig. 1e, Supplementary Fig. 1d, Supplementary Data 2). Furthermore, differential expression analysis between PGCLC and in vitro pluripotent cell types (i.e. ESC, d2 EpiLC, and EpiSC) showed that ~25% of the PGCLC genes (100/389) displayed a PGCLC specific expression pattern (Supplementary Fig. 1e). Therefore, despite their similarities, 2i ESC and PGCLC are transcriptionally distinct. Lastly, we also defined gene sets

specific for the three investigated in vitro pluripotent cell types (i.e. ESC, d2 EpiLC, and EpiSC) by performing differential gene expression analysis between each cell type and the other two (Supplementary Data 2).

**Germline competent EpiLC are transcriptionally homogeneous.** Previous work indicates that the acquisition of germline

**Fig. 1 scRNA-seq profiling of the in vitro PGCLC differentiation system. a** Schematic representation of the in vitro PGCLC differentiation system. Embryonic stem cells (ESC) are differentiated into epiblast-like cells (EpiLC). Day 2 (d2) EpiLC are germline competent and can be differentiated into embryoid bodies (EB) containing primordial germ cell-like cells (PGCLC). In contrast, the progress of EpiLC into epiblast stem cells (EpiSC) restricts germline competence. The diagram to the far right illustrates the in vivo location of PGCs at the proximal-posterior end of a mouse embryo (ExE: Extraembryonic Ectoderm; ExM: Extraembryonic Mesoderm, PGC: Primordial germ cells). **b** t-SNE plot of the scRNA-seq data ($n = 2782$ cells) generated across the main stages of the in vitro PGCLC differentiation protocol. **c** t-SNE plot based on the scRNA-seq data generated in d4 EB ($n = 368$ cells). K-means clustering of d4 EB identified four clusters resembling the transcriptomes of PGCs and their main surrounding tissues in vivo: ExE, ExM, and Endothelial cells. **d** Gene expression summary for the different gene sets defined using the scRNA-seq data (see Methods). The number of genes in each set is indicated below and the size of a dot represents the average expression of all genes within a gene set for the indicated cellular stages (UMI: unique molecular identifier). **e** Heatmap showing the expression of selected genes from the gene sets shown to the left (i.e. ESC, EpiLC, EpiSC, and PGCLC gene sets) within individual cells belonging to the indicated cellular stages. The expression values are displayed as UMI counts. **f** t-SNE plot for the d2 EpiLC scRNA-seq data alone ($n = 289$ cells). **g** Violin plots showing transcriptional noise (defined as cell-to-cell transcript variation for the 500 most variable genes) for ESC, EpiLC, and EpiSC. Lower transcriptional noise indicates higher transcriptional similarity between the cells belonging to a particular stage. All cellular stages shown were compared to d2 EpiLC using two-sided Wilcoxon tests (*p-value $< 2.2 \times 10^{-16}$). The horizontal line in the boxplots indicates the median, the box indicates the first and third quartiles and the whiskers indicate $\pm 1.5 \times$ interquartile range.

competence in day 2 (d2) EpiLC entails the complete dismantling of the naïve gene expression program[5,6], which is then re-activated during PGCLC induction. In agreement with this, all d2 EpiLC clustered together and neither a distinct subpopulation indicative of a retained naïve pluripotency expression program nor signs of precocious germline induction could be identified (Fig. 1b, f; Supplementary Fig. 1f). Congruently, the cell-to-cell variability in gene expression levels, defined as transcriptional noise, was significantly lower in d2 EpiLC than in the preceding or subsequent cellular stages (Fig. 1g). This is in agreement with the transcriptional homogeneity of the E5.5 epiblast in vivo[43], which bears the lowest transcriptional noise during mouse peri-implantation development[44]. Lastly, we analyzed bulk RNA-seq data generated in $Otx2^{-/-}$[20] and $Prdm14^{-/-}$[14] EpiLC, which despite displaying increased and decreased germline competence, respectively, showed normal expression of PGCLC genes (Supplementary Fig. 1g). Therefore, in agreement with the previous work[6], our scRNA-seq analysis indicates that in vitro germline competence entails a transcriptionally homogeneous stage in which the gene expression program shared between naïve pluripotency and PGCLC is silenced.

**Identification of PGCLC active enhancers.** Many PGCLC genes, especially those active in ESC, are silenced in EpiLC and EpiSC (Fig. 1d, e, Supplementary Fig. 1c), yet only EpiLC display high germline competence. Taking previous observations into account[9,45,46], we hypothesized that enhancers involved in the induction of PGCLC genes might display epigenetic differences between EpiLC and EpiSC that could explain their distinct germline competence. To test this hypothesis, we first identified distal H3K27ac peaks in d6-sorted PGCLC using publically available data[15]. In agreement with our previous observations[9], a large fraction of the d6 PGCLC H3K27ac peaks were initially active in ESC, lost H3K27ac in EpiLC, and became progressively reactivated in d2 and d6 PGCLC (Supplementary Fig. 2a). Since most of the d6 PGCLC H3K27ac peaks were also active in ESC, we then used Capture Hi-C data generated in ESC[47] to system-atically associate these distal peaks to their putative target genes. Finally, we defined PGCLC enhancers as those distal d6 PGCLC H3K27ac peaks that could be physically linked to our PGCLC gene set (Supplementary Data 2). This resulted in 415 PGCLC enhancers linked to 216 of the 389 PGCLC genes (Fig. 2a). Furthermore, to compare epigenetic changes between different enhancer groups, EpiLC and EpiSC enhancers were defined using similar criteria (Methods; Supplementary Data 3).

To validate our PGCLC enhancer calling strategy, we first selected three representative enhancers linked to *Esrrb*, *Klf5*, and *Lrrc31/Lrrc34*, respectively (Fig. 2b, Supplementary Fig. 2b).

These three enhancers were initially active in ESC, got silenced in EpiLC, and finally became re-activated in PGCLC. Each enhancer was individually deleted in ESC using CRISPR/Cas9 technology (Supplementary Fig. 2c). The deletion of the enhancers associated with *Lrrc31/Lrrc34* and *Klf5* significantly reduced the expression of the corresponding target genes in ESC and d4 EB (Supplementary Fig. 2d, e). The *Esrrb* enhancer deletion had a moderate effect in ESC and severely diminished *Esrrb* expression in d4 EB (Fig. 2c). The moderate effect in ESC could be explained by the presence of additional ESC-specific enhancers (Fig. 2b) that might control *Esrrb* expression in ESC and compensate for the enhancer deletion[48]. Similarly, since some PGCLC genes are associated with multiple and potentially redundant regulatory elements in PGCLC (i.e. 415 enhancers linked to 216 PGCLC genes; 1.9 enhancers/gene), we individu-ally deleted three different enhancers (i.e. E1, E2, and E3) (Fig. 2d; Supplementary Fig. 2c) previously described as components of a *Prdm14* super-enhancer[49]. In agreement with previous work, the E2 deletion strongly reduced *Prdm14* expression in ESC, while the deletion of E3 or E1 had considerably smaller effects (Fig. 2e)[49]. Upon PGCLC differ-entiation, the regulatory importance of each enhancer changed and E1 clearly contributed to *Prdm14* expression, especially during early PGCLC induction (Fig. 2e). These results suggest that, rather than being components of an ESC super-enhancer, the E1–E3 elements differentially contribute to *Prdm14* expression in either ESC (i.e. E2) or early (d2 EBs) PGCLC (i.e. E1). Furthermore, in agreement with the role of *Prdm14* as a PGC master regulator[50], we found that the individual E1–E3 deletions significantly impaired PGCLC differentiation (Fig. 2f, Supplementary Fig. 2f). However, for the E2 and E3 enhancer deletions, we can not exclude that the impaired PGCC differentiation might be caused, at least partially, by the initial decrease in Prdm14 expression in ESC, which might compro-mise naïve pluripotency by increasing the levels of de novo DNA methyltransferases and, thus, of CpG methylation[51–53]. Altogether, the previous deletions support the relevance of the identified PGCLC enhancers and suggest that some of them (e.g. *Esrrb* and *Prdm14* E1 enhancers) are particularly relevant during PGCLC induction, while others might be important in both ESC and PGCLC.

**Partial decommissioning of PGCLC enhancers in EpiLC.** To explore whether PGCLC enhancers display any chromatin dif-ferences between EpiLC and EpiSC, we generated an extensive set of ChIP-seq and ATAC-seq data sets in ESC, EpiLC, and EpiSC (Fig. 3a; Supplementary Fig. 3a). Overall, the most pronounced changes within PGCLC enhancers were observed for H3K4me1,

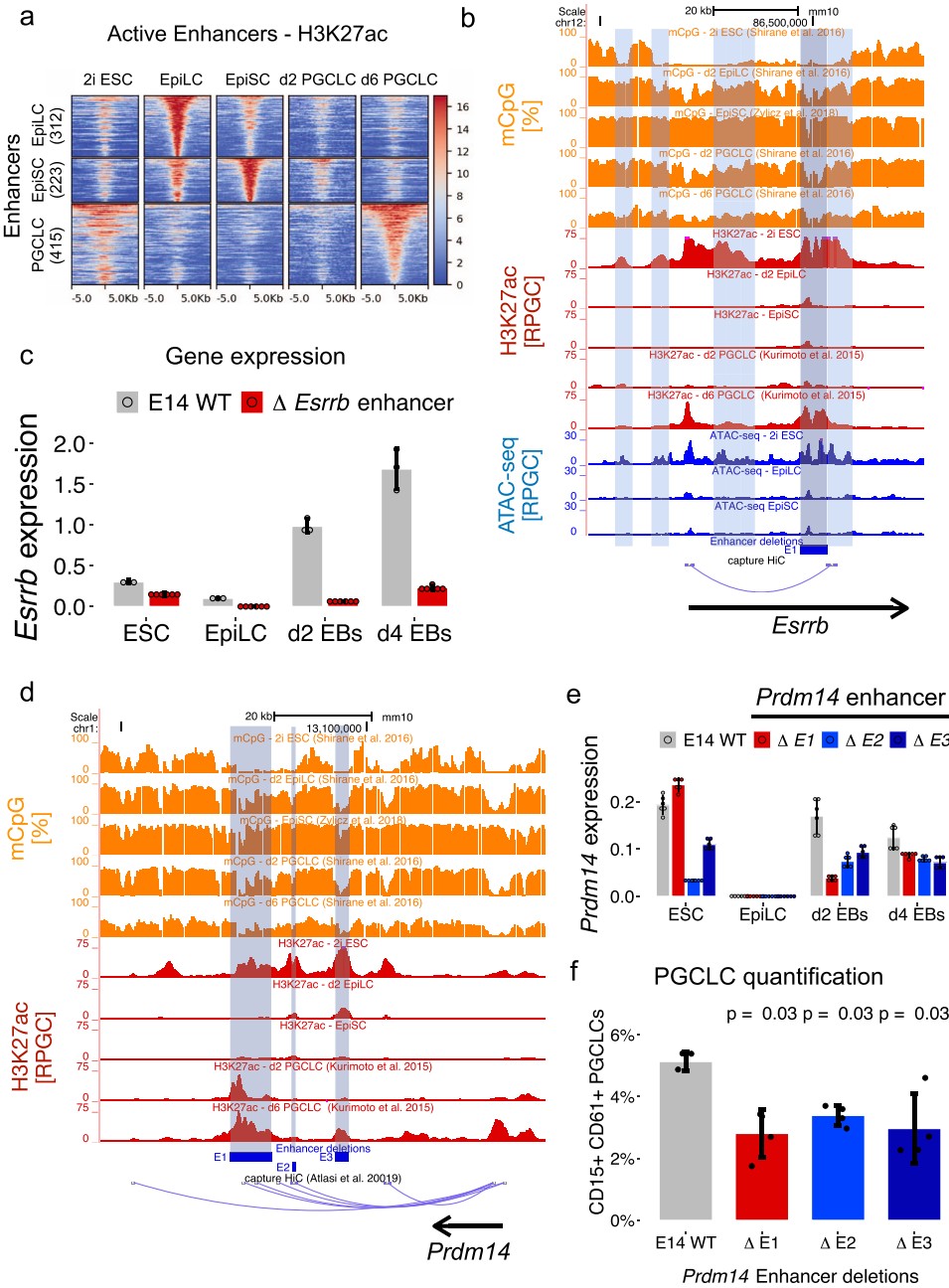

**Fig. 2 Identification and functional assessment of PGCLC enhancers. a** H3K27ac dynamics for EpiLC, EpiSC, and PGCLC enhancers during PGCLC differentiation (see *Methods* for the definition of enhancer sets). Enhancers within each set were ordered according to the H3K27ac levels in the corresponding cell type. RPGC: reads per genomic content. **b** ATAC-seq, H3K27ac, and CpG methylation dynamics during PGCLC differentiation for the *Esrrb* locus. A PGCLC enhancer physically linked to the *Esrrb* according to capture Hi-C data[47] is highlighted in gray. H3K27ac peaks found only in ESC and representing putative ESC-specific enhancers are highlighted in light blue. The blue rectangle denotes the generated enhancer deletion. **c** *Esrrb* expression levels were measured by RT-qPCR in d4 EB differentiated from WT ESC and the two ESC clonal lines with the *Esrrb* enhancer deletion shown in (**b**). The expression values were normalized to two housekeeping genes (*Eef1a1* and *Hprt*). The bar plots show the mean expression ± SD from 6 measurements (two clonal lines × three technical replicates shown as circles). **d** H3K27ac and CpG methylation dynamics during PGCLC differentiation for three PGCLC enhancers (E1–E3) found within the *Prdm14* locus. The blue rectangles denote the generated enhancer deletions. The E1-E2 enhancers are physically linked to *Prdm14* in ESC according to capture Hi-C data[47]. RPGC: reads per genomic content. **e** *Prdm14* expression levels were measured by RT-qPCR in d4 EB differentiated from WT ESC and ESC with the indicated *Prdm14* enhancer deletions (two clonal lines for each deletion). The expression values were normalized to two housekeeping genes (*Eef1a1* and *Hprt*). The bar plots show the mean expression ± SD from 6 measurements (two clonal lines x three technical replicates shown as circles). **f** WT ESC and ESC lines with the indicated *Prdm14* enhancer deletions were differentiated into PGCLC. PGCLC were measured as CD15+CD61+ cells within d4 EB. Each PGCLC quantification is shown as means ± SD from biological duplicates and two different clonal lines for each enhancer deletion ($n = 2 \times 2$). The p-values were calculated using two-sided Wilcoxon tests comparing cells with enhancer deletions to WT cells.

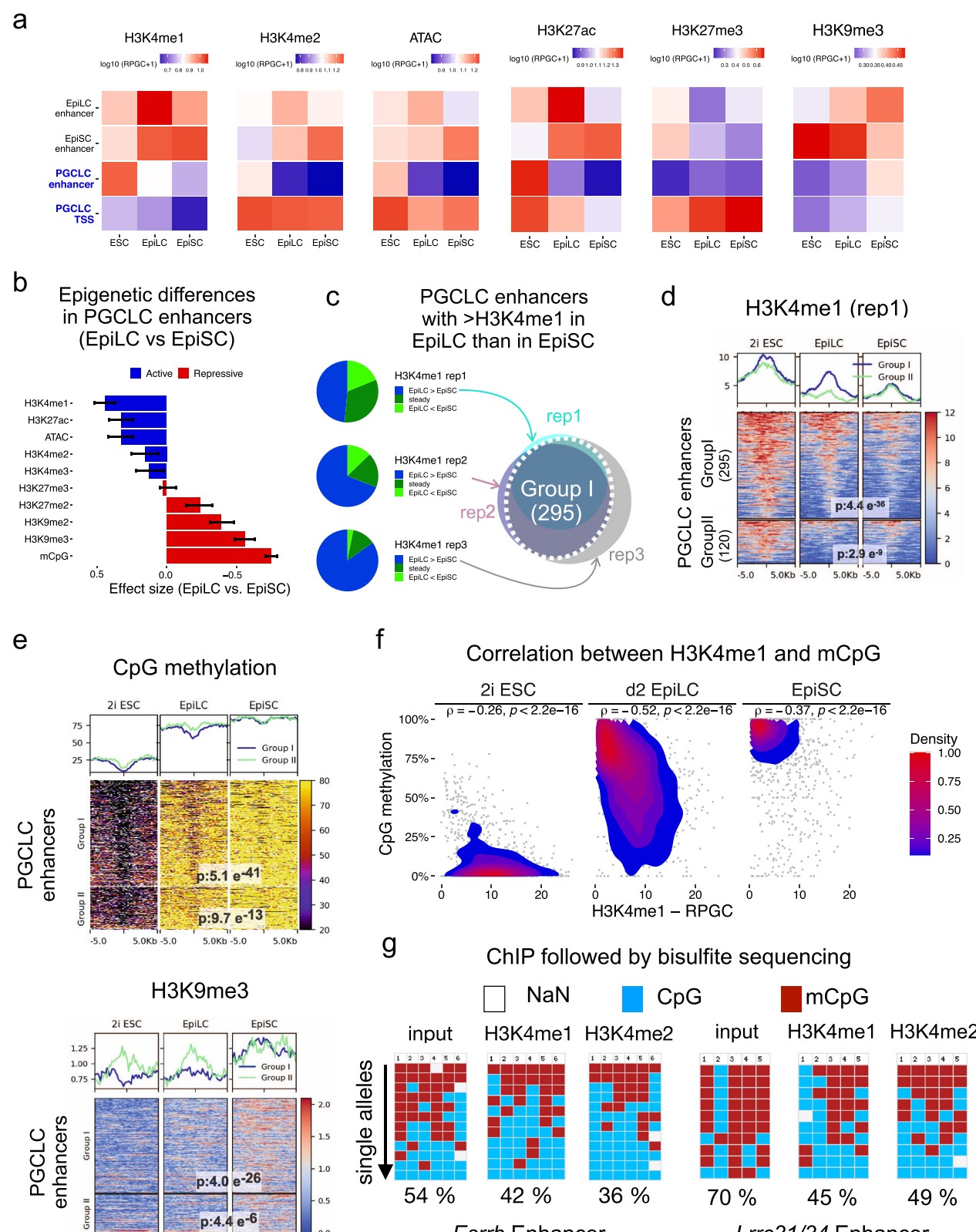

which was higher in EpiLC, and H3K9me3 and mCpG, which were increased in EpiSC (Fig. 3a, b). These differences were not observed for other enhancer groups (e.g. EpiSC enhancers), suggesting that they were not due to technical reasons or global epigenomic differences between EpiLC and EpiSC (Fig. 3a). Moreover, when analyzing the transcription start sites (TSS) of

the PGCLC genes we found that, although H3K4me1 was higher in EpiLC than in EpiSC, its overall levels were low compared to PGCLC enhancers (Fig. 3a). Similarly, constitutive hetero-chromatin marks (e.g. H3K9me3, mCpG) around TSS increased in EpiSC, but their levels were lower than within PGCLC enhancers. Other chromatin features typically found at promoter

**Fig. 3 Chromatin features of PGCLC enhancers in ESC, EpiLC, and EpiSC. a** Chromatin marks were measured in ESC, EpiLC, and EpiSC for the indicated enhancers and the TSS of the PGCLC genes. Quantifications were performed by measuring average signals within 1 kb of the enhancers or TSS (RPGC: Reads per genomic content). **b** The magnitude of the differences in chromatin marks between EpiLC and EpiSC within PGCLC enhancers ($n = 415$) was determined as the effect sizes from paired two-sided Wilcoxon tests. The bar plots show the effect size and the error bars represent the confidence intervals (0.95). **c** Three H3K4me1 ChIP-seq replicates were generated in EpiLC and EpiSC, respectively. For each replicate, the average H3K4me1 signal within 1 kb of each PGCLC enhancer was calculated. PGCLC enhancers showing EpiLC/EpiSC ratios >1.2-fold were assigned to the 'EpiLC>EpiSC' category. PGCLC enhancers assigned to the 'EpiLC>EpiSC' category in at least two replicates were defined as *Group I*. All other enhancers were assigned to *Group II*. **d** H3K4me1 levels (first replicate) for Group I and II PGCLC enhancers in EpiLC and EpiSC. *P*-values were calculated using paired two-sided Wilcoxon tests. **e** H3K9me3 and CpG methylation levels for Group I and II PGCLC enhancers in EpiLC and EpiSC. *P*-values were calculated using paired two-sided Wilcoxon tests. The scales for H3K9me3 and CpG methylation data are shown as RPGC and % of methylated CpGs, respectively. **f** Correlation plots between CpG methylation and H3K4me1 at Group I PGCLC enhancers. For each enhancer, the average CpG methylation and H3K4me1 levels (first ChIP-seq replicate) were calculated for two 500 bp bins up- and downstream of each enhancer, and spearman correlations (ρ) were determined. **g** Bisulfite sequencing analysis of the PGCLC enhancers associated with *Esrrb* and *Lrrc31/34* using as templates input DNA, H3K4me1 ChIP DNA, and H3K4me2 ChIP DNA generated in EpiLC. The columns correspond to individual CpGs within each enhancer that are unmethylated (blue), methylated (red), or not sequenced (gray). At least 10 alleles were analyzed for each template DNA (rows). In (**e**) and (**f**) the CpG methylation data were obtained from Zylicz et al. 2015[45].

regions (e.g. H3K4me2/3, high chromatin accessibility) were similar around PGCLC TSS in EpiLC and EpiSC (Fig. 3a). Therefore, subsequent analyses were focused on PGCLC enhancers rather than promoters.

To identify PGCLC enhancers showing consistent differences between EpiLC and EpiSC, we focused on H3K4me1, which is widely considered as an enhancer mark[54]. We generated two additional H3K4me1 ChIP-seq replicates in EpiLC and EpiSC (including data from a different ESC strain i.e. R1) and classified PGCLC enhancers in two groups: Group I: PGCLC enhancers showing an EpiLC/EpiSC H3K4me1 ratio higher than 1.2-fold in at least two replicates Group II: all other PGCLC enhancers showing EpiLC/EpiSC H3K4me1 ratios lower than 1.2-fold. Based on this classification, 71% of the PGCLC enhancers were assigned to Group I and, thus, showed higher H3K4me1 levels in EpiLC than in EpiSC in at least two replicates (71% Group I; 29% Group II) (Fig. 3c, d; Supplementary Fig. 3b, c). The Group I PGCLC enhancers also retained more H3K27ac in EpiLC, while the differences in H3K4me2 and chromatin accessibility, as measured by ATAC-seq, were rather moderate (Supplementary Fig. 3d). In contrast, the Group I enhancers displayed lower mCpG and H3K9me3 levels in EpiLC than in EpiSC, while these differences were considerably less obvious for the Group II enhancers (Fig. 3d, e, Supplementary Fig. 3e). Notably, previous work showed that germline genes and enhancers are repressed by the H3K9 methyltransferase EHMT2 during mouse post-implantation development[45], suggesting that Group I enhancers get progressively silenced as ESC differentiate into EpiLC and EpiSC. Furthermore, H3K4me1 and mCpG were inversely correlated across Group I PGCLC enhancers, particularly in EpiLC (Fig. 3f). This negative correlation was also investigated by performing H3K4me1 ChIP in EpiLC followed by bisulfite sequencing of two representative Group I PGCLC enhancers (i.e. *Esrrb* and *Lrrc31/Lrrc34* enhancers) (Fig. 3g). In comparison to the input genomic DNA, the H3K4me1-enriched DNA showed lower mCpG levels at the two analyzed enhancers (Fig. 3g), further supporting the antagonism of CpG methylation and H3K4 methylation[55,56].

In summary, a subset of the PGCLC enhancers shows partial retention of H3K4me1 and lower levels of heterochromatic marks (i.e. mCpG, H3K9me3) in EpiLC compared to EpiSC (Fig. 3d, e). These enhancers tend to be initially active in ESC but lose H3K27ac and TF/coactivator binding (indirectly measured by ATAC-seq) already in EpiLC (Supplementary Fig. 3d). Therefore, our data suggest that a significant fraction of PGCLC enhancers are not fully decommissioned in EpiLC, which we hypothesize could facilitate their re-activation in PGCLC and, thus, contribute to germline competence.

**Impaired decommissioning of PGCLC enhancers in *Otx2* deficient cells**. The deletion of *Otx2* increases and prolongs germline competence in EpiLC[20]. Hence, to further explore the relationship between the chromatin features of PGCLC enhancers and germline competence, we analyzed $Otx2^{-/-}$ cells[57]. Firstly, we confirmed the extended germline competence of $Otx2^{-/-}$ cells, which can be robustly differentiated into PGCLC after keeping them for up to seven days in EpiSC culture conditions (Fig. 4a, Supplementary Fig. 4a). ChIP-seq experiments revealed that the increased germline competence of $Otx2^{-/-}$ cells was correlated with the retention of H3K4me1, H3K4me2, and H3K27ac in PGCLC enhancers, particularly within those displaying higher levels of H3K4me1 in WT EpiLC (i.e. Group I enhancers) (Fig. 4b, c, Supplementary Fig. 4b). Moreover, H3K4me1/2 levels were higher in $Otx2^{-/-}$ ESC than in WT ESC in some but not all of the investigated Group I enhancers (Supplementary Fig. 4c). Nevertheless, the correlation between germline competence and H3K4me1 levels within PGCLC enhancers was not perfect, since $Otx2^{-/-}$ d4 EpiSC displayed higher germline competence than WT EpiLC, yet slightly lower H3K4me1 levels within Group I enhancers (Fig. 4c). Therefore, in addition to H3K4me1, other chromatin features within PGCLC enhancers might also contribute to the extended germline competence of $Otx2^{-/-}$ cells. In agreement with this possibility, Group I enhancers showed slightly higher H3K4me2 in $Otx2^{-/-}$ d4 EpiSC than in WT EpiLC (Fig. 4c). Furthermore, genome-wide as well as detailed analysis of representative enhancers (*i.e.* *Esrrb* and *Prdm1* enhancers) showed that the increased competence of $Otx2^{-/-}$ EpiLC and d4 EpiSC was also reflected in reduced CpG methylation levels within Group I PGCLC enhancers (Fig. 4d, Supplementary Fig. 4d). Overall, as the PGCLC genes get properly silenced in $Otx2^{-/-}$ EpiLC (Supplementary Fig. 1g), these results suggest that the extended germline competence of $Otx2^{-/-}$ cells could be linked to the impaired and delayed decommissioning of a subset of PGCLC enhancers.

**Partially decommissioned PGCLC enhancers are accessible and responsive to transcriptional activators in EpiLC.** To investigate whether the partial decommissioning of PGCLC enhancers in EpiLC compared to EpiSC could have any functional consequences, we evaluated the accessibility of these enhancers to major PGCLC transcriptional activators. To this end, we generated clonal ESC lines in which HA-tagged PRDM14 or NANOG could be overexpressed upon the addition of Doxycycline (Dox) (Supplementary Fig. 4e, f). In agreement with previous reports, the overexpression of either PRDM14 or NANOG upon differentiation of EpiLC into PGCLC yielded a high percentage of

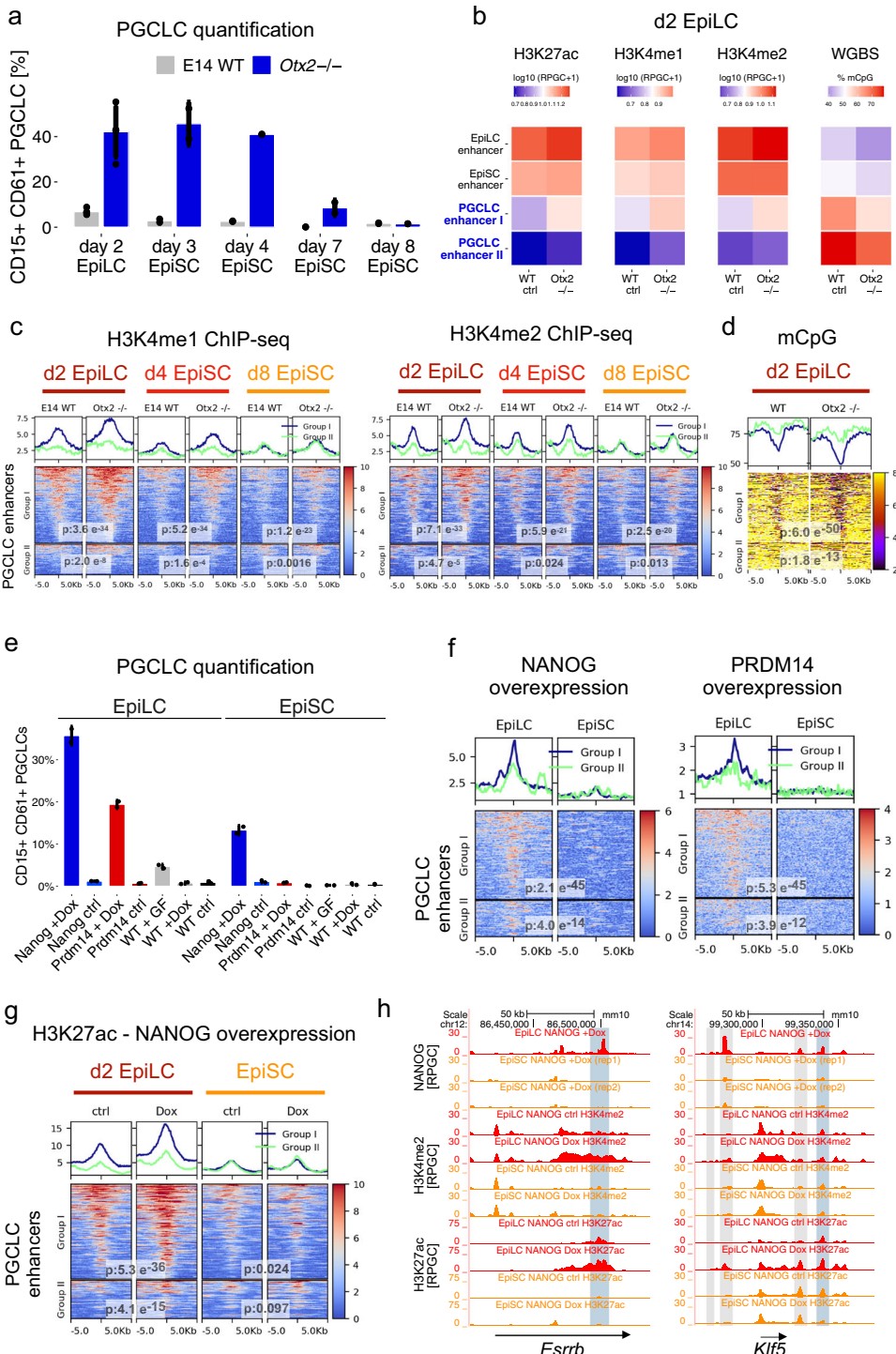

PGCLC in the absence of growth factors (Fig. 4e, Supplementary Fig. 4g)[12,58]. In contrast, the overexpression of PRDM14 or NANOG upon differentiation from EpiSC resulted in considerably less PGCLC (Fig. 4e, Supplementary Fig. 4g). To assess whether PGCLC enhancers were differentially accessible to these TFs in EpiLC and EpiSC, we performed ChIP-seq experiments after a short induction of HA-tagged PRDM14 or NANOG in both cell types. Importantly, PGCLC enhancers, especially those belonging to Group I, were considerably more bound by PRDM14-HA and NANOG-HA in EpiLC than in EpiSC (Fig. 4f). Furthermore, NANOG-HA binding to the PGCLC enhancers resulted in increased H3K27ac and H3K4me2 levels in EpiLC but

not in EpiSC, particularly within Group I enhancers (Fig. 4g, h, Supplementary Fig. 4h). Altogether, these results indicate that PGCLC enhancers, especially those belonging to Group I, are both accessible and responsive to transcriptional activators in EpiLC but not in EpiSC.

**Increased decommissioning of PGCLC enhancers in the absence of H3K4me1.** Our data suggest that a significant fraction of PGCLC enhancers display permissive chromatin features in EpiLC that might contribute to their increased germline competence compared to EpiSC. These permissive features could be attributed, at least partly, to the persistence of H3K4me1 in

**Fig. 4 In vitro germline competence is correlated with the partial decommissioning of PGCLC enhancers. a** E14 WT and *Otx2*⁻/⁻ ESC were differentiated into d2 EpiLC, d4 EpiSC or d8 EpiSC, which were then differentiated into PGCLC. PGCLC were quantified as the percentage of CD15⁺CD61⁺ cells within d4 EB. Bar plots: mean percentage of PGCLC ± SD from at least two biological replicates. **b** Average levels of chromatin marks measured in WT and *Otx2*⁻/⁻ d2 EpiLC within 1 kb of the indicated enhancers. **c** H3K4me1 and H3K4me2 levels for Group I and II PGCLC enhancers in d2 EpiLC, d4 EpiSC, and d8 EpiSC differentiated from E14 WT or *Otx2*⁻/⁻ ESC. P-values were calculated using paired two-sided Wilcoxon tests. Scales: RPGC. **d** CpG methylation for Group I and II PGCLC enhancers in d2 EpiLC differentiated from WT or *Otx2*⁻/⁻ ESC. P-values were calculated using paired two-sided Wilcoxon tests. Scale: percentage of methylated CpGs. **e** WT ESC and ESC lines enabling overexpression of NANOG-HA (blue) or PRMD14-HA (red) were differentiated into d2 EpiLC and EpiSC. D2 EpiLC and EpiSC were differentiated into PGCLC with (+Dox) or without (ctrl) Doxycycline and in the absence of growth factors. As a positive control, WT d2 EpiLC and EpiSC were differentiated with growth factors (WT + GF). Bar plots: mean percentage of PGCLC (CD15⁺CD61⁺ cells) in d4 EB ± SD from two biological replicates. **f** PRDM14-HA and NANOG-HA ChIP-seq signals in EpiLC and EpiSC in which PRDM14-HA or NANOG-HA were overexpressed. *P*-values were calculated using paired two-sided Wilcoxon tests. Scales: RPGC. **g** ESC enabling inducible overexpression of NANOG-HA were differentiated into d2 EpiLC and EpiSC. Cells were left untreated or treated with Doxycycline and H3K27ac ChIP-seq experiments were performed. P-values were calculated using paired two-sided Wilcoxon tests. Scales: RPGC. **h** NANOG-HA (two replicates in EpiSC), H3K27ac, and H3K4me2 signals in untreated (-Dox) and Dox-treated (i.e. NANOG-HA overexpression) EpiLC and EpiSC are shown for the *Klf5* and *Esrrb* loci. Deleted PGCLC enhancers are highlighted in blue and other PGCLC enhancers in gray. In (**b–c**), the H3K4me1 data for WT d2 EpiLC and d8 EpiSC corresponds to the second replicates from Fig. 3c.

EpiLC, as this histone mark could (i) protect PGCLC enhancers from mCpG and H3K9me2/3[56,59–63] and/or (ii) facilitate the recruitment of chromatin remodeling complexes[40]. To directly assess the importance of H3K4me1 for germline competence and PGCLC specification, we used mESC catalytically deficient for MLL3 and MLL4 (dCD and dCT ESC lines; Fig. 5a)[38]. Previous work showed that in dCD ESC, active enhancers show a moderate reduction in H3K27ac, while gene expression, eRNA levels, and the binding of MLL3/4 and their associated complexes are not affected, indicating that H3K4me1 is largely dispensable for the maintenance of enhancer activity (Supplementary Fig. 5a)[38,39]. Additional characterization of dCD and dCT cell lines revealed that, upon differentiation, the major reductions in H3K4me1 and H3K4me2 that PGCLC enhancers already displayed in ESC became further exacerbated in EpiLC and EpiSC (Fig. 5b, c, Supplementary Fig. 5b–d). Moreover, H3K27ac levels within PGCLC enhancers were also reduced in dCD/dCT cells compared to their WT counterparts, although such differences were not as pronounced as for H3K4me1 (Fig. 5b, Supplementary Fig. 5b, d). Similarly, other enhancer groups (i.e. EpiLC and EpiSC enhancers) also displayed strong H3K4me1/2 losses and milder H3K27ac reductions in dCD cells (Fig. 5c, Supplementary Fig. 5d, g). In contrast, the levels of the previous active histone modifications were rather similar around the TSS of PGCLC genes, with H3K4me1 even showing a slight increase in dCD cells (Supplementary Fig. 5d). In agreement with the protective role of H3K4me1/2 against CpG methylation[56,60], mCpG levels within PGCLC enhancers were generally elevated in dCD ESC[64], while in dCD EpiLC the increased methylation was evident among some Group I enhancers (Fig. 5d, Supplementary Fig. 5e, f). Next, to evaluate the potential functional consequences of the previous epigenetic changes, we generated RNA-seq data in WT and dCD ESC, EpiLC, and EpiSC. In agreement with previous work in ESC indicating that H3K4me1 is dispensable for enhancer function[38,39], the genes associated with PGCLC, EpiLC, and EpiSC enhancers showed rather minor expression differences between WT and dCD cells in the three investigated cell types (Fig. 5e, Supplementary Fig. 5h, i, Supplementary Data 4).

Overall, our analyses indicate that the decommissioning of a subset of PGCLC enhancers gets exacerbated in dCD EpiLC compared to their WT counterparts, resulting in a chromatin state similar to the one observed in WT EpiSC (i.e. lower H3K4me1 and higher mCpG; Fig. 3). However, these epigenetic changes do not result in major gene expression changes in any of the investigated in vitro pluripotent cell types.

**H3K4me1 is necessary for in vitro germline competence**. To address whether the increased decommissioning of PGCLC

enhancers observed in dCD/dCT EpiLC could compromise their germline competence, these MLL3/4 catalytic mutant cell lines were differentiated into PGCLC. Chiefly, both dCD and dCT cells showed a significant reduction in their PGCLC differentiation capacity (Fig. 6a, Supplementary Fig. 6a). In agreement with MLL3 and MLL4 being functionally redundant[65], such PGCLC differentiation defects were not observed when using cells that were catalytic mutant for MLL4 but not MLL3 (*i.e.* 4CT cells) (Fig. 5a, Fig. 6a). Furthermore, the compromised PGCLC differentiation of the dCD cells was still observed at a later time point (day 6, Supplementary Fig. 6b), indicating that the observed defects are not simply explained by a delay in PGCLC specification.

To gain molecular insights into the compromised PGCLC differentiation capacity of dCD cells, we performed scRNA-seq analyses of WT (1416 cells) and dCD (1699 cells) d4 EBs (Supplementary Data 5). UMAP (Uniform Manifold Approximation and Projection for Dimension Reduction) analysis of the resulting single-cell transcriptomes confirmed the presence within d4 EBs of subclusters resembling the main cell populations (*i.e.* PGCLC, ExEctorderm-like, ExMesoderm-like) that characterize the proximo-posterior end of the mouse embryo following germline specification (Fig. 6b). In addition, we also identified additional subclusters (i.e. ExEndoderm/Gut-like, 2-cell-like; Fig. 6b) that were not observed in the previous scRNA-seq analyses (Fig. 1, Supplementary Fig. 1), probably due to the lower number of d4 EB cells sequenced in those initial experiments. Most importantly, the PGCLC cluster consisted mostly of WT cells (162 WT cells (11.4% of all cells in WT d4 EBs) vs 8 dCD cells (0.5% of all cells in dCD d4 EB); Fig. 6c, d), suggesting that the defective PGCLC differentiation of dCD cells might be even more pronounced than our experiments using FACS and cell surface markers indicate (Fig. 6a). Moreover, while the transcriptomes of some extraembryonic-like cell types were similar between WT and dCD cells (e.g. Ex-Endoderm/Gut-like, Endothelial-like), many dCD cells were located in clusters with poorly defined identity that did not show differential expression of specific markers (Fig. 6b, Supplementary Fig. 6c). These undefined clusters were transcriptionally more heterogeneous than the remaining clusters found within the d4 EBs (Supplementary Fig. 6d). Interestingly, we noticed that within some of the non-PGCLC clusters there were cells showing high expression of PGCLC genes (Supplementary Fig. 6e) and, thus, a cellular identity somehow similar to PGCLC. Therefore, to further explore the PGCLC differentiation defects of the dCD cells we used an alternative approach to identify all cells within the WT and dCD d4 EBs that express major PGC markers (i.e. *Prdm1* or *Dppa3*) but not naïve pluripotency ones (i.e. *Klf4*). Although the

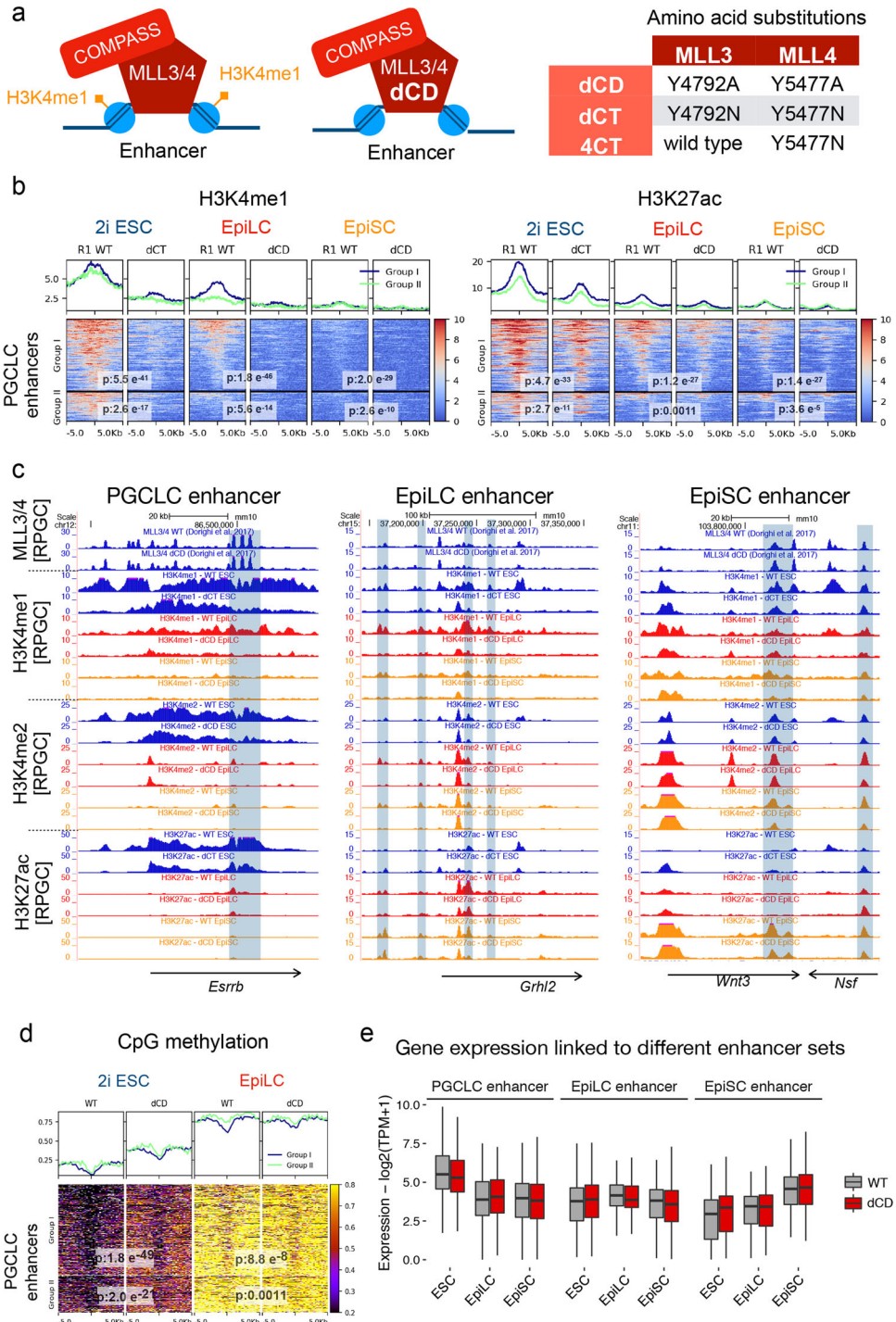

**Fig. 5 H3K4me1 is dispensable for enhancer-dependent gene expression in in vitro pluripotent cells. a** dCD cells express catalytic mutant MLL3 and MLL4 proteins without histone methyltransferase activity but capable of binding to their target sites and interacting with other proteins as part of the COMPASS complex. The different *Mll3/4* catalytic mutants used in this study are shown to the right. **b** H3K4me1 and H3K27ac levels for the Group I and II PGCLC enhancers in R1 WT and MLL3/4 catalytic mutant ESC lines as well as upon their differentiation into d2 EpiLC and EpiSC. *P*-values were calculated using paired two-sided Wilcoxon tests. The H3K4me1 ChIP-seq data shown for WT d2 EpiLC and EpiSC are the same ones used in Fig. 3c as a third replicate. Scales: RPGC. **c** H3K4me1, H3K4me2, and H3K27ac profiles in both WT and MLL3/4 catalytic mutant cells (ESC, EpiLC, and EpiSC) around representative PGCLC (*Esrrb*), EpiLC (*Grhl2*), and EpiSC (*Wnt3*) genes and their associated enhancers. **d** Percentage of CpG methylation for the Group I and II PGCLC enhancers in WT and dCD ESC as well as upon their differentiation into d2 EpiLC. P-values were calculated using paired two-sided Wilcoxon tests. The WGBS data in 2i ESC was obtained from Skvortsova et al. 2019[64]. Scale: percentage of methylated CpGs. **e** Expression (as measured by RNA-seq) of the genes associated with the PGCLC, EpiLC, and EpiSC enhancers in WT (gray) and dCD (red) ESC cells as well as upon their differentiation into EpiLC and EpiSC. The RNA-seq data in ESC was obtained from Dorighi et al. 2017[38] and the experiments in d2 EpiLC and EpiSC were performed in duplicates. The horizontal line in the boxplots indicates the median, the box indicates the first and third quartiles and the whiskers indicate ± 1.5 × interquartile range. TPM transcripts per million.

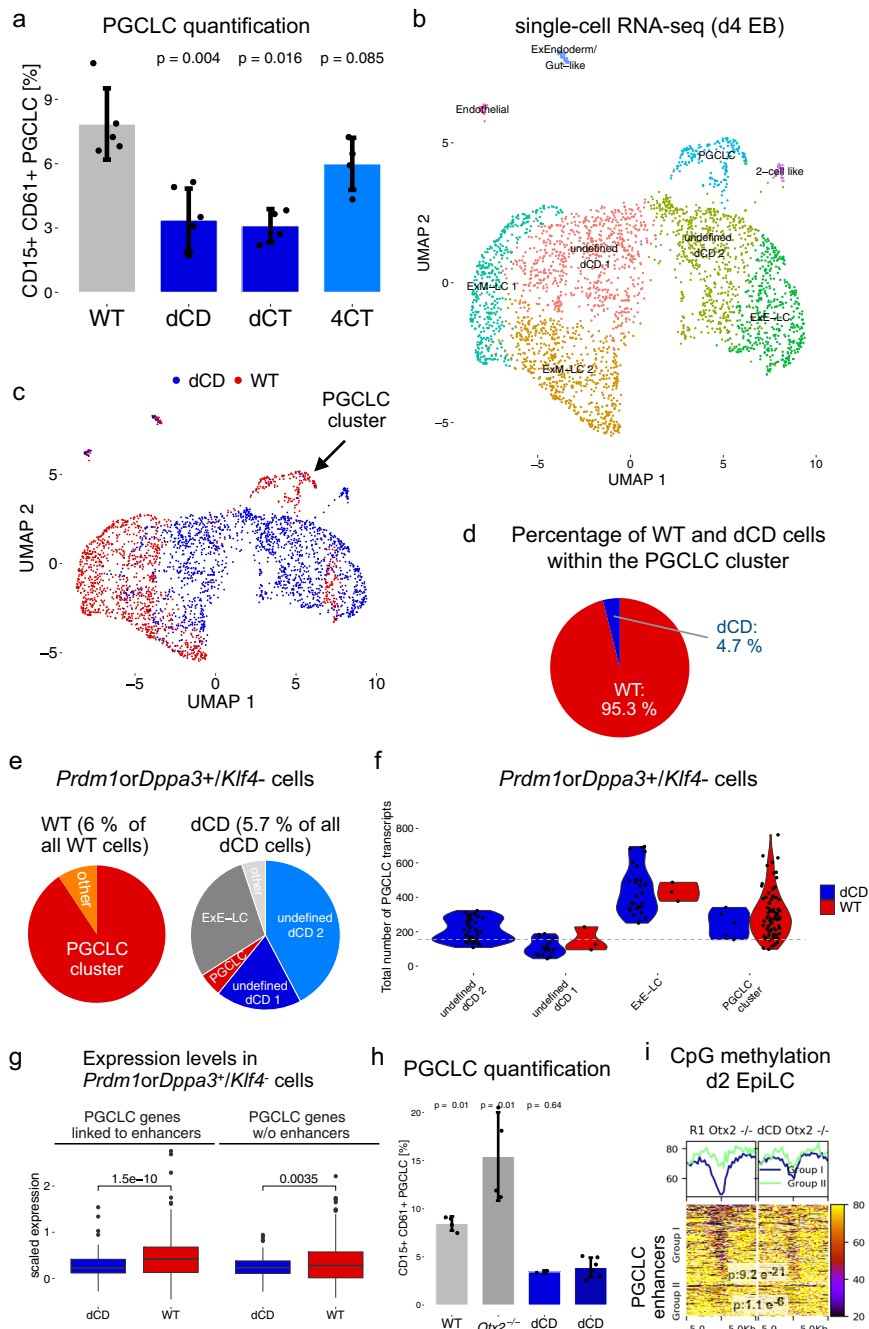

**Fig. 6 H3K4me1 is important for the proper induction of PGCLC and their associated gene expression program. a** WT ESC and ESC lines with MLL3/4 amino acid substitutions were differentiated into PGCLC. PGCLC were quantified as the percentage of CD15+CD61+ cells within d4 EB. Barplots: mean percentage of PGCLC ± SD from at least four biological replicates. *P*-values were calculated using two-sided Wilcoxon tests. **b** UMAP plot showing cell clusters identified within WT and dCD d4 EB according to scRNA-seq data. Cluster identity was determined by differential expression analysis between clusters and using specific tissue markers found in E8.25 mouse embryos[42]. **c** Same UMAP plot as in b, with WT and dCD cells represented by red and blue dots, respectively. **d** Percentage of WT and dCD cells found within the PGCLC cluster shown in **c**. **e** Cluster distribution of WT (left) and dCD (right) *Prdm1*or*Dppa3+*/*Klf4-* cells. **f** Transcript levels of all PGCLC genes in WT or dCD *Prdm1*or*Dppa3+*/*Klf4-* cells located within the indicated d4 EB clusters. Dots represent cells and the dashed line shows the mean transcript level of the PGCLC genes for all WT and dCD cells except those within the PGCLC cluster. **g** Expression of the PGCLC genes linked (*n* = 216) or not (*n* = 170) to at least one PGCLC enhancer within WT and dCD *Prdm1orDppa3* + /*Klf4-* cells. *P*-values were calculated using paired two-sided student *t*-tests. The horizontal lines in the boxplots indicate the median, the boxes indicate the first and third quartiles, and the whiskers the ± 1.5 × interquartile range. **h** R1 WT, *Otx2*−/−, dCD, and dCD *Otx2*−/− ESC were differentiated into PGCLC. PGCLC were quantified as the proportion of CD15+CD61+ cells within d4 EB. dCD *Otx2*−/− differentiations were performed in biological triplicates using two different clonal lines (*n* = 3 × 2). Other PGCLC measurements were performed in biological triplicates. Barplots: mean percentage of PGCLC ± SD. *P*-values were calculated using two-sided Wilcoxon tests. **i** Percentage of CpG methylation for Group I and II PGCLC enhancers in R1 *Otx2*−/− and dCD *Otx2*−/− d2 EpiLC. P-values were calculated using paired two-sided Wilcoxon tests. Scales: percentage of methylated CpGs.

overall abundance of these *Prdm1*or*Dppa3* + /*Klf4-* cells was similar in WT and dCD EBs, their distribution among the different cell clusters was quite different. Namely, the vast majority of WT *Prdm1*or*Dppa3* + /*Klf4-* cells were found within the PGCLC cluster (91%), while many dCD cells were part of the undefined subclusters (61%) (Fig. 6e), suggesting important transcriptional differences between WT and dCD *Prdm1*or*Dppa3* + /*Klf4-* cells (Fig. 6f). Congruently, the expression of the PGCLC genes associated with PGCLC enhancers (e.g. *Tfap2c*, *Prdm14*, *Utf1*, *Esrrb*) was significantly reduced in dCD *Prdm1*or*Dppa3* + /*Klf4-* cells in comparison to their WT counterparts, while smaller differences were observed for the PGCLC genes without associated enhancers (Fig. 6g, Supplementary Fig. 6f). Overall, our scRNA-seq analyses suggest that the induction of the PGCLC genes, particularly of those linked to PGCLC enhancers, is reduced but not fully abrogated in the dCD cells. Consequently, many dCD cells display a poorly defined identity in which the PGCLC expression program is only partially established.

Finally, to further assess the importance of H3K4me1 for PGCLC induction, we investigated whether the extended germline competence of *Otx2*[−/−] EpiLC could be also attributed, at least partly, to the retention of H3K4me1 and the impaired decommissioning of PGCLC enhancers (Fig. 4). To this end, we deleted *Otx2* in dCD ESC (i.e. dCD *Otx2*[−/−]) as well as in their parental WT ESC (i.e. R1 *Otx2*[−/−]) (Supplementary Fig. 6g) and differentiated them into PGCLC. As expected, the deletion of *Otx2* in the R1 ESC resulted in increased germline competence (Fig. 6h), although not as pronounced as in E14 ESC (Fig. 4a)[20], which could be attributed to the variable germline competence observed among different ESC lines[10]. Most importantly, both dCD and dCD *Otx2*[−/−] EpiLC showed a strong and similar reduction in their PGCLC differentiation capacity (Fig. 6h). Furthermore, genome-wide and locus-specific analyses of mCpG levels revealed that Group I PGCLC enhancers were considerably more methylated in dCD *Otx2*[−/−] EpiLC than in *Otx2*[−/−] EpiLC (Fig. 6i, Supplementary Fig. 6h).

Altogether, our data strongly suggest that H3K4me1 is important for in vitro germline competence and proper PGCLC induction. Although we cannot rule out that gene expression and epigenetic changes in ESC and/or extraembryonic-like cell types[66] might also contribute to the PGCLC differentiation defects observed in dCD/dCT cells, our data suggests that the persistence of H3K4me1 within the PGCLC enhancers might facilitate their reactivation during PGCLC induction.

**PGCLC enhancers get heterogeneously decommissioned in formative epiblast cells in vivo.** Overall, our work using the PGCLC in vitro differentiation system suggests that germline competence entails the persistence of permissive chromatin features within a significant fraction of PGCLC enhancers. To evaluate whether this is also true in vivo, we took advantage of several epigenomic datasets recently generated from mouse embryos[67]. Firstly, analysis of DNAse I data obtained in E9.5 and E10.5 PGC showed that our in vitro defined PGCLC enhancers are highly accessible in vivo (Fig. 7a). Next, to evaluate the dynamics of PGCLC enhancer decommissioning, we analyzed genome-wide CpG methylation data from mouse epiblasts[68]. In agreement with our in vitro observations, PGCLC enhancers showing incomplete decommissioning in EpiLC (i.e. Group I enhancers) displayed lower CpG methylation levels in germline competent E5.5 epiblast cells than in the E6.5 epiblast (Fig. 7b), in which germline competence is already reduced[4].

Previous single-cell CpG methylation analyses revealed that, during mouse peri-implantation stages, the formative epiblast is

particularly heterogeneous, especially within enhancers[69]. Therefore, we hypothesize that the lower CpG methylation levels observed for some PGCLC enhancers in the E5.5 epiblast in comparison to the E6.5 epiblast could be the result of the increased cell-to-cell variation. To evaluate this idea, we analyzed single-cell data in which DNA methylation and gene expression were measured for the same cells across different epiblast stages (i.e. E4.5, E5.5, and E6.5)[43]. Firstly, we measured mCpG heterogeneity by comparing the methylation status of individual CpGs within PGCLC enhancers[70] (Fig. 7c). This analysis revealed that the formative epiblast (E5.5) displayed the highest variation in mCpG, while in the primed epiblast (E6.5) the PGCLC enhancers were more homogeneously methylated (Fig. 7d). Interestingly, when comparing different enhancer sets across epiblast stages, the highest epigenetic heterogeneity (~30 %) was observed for the Group I PGCLC enhancers in the E5.5 epiblast (Fig. 7e). As the CpG coverage for the Group I PGCLC enhancers was similar across epiblast stages, the previous differences in mCpG heterogeneity are unlikely to be caused by technical reasons (Supplementary Fig. 7a). Lastly, we compared mCpG levels within Group I PGCLC enhancers and the expression of their associated genes across the profiled epiblast cells (Fig. 7f). Similar to what we observed for EpiLC and EpiSC, the PGCLC genes associated with the Group I enhancers display low and similar expression in E5.5 and E6.5 epiblast cells despite the differences in CpG methylation.

Overall, the previous data support that PGCLC enhancers are also partially decommissioned in the formative epiblast and indicate that such partial decommissioning could indeed reflect epigenetic heterogeneity. Nevertheless, the relevance of the partial decommissioning and epigenetic heterogeneity of PGCLC enhancers for germline competence in vivo remains to be demonstrated.

## Discussion
Direct evidences supporting the functional relevance of H3K4me1 are scarce, partly due to the difficulties in separating the enzymatic and non-enzymatic functions of histone methyltransferases[71]. This limitation was recently overcome by establishing *Mll3/4* catalytic mutant ESC lines in which H3K4me1 is lost from enhancers[38]. Under self-renewing conditions, the loss of H3K4me1 partly reduced H3K27ac but did not affect transcription from either enhancers or gene promoters, suggesting that H3K4me1 is dispensable for the maintenance of enhancer activity[38]. Similarly, H3K4me1 was not required for de novo enhancer activation (i.e. without the prior presence of H3K4me1) upon somatic differentiation of ESC (Fig. 5)[66]. However, it is still unclear whether H3K4me1 is important for the activation of primed enhancers already pre-marked with this histone modification[25,31]. In the case of in vitro germline competence, here we report that a subset of PGCLC enhancers gets partly decommissioned in EpiLC and retains permissive chromatin features, including H3K4me1, already present in a preceding active state (*i.e.* in 2i ESC) (Fig. 7g). This resembles the so-called latent enhancers previously described in differentiated macrophages, in which, following an initial round of activation and silencing, the persistence of H3K4me1 was proposed to facilitate subsequent enhancer induction upon restimulation[27]. The mechanisms involved in the persistence of H3K4me1 and other permissive chromatin features are still unknown, although we can envision at least two non-mutually exclusive possibilities: (i) a passive mechanism whereby MLL3/4 binding to PGCLC enhancers is already lost in EpiLC, but H3K4me1 can still be transiently retained due to the slow dynamics of H3K4 demethylation[72,73]; (ii) an active maintenance mechanism similar to the one reported

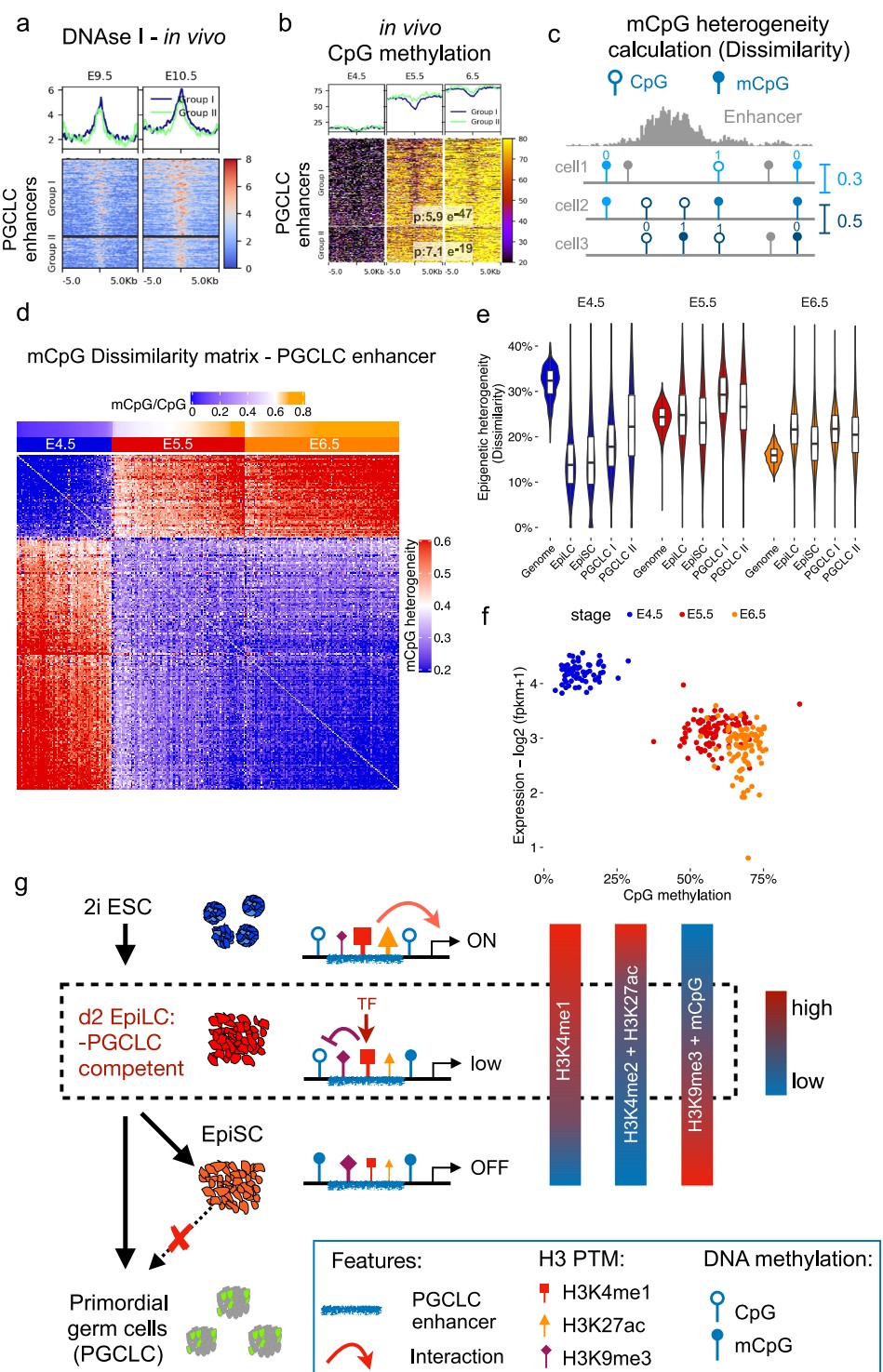

for enhancer priming[24,25], whereby the binding of certain TFs might enable the persistent recruitment of MLL3/4 and the retention of H3K4me1 within PGCLC enhancers. Since PGCLC enhancers display low and similar ATAC-seq signals in EpiLC and EpiSC (Fig. 3), this would argue in favor of passive mechanisms rather than active retention of TFs and co-activators (e.g. MLL3/4) in EpiLC.

*Drosophila melanogaster* embryos expressing catalytic deficient *Trr*, the homolog of mammalian *Mll3/4*, develop normally, thus arguing against a major role for H3K4me1 in enhancer function[39]. In contrast, the loss of MLL3/4 catalytic activities in mouse embryos resulted in early embryonic lethality (~E8.5)[66], suggesting that H3K4me1 is functionally relevant during mammalian embryogenesis. It is important to consider that, in comparison to mammalian cells, the *D. melanogaster* genome is largely devoid (<0.03%) of CpG methylation throughout its entire life cycle[74]. Previous studies have extensively documented the protective role of H3K4me1/2 against mCpG and heterochromatinization[56,59–63]. Congruently, we showed that H3K4me1 and mCpG levels within PGCLC enhancers are negatively correlated and that H3K4me1-deficient cells displayed increased mCpG levels within Group I PGCLC enhancers.

**Fig. 7 Chromatin features of PGCLC enhancers in vivo. a** DNAse-seq levels for Group I and II PGCLC enhancers in PGCs from E9.5 and E10.5 mouse embryos[67]. Scales are in RPGC. **b** CpG methylation levels for Group I and II PGCLC enhancers in E4.5, E5.5, and E6.5 mouse epiblasts[68]. P-values were calculated using paired two-sided Wilcoxon tests. Scales: percentage of methylated CpGs. **c** mCpG heterogeneity was estimated based on the mCpG dissimilarity concept. For each pairwise comparison, the methylation status of CpGs covered in the two cells being compared is considered (blue lollipops). If two cells show the same or different methylation status they receive a value of 0 (similarity) or 1 (dissimilarity), respectively. The mean of all pairwise comparisons reflects CpG methylation heterogeneity. **d** Heatmap showing mCpG heterogeneity within PGCLC enhancers between pairs of individual cells. mCpG heterogeneity values are presented with a blue-red scale (blue: similar; red: dissimilar). The developmental stages of the investigated cells (E4.5, E5.5, or E6.5) and the average CpG methylation (blue-orange scale) measured for all PGCLC enhancers within each cell ($n = 261$ cells) are shown above the heatmap. **e** CpG methylation heterogeneity was measured in E4.5, E5.5, and E6.5 epiblast cells for all covered CpGs in the mouse genome as well as within EpiLC, EpiSC, and PGCLC enhancers. The horizontal lines in the boxplots indicate the median, the boxes the first and third quartiles, and the whiskers the ±1.5×interquartile range. **f** Comparison of the single-cell CpG methylation of Group I PGCLC enhancers and the single-cell RNA expression of genes linked to them ($n = 256$) in E4.5, E5.5, and E6.5 epiblasts. fpkm: Fragments per kilobase per million mapped reads. **g** Model illustrating the potential relevance of the partial decommissioning of Group I PGCLC enhancers for in vitro germline competence. The persistence of H3K4me1 within PGCLC enhancers in EpiLC might protect them from heterochromatinization (H3K9me3 and CpG methylation) and render them accessible and responsive to transcriptional activators. In turn, this can confer germline competence by facilitating PGCLC enhancer (re)activation during PGCLC differentiation.

Therefore, although the importance of mCpG for enhancer activity is still debated[75–77], it is tempting to speculate that the lack of DNA methylation in flies renders them less sensitive to H3K4me1 loss. Nevertheless, *Trr* catalytic mutant fly embryos displayed aberrant phenotypes under stress conditions, suggesting that H3K4me1 might help to fine-tune enhancer activity under suboptimal conditions. Similarly, we found that the induction of PGCLC genes linked to enhancers was reduced in dCD cells, although the magnitude of the observed gene expression changes was moderate (Fig. 6g). Therefore, we propose that H3K4me1 might facilitate, rather than being essential for, enhancer (re) activation and the robust induction of developmental gene expression programs (Fig. 7g). This H3K4me1 facilitator role might involve not only protection from DNA methylation and heterochromatinization but also increased accessibility and responsiveness to transcriptional activators (Fig. 4)[40] as well as physical proximity between genes and enhancers[32]. Future work in additional species and cellular transitions will establish the relevance and prevalence of H3K4me1 function within enhancers.

Finally, our work highlights the diverse mechanisms whereby the epigenetic status of enhancers can contribute to developmental competence. For instance, a subset of early brain enhancers displays a poised state in ESC (i.e. H3K4me1 + H3K27me3) that might facilitate their activation upon neural differentiation[78], while in endodermal progenitors the binding of pioneer TFs and the consequent priming of relevant enhancers by H3K4me1 might facilitate their activation during subsequent differentiation[24,25]. Moreover, recent work based on chromatin accessibility or multi-omic profiling indicates that the commitment towards certain cell lineages involves the activation of already accessible enhancers, while in other cases this can occur through de novo enhancer activation[43,79]. Future work will elucidate the prevalence and regulatory mechanisms by which the chromatin state of enhancers can contribute to cellular competence and the robust deployment of developmental gene expression programs. In this regard, it would be important to evaluate whether, as reported here for PGCLC induction, the partial decommissioning of enhancers can be involved in their subsequent reactivation and, thus, in the induction of gene expression programs in other developmental contexts. Similar mechanisms might be also important in other physiological[27] and pathological[80,81] contexts in which a previously used but already dismantled gene expression program gets re-activated.

## Methods

**WT and transgenic ESC lines**. Two different male mouse ESC lines (i.e. E14Tg2a and R1) were used as indicated for each particular experiment. The *Mll3/4* dCD, *Mll3/4* dCT, and *Mll4* CT ESC lines were derived from WT R1 ESC as described

in[38]. The E14 *Otx2*[−/−] ESC line[57,82] and its parental E14Tg2a ESC were kindly provided by Christa Bücker. Other transgenic ESC lines generated in this study are more extensively described in the following sections and are derived from E14Tg2a ESC.

**Cell culture and differentiation protocols**. All ESC lines were cultured under serum+LIF conditions both for regular maintenance and CRISPR/Cas9 genome editing experiments. More specifically, ESC were cultured on gelatin-coated plates using Knock-out DMEM (*Life Technologies*) supplemented with 15% FBS (*Life Technologies*) and LIF. Before each PGCLC differentiation, ESC were adapted to 2i media (serum-free N2B27 medium supplemented with MEK inhibitor PD0325901 [0.4 μM, *Miltenyi Biotec*], GSK3β inhibitor CHIR99021 [3 μM, *Amsbio*], and LIF) for at least 4 days in gelatin-coated tissue culture plates[83]. All measurements described in the manuscript for ESC were performed under 2i conditions.

For EpiLC and PGCLC differentiation, the protocols from[83] were followed. For the EpiLC differentiation, 6-well plates were coated with 15 μg ml$^{-1}$ Fibronectin and $2 × 10^5$ 2i ESC were differentiated in N2B27 media supplemented with 20 ng ml$^{-1}$ Activin A, 12 ng ml$^{-1}$ bFGF, and 10 μl ml$^{-1}$ KSR for two days (unless indicated otherwise for particular experiments). For the PGCLC differentiation, EpiLC or EpiSC were plated at a density of $2 × 10^4$ cells ml$^{-1}$ and cultured as embryoid bodies (EB) on Ultra-Low attachment multiwell plates (*Corning® Costar®*) in GK15 medium supplemented with growth factors (0.5 μg ml$^{-1}$ BMP4, 0.1 μg ml$^{-1}$ SCF, 1000 U ml$^{-1}$ LIF and 50 ng ml$^{-1}$ EGF, no BMP8a). Unless otherwise indicated, PGCLC were typically analyzed after four days.

EpiSC differentiation was performed according to[84]. 2i ESC were plated at a density of $2.5 × 10^5$ cells ml$^{-1}$ on plates coated with 15 μg ml$^{-1}$ Fibronectin and differentiated into EpiSC by passaging them in N2B27 media supplemented with 20 ng ml$^{-1}$ Activin A and 12 ng ml$^{-1}$ bFGF for at least eight days or as indicated in the results section for particular experiments.

**Enhancer and gene deletions with CRISPR/Cas9**. Pairs of gRNAs flanking each of the selected enhancers/genes were designed with *benchling* (https://www.benchling.com). ESC were transfected with pairs of gRNA-Cas9 expressing vectors[85] specific for each enhancer/gene using Lipofectamine 3000 (*invitrogen*). After transfection and puromycin selection, single cells were seeded into a 96 well plate to derive clonal cell lines. Then, clonal lines were genotyped by PCR and the presence of the intended deletions in each clonal line was verified by Sanger-sequencing. gRNA sequences and genomic coordinates of the different deletions are listed in Supplementary Data 6.

**Generation of the Mll3/4 dCD, Mll3/4 dCT and Mll4 CT ESC lines**. The *Mll3/4* dCD ESC line was previously generated and described[38]. The *Mll3/4* dCT and *Mll4* CT ESC lines were also generated in[38]. Briefly, R1 ESC were co-transfected with a 200 bp single-stranded oligonucleotide donor template harboring the desired point mutations and a vector coexpressing Cas9 and gRNAs targeting *Mll3* and *Mll4*. After 48 h, single GFP + cells were sorted into 96 well plates coated with fibronectin (5 mg ml$^{-1}$). The resulting clonal ESC lines were genotyped by PCR amplification of a region spanning the cleavage site, followed by digestion with a restriction enzyme predicted to only cut the wild-type PCR products. Once candidate ESC lines were selected, the presence of the intended *Mll3/4* mutations was confirmed by Sanger sequencing.

**Ectopic and inducible expression of candidate transcription factors**. Mouse *Prdm14* and *Nanog* cDNAs were amplified and cloned into a Doxycycline (Dox)-inducible piggyBac vector[86] enabling the ectopic expression of selected genes fused with an HA-tag at their C-terminus. The resulting piggyBac vectors were transfected together with a Super PiggyBac Transposase expressing vector (*Systems*

Bioscience) and a Tet transactivator. After transfection, single cells were seeded into a 96 well plate to derive clonal ESC lines. The clonal ESC lines with the lowest expression of the transgenic genes in the absence of Dox were selected. To evaluate the effects of PRDM14 and NANOG ectopic expression on PGCLC specification, 1 µg ml$^{-1}$ Dox was added once PGCLC differentiation was started. To investigate the effects of the ectopic expression of PRDM14 or NANOG in EpiLC and EpiSC, 1 µg ml$^{-1}$ Dox was added 18 h before the cells were collected for downstream ChIP-seq experiments (i.e. EpiLC differentiation was initiated, after 30 h Dox was added and cells were collected after 48 h; EpiSC were maintained in differentiation media for at least eight passages and then treated with Dox for 18 h).

**Quantification of PGCLC by flow cytometry.** In general, PGCLC were quantified using antibody staining and flow cytometry[83]. Briefly, after four days of PGCLC differentiation, the resulting EB were dissociated and stained for 45 min with antibodies against CD61 (Biolegends, cat#104307; dilution 1:500) and CD15 (Thermo Fisher, cat#50-8813-41; dilution 1:200) conjugated with PE and Alexa Fluor 647, respectively. All PGCLC quantifications were performed using the FACS CantoII Cytometer (BD Bioscience) equipped with the BD FACSDiva Software. PGCLC sorting was performed on a FACS AriaIII cell sorter (BD Bioscience) and the data were analyzed with FlowJo (Version 10.6.1).

**ATAC-seq.** The ATAC-seq protocol was adapted from[87] using $3.8 \times 10^4$ and $5.0 \times 10^4$ cells for the two replicates generated for each of the investigated cell types, respectively. Briefly, the cells were lysed in lysis buffer (10 mM Tris-HCl, pH 7.4, 10 mM NaCl, 3 mM MgCl2, 0.1% IGEPAL CA-630) supplemented with freshly added protease inhibitor for 15 min. Following centrifugation, the pellet was resuspended in a Tn5 transposase reaction mix (Illumina) for 30 min at 37 °C. Following DNA purification with the MinElute PCR purification kit (Qiagen), libraries were prepared with the Nextera DNA library prep kit (Illumina).

**ChIP-seq and ChIPmentation.** Cells were cross-linked with 1% formaldehyde for 10 min, followed by quenching with 0.125 M glycine, harvesting, and washing in PBS containing protease inhibitors. The cells were sequentially lysed in three buffers (lysis buffer 1: 50 mM HEPES, 140 mM NaCl, 1 mM EDTA, 10% glycerol, 0.5% NP-40, 0.25% TX-100; lysis buffer 2: 10 mM Tris, 200 mM NaCl, 1 mM EDTA, 0.5% EGTA; lysis Buffer 3: 10 mM Tris, 100 mM NaCl, 1 mM EDTA, 0.5% EGTA, 0.1% Na-Deoxycholate, 0.5% N-lauroylsarcosine) with rotation for 10 min in between. Then, the chromatin was sonicated with the Epishear™ Probe Sonicator (Active Motif) with 20 s ON and 30 s OFF for 8 cycles. After centrifugation, the supernatant was divided into input and ChIP samples. The ChIP samples were incubated with specific antibodies (Supplementary Data 6) overnight, followed by immunoprecipitation with Dynabeads Protein G beads (invitrogen). Next, the beads were washed with RIPA buffer (50 mM Hepes, 500 mM LiCl, 1 mM EDTA, 1% NP-40, 0.7% Na-Deoxycholate) on a magnet, eluted (50 mM Tris, 10 mM EDTA, 1% SDS), and reverse cross-linked at 65 °C overnight in parallel with the input. Finally, DNAs were purified with the ChIP DNA Clean & Concentrator (Zymo Research) and analyzed by qPCR (Primers in Supplementary Data 6), or ChIP libraries were generated with the TruSeq kit (Illumina).

When ChIP-seq profiles were generated from low cell numbers (e.g. d4 EB obtained with the PGCLC differentiation protocol), the ChIPmentation protocol described by Schmidl et al. 2015[88] was used. Following immunoprecipitation with Dynabeads Protein G beads (invitrogen) in PCR tubes, samples were subject to Tagmentation (5 µl Tagmentation Buffer, 1 µl Tagmentation DNA Enzyme (Illumina), 19 µl Nuclease free water) in order to incorporate sequencing adapters. Lastly, DNAs were eluted from the beads and used for library preparation with the Nextera DNA library preparation kit (Illumina).

**Locus-specific bisulfite sequencing.** Genomic DNA was purified with phenol-chloroform and subjected to bisulfite conversion according to the EZ DNA Methylation-Direct™ Kit (Zymo Research). Then, primer pairs specific for each investigated enhancer were used for PCR amplification with the EpiTaq HS polymerase (TaKaRa). Finally, the resulting amplicons were gel-purified, subjected to blunt-end cloning, and analyzed by Sanger sequencing. The target sequence and the bisulfite primer for the selected PGCLC enhancers are listed in Supplementary Data 6.

**ChIP-bisulfite sequencing.** ChIP-bisulfite experiments were performed as described in[89] with slight modifications. Firstly, the ChIP protocol described above was followed. After the final DNA purification, all the resulting H3K4me1 and H3K4me2 ChIP DNAs and 200 ng of the corresponding input DNA were subjected to bisulfite conversion according to the EZ DNA Methylation-Direct™ Kit (Zymo Research). Then, the subsequent amplification, purification, and sequencing steps were performed as described for the locus-specific bisulfite sequencing experiments.

**Genome-wide bisulfite sequencing.** Bisulfite sequencing libraries were prepared from column-purified DNA of d2 EpiLC using the PBAT method[90]. Briefly, for the bisulfite conversion the instructions of the EZ Methylation Direct MagPrep Kit

(Zymo) were followed. After purification, bisulfite-converted DNAs were eluted from MagBeads directly into 39 µl of first-strand synthesis reaction master mix (1× Blue Buffer (Enzymatics), 0.4 mM dNTP mix (Roche), 0.4 µM 6NF preamp oligo (IDT)), heated to 65 °C for 3 min and cooled on ice. 50 U of klenow exo- (Enzymatics) was added and the mixture incubated on a thermocycler at 37 °C for 30 min after slowly ramping from 4 °C. Reactions were diluted to 100 µl and 20 U of exonuclease I (NEB) added and incubated at 37 °C before purification using a 0.8:1 ratio of AMPure XP beads. Purified products were resuspended in 50 µl of second strand mastermix (1× Blue Buffer (Enzymatics), 0.4 mM dNTP mix (Roche), 0.4 µM 6NR adapter 2 oligo (IDT) then heated to 98 °C for 2 min and cooled on ice. 50 U of Klenow exo- (Enzymatics) was added and the mixture was incubated on a thermocycler at 37 °C for 90 min after slowly ramping from 4 °C. Second strand products were purified using a 0.8:1 ratio of AMPure XP beads and resuspended in 50 µl of PCR master mix (1x KAPA HiFi Readymix, 0.2 µM PE1.0 primer, 0.2 µM iTAG index primer) and amplified with 12 cycles. The final libraries were purified using a 0.8:1 volumetric ratio of AMPure XP beads before pooling and sequencing. All libraries were prepared in parallel with the pre-PCR purification steps carried out using a Bravo Workstation pipetting robot (Agilent Technologies).

**RNA isolation, cDNA synthesis and RT-qPCR.** Total RNA from ESC, EpiLC, and EpiSC was extracted following the protocol of the innuPREP DNA/RNA mini kit (Analytik Jena), while for the RNA extraction from d2 EB, d4 EB and sorted PGCLC the ReliaPrep™ RNA Miniprep Systems (Promega) was used. cDNAs were generated using the ProtoScript II First Strand cDNA Synthesis Kit and Oligo(dT) primers (New England Biolabs). RT-qPCR were performed on the Light Cycler 480II (Roche) with the primers listed in Supplementary Data 6 using Eef1a1 and Hprt as housekeeping controls.

**RNA-seq and scRNA-seq.** Total RNA from WT and dCD EpiLC (2 replicates) and EpiSC (2 replicates) were purified as described above. Bulk RNA-seq were generated following the protocol of the TruSeq stranded kit (Illumina).

For scRNA-seq, EpiSC were dissociated with Accutase and all other cell types with TripleExpress. Cells were then centrifuged and resuspended in PBS containing 0.04% BSA. Next, cells were passed through a strainer, the cell concentration was determined and the scRNA-seq libraries were prepared using the Chromium Single-Cell Gene Expression (10× Genomics) according to the Single-Cell 3′ Reagents Kit (v2) protocol for the single-cell experiment of the PGCLC differentiation (Fig. 1) and the Single-Cell 3′ Reagents Kit (v3) protocol for the d4 EB from R1 WT and dCD cells (Fig. 6).

**Western Blot.** Nuclei were isolated by incubating cells with lysis buffer (20 mM Tris pH 7.6, 100 mM NaCl, 300 mM sucrose, 3 mM MgCl$_2$) containing freshly added protease inhibitors for 10 min at 4 °C and then centrifuged for 10 min at 4 °C and 1680 g. The resulting pellets, containing the cell nuclei, were treated with a high salt buffer (20 mM Tris pH 8.0, 400 mM NaCl, 2 mM EDTA pH 8.0) and disrupted with a glass homogenizer on ice. After incubation on ice for 30 min and centrifugation (24000 × g for 20 min at 4 °C), supernatants were collected and protein concentration was estimated by a BCA-Assay. 20 µg of the resulting protein extracts were heated in Laemmli buffer at 95 °C for 5 min, loaded on 4–15% Mini-PROTEAN® TGX™ Precast Protein Gels (Bio-Rad) and transferred (190 mM glycine, 25 mM Tris, 20% Methanol, 0.1% SDS) to a PVDF membrane. After blocking with 5% milk, the primary antibody (Supplementary Data 6) was incubated overnight at 4 °C, and the secondary antibody (coupled to horseradish peroxidase (HRP)) for 1 h at RT with washes in between. Finally, proteins were visualized using the lumi-light plus western blotting substrate (Roche). Full scan images are provided in the Source Data file.

**scRNA-seq data processing.** The 10x Genomics scRNA-seq data generated in this study across various stages of PGCLC differentiation can be explored by opening the.cloupe file available through GEO [https://www.ncbi.nlm.nih.gov/geo/query/acc.cgi?acc=GSE155088] with the Loupe Cell Browser (https://www.10xgenomics.com). UMIs were counted using NCBI:GCA_000001635.6 and cellranger-2.1.0[91]. The resulting UMI values were aggregated into a single matrix with default normalization ("–normalize=mapped") (Supplementary Data 1).

For the scRNA-seq data from E4.5–E6.5 mouse embryos[43] the count matrix (GSE121650) was normalized to FPKM (edgeR, Ensembl gene annotation, v87). Then, the previously generated lineage assignments[43] were used to solely select epiblast cells for further analysis.

**scRNA-seq data analysis.** The code used to define PGCLC genes is available through Github (https://github.com/ToreBle/Germline_competence). Briefly, monocle2[92] was used to evaluate the in vitro scRNA-seq data generated across the different PGCLC differentiation stages. Therefore, k-means clustering was performed on the t-SNE plots (with $k = 3$ for d2 EB and $k = 4$ for d4 EB). From the resulting clusters, those containing PGCLC were identified by the enrichment of previously defined core PGC genes from d4/d6 PGCLC and E9.5 PGCs[93]. To determine the cellular identity of the remaining clusters found within the EB, the expression of lineage-specific markers identified in E8.25 mouse embryos[42] was

used. To this end, all markers with a log2FoldChange >2.5 were considered. Each EB cluster was annotated as equivalent to the mouse embryonic tissue for which we observed the most significant enrichment in the expression of the corresponding marker genes.

The ESC, EpiLC, EpiSC, and PGCLC gene sets were defined by differential expression using *Seurat*[94] and the negbinom option for differential expression testing. ESC, EpiLC, and EpiSC genes were determined by differential expression between the stage indicated vs the remaining two (e.g. ESC vs. d2 EpiLC and EpiSC) From this analysis only the genes upregulated in the particular stage (adjusted *p*-value < 0.005), a high expression in the stage of interest (more than 20%, pct.1 >0.2) and a low expression distribution in the others (less than 40%, pct.2 <0.4) were considered. Similar PGCLC genes were determined by differential expression between the d2 + d4 PGCLC clusters and the remaining clusters from d2 and d4 EB. Again, from this analysis only the genes upregulated in PGCLC (adjusted *p*-value < 0.005) and with high expression in PGCLC (expressed in 20% of the PGCLC) and a low expression distribution in the other analyzed cells (expressed in less than 40% of non-PGCLC EB cells) were considered. In the case of PGCLC genes this resulted in 389 PGCLC genes (Supplementary Data 2). To quantify the expression of PGCLC genes in different analyses, the UMI count matrix (of the in vitro stages) or FPKM-normalized data (of the in vivo stages) were used to calculate the mean expression of all PGCLC genes within a cell or the mean expression of each PGCLC gene across all cells of a stage.

For the RNA velocity analysis (Supplementary Fig. 1f), spliced and unspliced read counts were obtained with *kallisto*[95] and *bustools*[96], parsed into R-3.6.1 to create a *Seurat* object and t-SNE plot which was then overlaid by the RNA velocity calculations from *velocyto.R*[97].

The estimation of transcriptional noise for the epiblast stages in vivo and in vitro was performed like in[44]. First, the 500 most variable genes for each stage were selected and pairwise compared by Spearman correlations. Then the Spearman correlation values were transformed into distance measurements ($\sqrt{(1 - \rho)/2}$) that were considered to represent transcriptional noise.

For the single-cell RNA-seq data set of the d4 EB from R1 WT and dCD cells, single cells with a high percentage of mitochondrial gene expression were discarded, by considering only single cells with at least 2000 expressed genes. The clusters (Fig. 6b) were identified in the combined dataset of R1 WT and dCD cells by shared nearest neighbor (SNN) modularity optimization-based clustering algorithm from Seurat with a resolution of 0.3. This resulted in 9 clusters that were classified based on the marker gene expression of E8.25 mouse embryos[42]. Those clusters that did not show any specific marker gene expression from this cluster (e.g. 2-cell-like) were identified by differentially expressed genes within the cluster. The *Prdm1orDppa3 + /Klf4-* cells were defined using the following gene expression values; *Dppa3* >0.1 OR *Prdm1* >1 AND *Klf4* <1. The single-cell expression correlation values were determined by the Spearman correlation of the top 1000 most variable genes in each cluster and the correlation of their expression among the cells belonging to the same cluster.

**ChIP-seq data processing**. For ChIP-seq data processing, single-end reads were mapped to the mouse genome (*mm10*) using *BWA*[98]. After duplication removal with the *MarkDuplicates* function from the *Picard* tools (http://broadinstitute.github.io/picard/), reads within blacklisted regions (https://www.encodeproject.org/annotations/ENCSR636HFF/) were discarded and the aligned reads were normalized with *deeptools-3.3.1*[99] to 1x sequencing depth (as RPGC: Reads per genomic content). An overview of all the ChIP-seq experiments performed in this study is listed in Supplementary Data 6 and, briefly, it includes the following datasets:

- H3K4me1 ChIP-seq experiments in ESC (n = 2), EpiLC (n = 4) and EpiSC (n = 2) performed in R1 and E14Tg2a cell lines.
- H3K4me2 ChIP-seq experiments in ESC (n = 2), EpiLC (n = 3) and EpiSC (n = 3) performed in R1 and E14Tg2a cell lines.
- H3K4me3 ChIP-seq experiments in ESC (n = 2), EpiLC (n = 2) and EpiSC (n = 2) performed in R1 and E14Tg2a cell lines.
- H3K27ac ChIP-seq experiments in ESC (n = 2), EpiLC (n = 4) and EpiSC (n = 2) performed in R1 and E14Tg2a cell lines.
- NANOG-HA ChIP-seq experiments in EpiSC were performed as two biological replicates in E14Tg2a.
- Additional ChIP-seq experiments were performed as single replicates (Supplementary Data 6).

All the processed ChIP-seq data can be explored using the following UCSC browser sessions:

- Comparison of the pluripotent stages (related to Figs. 2, 3): http://genome-euro.ucsc.edu/s/Tore/Comparison%20of%20the%20pluripotent%20stages
- PRDM14/NANOG overexpression (related to Fig. 4): http://genome-euro.ucsc.edu/cgi-bin/hgTracks?hgS_doOtherUser=submit&hgS_otherUserName=Tore&hgS_otherUserSessionName=PRDM14%2FNANOG%20overexpression
- H3K4me1/2 deficient cells and *Otx2*-/- cells (related to Figs. 4, 5): http://genome-euro.ucsc.edu/cgi-bin/hgTracks?hgS_doOtherUser=submit&hgS_otherUserName=Tore&hgS_otherUserSessionName=H3K4me1%2F2%20deficiency%20and%20Otx2%2D%2F%2D.

**ATAC-seq data processing**. ATAC-seq experiments were performed as two biological replicates for each of the investigated cell types (i.e. ESC, EpiLC, EpiSC). For ATAC-seq data processing, paired-end reads were mapped to the mouse genome (*mm10*) using *BWA*[98]. Read duplicates and reads within blacklisted regions were discarded. Given the concordance of the ATAC-seq replicates (Spearman correlation of a 2 kb window for ESC: 0.86; EpiLC: 0.88; EpiSC: 0.86), BAM files for each stage were merged and converted into bigWig files by normalization to 1× sequencing depth with *deeptools-3.3.1*[99]. The processed ATAC-seq data can be also found in the UCSC browser sessions mentioned above.

**Enhancer definition**. The code used to define enhancers is available through Github (https://github.com/ToreBle/Germline_competence).

PGCLC enhancers: The H3K27ac ChIP-seq data (SOLiD sequencing) from d2 and d6 sorted PGCLC[15] were aligned with the default settings of *novoalignCS* (V1.06.09, Novocraft Technologies). For visualization, replicate BAM files for each stage were merged and normalized to 1 x sequencing depth. Briefly, H3K27ac peaks were called from both replicates of d6 PGCLC with MACS2[100] using broad settings (--broad -m 5 50 --fix-bimodal --extsize 200) and *q*-values <1 × 10$^{-3}$. Next, all peaks within a distance of 1 kb from both d6 PGCLC replicates were merged with *bedtools* (https://github.com/arq5x/bedtools2). The resulting regions were subtracted from blacklisted and promoter regions (± 2 kb of the Ensembl gene annotation, v86) with *bedtools* (Supplementary Fig. 2a). PGCLC enhancers were defined as the subset of the distal H3K27ac peaks identified in d6 PGCLC that could be physically linked to PGCLC genes according to capture HiC data generated in ESC by[47] (Supplementary Data 1). Thereby, distal d6 PGCLC H3K27ac peaks and PGCLC genes were linked if the two anchors of a Capture HiC interaction occurred within 1 Kb of a d6 PGCLC H3K27ac peak and 3 Kb of the PGCLC gene TSS.

For the statistical comparisons of epigenetic signals within PGCLC enhancers in different cell types, we first determined the average signals of the ChIP-seq, ATAC-seq, or genome-wide CpG bisulfite sequencing within ± 1 kb of the PGCLC enhancers in d2 EpiLC and EpiSC using *deeptools-3.3.1*[99]. Then, the effect size of paired Wilcoxon tests were calculated by dividing the z-statistics by the square roots of the sample sizes using *rstatix* and *boots.ci* for the approximation of the confidence intervals (confidence level: 0.95).

EpiLC/EpiSC enhancers: Briefly, H3K27ac peaks were called in both cell types separately with MACS2[100] using broad settings (--broad -m 5 50 --fix-bimodal --extsize 200) and *q*-values < 1 × 10$^{-3}$. The resulting regions were subtracted from blacklisted and promoter regions (± 2 kb of the Ensembl gene annotation, v86) using *bedtools*. EpiLC and EpiSC enhancers were obtained from these regions by selecting only H3K27ac peaks located within 500 kb of the gene sets defined by single-cell RNA-seq for each cell type. The enhancer assignment to the proximal genes was done with *GREAT-4.0.4*[101] and only enhancers with a minimum distance of 3.5 kb to the nearest transcription start site (TSS) were considered.

**CpG methylation analysis**. The analysis of the local bisulfite sequencing experiments performed for selected PGCLC enhancers was evaluated with BISMA[102].

Genome-wide DNA methylation data was analyzed with *Bismark*[103]. However, as the considered data sets were prepared with slightly different protocols, the preprocessing steps were adjusted accordingly: For the whole-genome bisulfite sequencing data from 2i ESC (GSE41923), d2 EpiLC, and EpiSC (GSE70355), the adapter trimming was performed with *Trim Galore* (http://www.bioinformatics.babraham.ac.uk/projects/trim_galore/) using the default settings; for data sets generated by post bisulfite adapter tagging (pbat), either 9 (data from d2 EpiLC and PGCLC (DRA003471)) or 6 bp (genome-wide methylation data generated in this study) were removed. The DNA methylation data from E4.5–E6.5 epiblasts were generated by STEM-seq (GSE76505), and, in this case, adapter sequences were removed with *cutadapt*[104] and *Trim Galore*, respectively. For all the previous samples, reads were mapped with *Bismark-v0.16.1*[103] and *bowtie2-2.2.9*[105], using the -pbat setting for the STEM-seq and pbat samples. For paired-end pbat samples the unmapped reads were remapped as single-end reads. Then, for each cell type all available datasets were combined to estimate the CpG methylation levels with the *Bismark methylation extractor*. Finally, only CpGs with a coverage of 3−100 reads were considered.

**CpG methylation heterogeneity and scNMT-seq analysis**. With the scNMT-seq method the transcriptome, methylome (CpG methylation), and chromatin accessibility (GpC methylation) are recorded from the same single-cell[106]. Among all the single cells analyzed in E4.5, E5.5, and E6.5 mouse embryos by[43], we only considered those assigned to the epiblast by scRNA-seq (https://github.com/rargelaguet/scnmt_gastrulation). Then, epiblast cells were additionally filtered according to their methylome (GSE121690) and only cells with a CpG coverage >10$^2$ at PGCLC enhancers were considered. This resulted in 258 epiblast cells from E4.5, E5.5, and E6.5 with high-quality single-cell CpG methylation data.

The single-cell CpG methylation for each cell was stored in a bedGraph file. From each bedGraph file the genome-wide CpG methylation levels of individual cells were determined as the mean methylation of all covered CpGs in each single-cell. For the analysis of mCpG levels and CpG coverage within PGCLC enhancer

Group I, the genome-wide bedGraph files were subtracted to obtain bedGraph files with the PGCLC enhancer regions (Supplementary Data 3) using *bedtools* (https://github.com/arq5x/bedtools2) from which the mean methylation of all covered CpGs was determined in each single-cell.

CpG methylation heterogeneity was estimated with the *PDclust* package[70]. The number of CpGs covered in each pair of cells resulted in approximately 150 CpGs for each pairwise comparison when PGCLC enhancers were considered. Then, the average of the absolute difference in the methylation values for all the CpGs covered for each pairwise comparison were computed as a dissimilarity matrix.

For the scRNA-seq data, the mean expression of all PGCLC genes (FPKM-normalized) linked to Group I PGCLC enhancers was determined per cell.

**Bulk RNA-sequencing**. Public available RNA-seq data generated across different time-points of EpiLC differentiation[107], *Otx2*[-/-] EpiLC[57], and *Prdm14*[-/-] EpiLC[14] were mapped with *STAR*[108] to the mouse reference genome (Ensembl gene annotation, v99), and reads within genes were counted with *featureCounts*[109]. The rlog normalization was generated with *DESeq2*[110].

For the bulk RNA-seq data generated in R1 WT and dCD cells, we analyzed three replicates for R1 WT and the dCD ESC previously generated by Dorighi et al., 2017, two replicates for R1 WT and dCD d2 EpiLC, and two replicates for R1 WT and dCD EpiSC. The transcript abundance was determined with the bootstrap mode of *kallisto* (-b 100)[95] and the differential expression analyses were performed with *sleuth* on the gene-level (Ensembl gene annotation, v96)[95,111].

**Reporting summary**. Further information on research design is available in the Nature Research Reporting Summary linked to this article.

## Data availability

The data that support this study are available from the corresponding authors upon reasonable request. The NGS data generated in this study have been deposited in the GEO database under accession code GSE155089. The enhancer sets of this study are provided in Supplementary Data 3. Public data sets analyzed in this study include single-cell RNA-seq in E8.5 mouse embryo: "EpiLC-E-MTAB-6153"; bulk RNA sequencing: "EpiLC-GSE117896", "*Otx2*[-/-]-GSE56138", "*Prdm14*[-/-]-DRA003471", and "Mll3/4dCD-GSE98063"; WGBS: "GSE70355", WGBS in WT and dCD ESC: "GSE118314"; H3K27ac in PGCLC: "GSE60204"; Mll3 ChIP-seq in ESC: "GSE98063"; DNAse I in PGC: "GSE109767"; STEM-seq in E4.5–E6.5: "GSE76505"; single-cell RNA-seq and single-cell DNA methylation: "GSE121708". Source data are provided with this paper.

## Code availability

Tore Bleckwehl (2021). ToreBle/Germline_competence: Bleckwehl et al., 2021 (Version 1.0.0) [Computer software]. Zenodo. https://doi.org/10.5281/zenodo.5295059.

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

## Acknowledgements

We thank the Rada-Iglesias lab members for insightful comments and critical reading of the manuscript, Antonio Simeone, and Christa Bücker for generously providing the *Otx2*⁻/⁻ cell line. Library preparations and next-generation sequencing experiments were performed in the NGS Core Facility of the Cologne Center for Genomics (CCG). Computational analyses were performed on the Cologne High Efficient Operating Platform for Science (CHEOPS). Cell sorting and Flow cytometry experiments were performed in the FACS Facility of the Center for Molecular Medicine (CMMC) and the FACS & Imaging Core Facility at the Max Planck Institute for Biology of Ageing. Tore Bleckwehl was supported by a doctoral fellowship from the Studienstiftung des deutschen Volkes (Germany). Giuliano Crispatzu is supported by funding within the CRU329 (DFG 386793560). Kaitlin Schaaf was supported by a Research Internships in Science and Engineering (RISE) Scholarship of the Deutscher Akademischer Austauschdienst (DAAD). Work in the Rada-Iglesias laboratory was supported by CMMC intramural funding (Germany), the German Research Foundation (DFG) (Research Grant RA 2547/2-1), "Programa STAR-Santander Universidades, Campus Cantabria Internacional de la convocatoria CEI 2015 de Campus de Excelencia Internacional" (Spain), the Spanish Ministry of Science, Innovation, and Universities (Research Grant PGC2018-095301-B-I00) and the European Research Council (ERC CoG "PoisedLogic"; 862022). AB and MM were supported by the Spanish Ministry of Science, Innovation, and Universities (BFU2017-84914-P), and the CNIC is supported by the Instituto de Salud Carlos III (ISCIII), the Spanish Ministry of Science, Innovation, and Universities, and the Pro CNIC Foundation, and is a Severo Ochoa Center of Excellence (SEV-2015-0505).

## Author contributions

Conceptualization, T.B., A.R.-I.; Experimental investigation, T.B., K.S., G.C., P.R., M.B., S.J.C, L.B., A.B., M.L., K.D; Data analysis: T.B., G.C.; Writing, Review & Editing, T.B., A.R.-I.; Resources, W.F.J. vI, M.M., J.W., W.R., A.R.-I.; Supervision and Funding Acquisition, A.R.-I.;

## Funding

## Competing interests

The authors declare no competing interests.
