## [Peer Review File · Nature Communications]

Reviewers' comments:

Reviewer #1 (Remarks to the Author):

Bleckwehl et al investigate enhancer regulation during the derivation of primordial germ cell-like cells (PGCLCs) in vitro. They begin by detailing gene expression changes as cells exit naïve pluripotency and acquire competence for PGCLC induction. They then describe a set of PGCLC enhancers and detail the changes in some chromatin features at these during the process of PGCLC derivation. This leads to a description of 'enhancer priming' – an apparent relative maintenance of H3K4me1 and loss of CpG methylation – at these enhancers, albeit this concept is never particularly developed mechanistically. Similarly the concept of 'epigenetic heterogeneity' in PGC competent tissues is entertained but not really satisfactorily demonstrated. Next the authors detail changes at some specific PGCLC enhancers. Finally, they use their previously published MLL3/4 catalytically deficient ES cell line to interrogate the consequences of a reduction in H3K4me1 throughout the entire process of PGCLC induction from ES cells. This suggests that H3K4me1 likely plays an important (but non-essential) role during the transition from ES cells to PGCLCs (via EpiLC), and contributes to the enhanced PGCLC competence of Otx2 knockout cells.

How PGC competence is regulated is an important topic of wide-ranging interest, and so any study which advances current understanding would be a strong candidate for publication. While some aspects of this study revealed intriguing data which could be the basis of just such a study, this was unfortunately overwhelmed by a long, verbose and ultimately flawed manuscript. I had major concerns regarding the authors background knowledge of the field, the conceptual underpinning of many of the experiments, as well as some technical aspects of the specific experiments performed. These concerns were amplified by the frequent overinterpretation of the data and the conclusions drawn, which were often not supported by the data. The many inaccuracies and the sheer length of this manuscript made it extremely challenging to review, and as such I suspect that I have not been able to detail all of the issues. I have tried to focus on the major flaws, in the hope that this will help the authors refashion this manuscript and perhaps undertake further experiments that build on its strengths – as I do believe they have some tools and approaches which would allow them to answer interesting questions. Unfortunately, as the manuscript currently stands it is not appropriate for publication and requires such a complete overhaul that a fresh review process would be best.

Major issues:

1. Introduction

a) Referencing for the formative state. The authors do not appropriately reference the key publications that detail the formative state hypothesis (Kalkan et al., 2017; Mulas, Kalkan, & Smith, 2017; Smith, 2017)

b) Page 3: 'Nevertheless, only a fraction (typically <20%) of the formative epiblast cells can give rise to PGCs when exposed to the appropriate signals'. This statement is both wrong and incorrectly referenced. Ohinata et al., 2009 demonstrated that essentially the entire epiblast can be converted to PGCLCs, and that the time window for this efficient induction is E5.5 – E6.25. That essentially all cells are competent for PGCLC induction during this time window, seems to undermine the authors claims that 'the formative epiblast is heterogeneous in terms of its intrinsic germline competence' which appears to be one of the main assumptions underlying the current study.

c) Page 4 'primordial germ cell-like cells (PGCLC). This system revealed transcription factors (TFs) and epigenomic reprogramming events involved in PGC specification and led to a better understanding of the mouse peri-implantation transitions in general'. This statement appears to reveal a fundamental misinterpretation of the literature. The critical transcription factors for PGC specification were all discovered in vivo (for instance, Ohinata et al., 2005; Weber et al., 2010; Yamaji et al., 2008) prior to the discovery of the PGCLC system. Epigenetic reprogramming again, has been best studied in vivo. The early stages of epigenetic reprogramming (in some nomenclatures 'reprogramming 1') have been characterised in vivo (Seki et al., 2005; 2007) and confirmed in vitro. The main epigenetic reprogramming event ('reprogramming 2') that occurs on colonisation of the gonads is particular difficult to study in vitro as there is no well-defined model for this stage of PGC development – and again in vivo observations have been much more informative here (Hajkova et al., 2008; 2010; P. W. S. Hill et al., 2018).

d) Naïve ESCs grown in 2i are not largely devoid of CpG methylation. Depending on the exact culture conditions there is an approximately 3-4 fold reduction compared to the levels observed in serum grown ESCs (Ficz et al., 2013; Habibi et al., 2013; Leitch et al., 2013)

e) Page 6 'Germline specification is impaired'. This an overclaim and one of many examples in which the authors switch between the in vivo process of PGC specification (occurring from the competent epiblast), and the in vitro observations made during PGCLC induction from EpiLCs. There is no clear evidence that PGCLC induction follows the same trajectory as PGC specification. It certainly does result in PGCLCs that have an identity similar to E9.5 PGCLCs, some of which are functional to make gametes. While the PGCLC system is very useful for studying many aspects of PGC biology, it is not clear that observations in this system are always an accurate reflection of the in vivo mechanisms, in fact there are clear examples in which this is not the case (R. J. Hill & Crossan, 2019; Senft, Bikoff, Robertson, & Costello, 2019).

f) Page 7: 'Previous work indicated that the acquisition of germline competence by the formative epiblast (i.e. E5.5-6.5 epiblast; EpiLC)... another possibility that has not been thoroughly investigated is that germline competence is associated with a few cells of the formative epiblast in which the naïve expression program is totally or partially retained'. In fact, the reference cited makes quite clear that the naïve network is dismantled at the single cell level. In addition, there is a huge amount of evidence that argues against a small subset of naïve cells maintaining expression of the naïve network – including observations in vivo (staining of these markers at relevant time points, the various reporter mice used in the field, scRNA-seq papers) and in vitro (the PGCLC system). As such

the authors create a false dichotomy. In the end their analysis supports the prevailing view and adds no novelty. I would remove from any subsequent manuscript.

2. Definition of priming. This leads to much confusion throughout the paper. PGCLC enhancers are marked with H3K4me1 and have low CpG methylation in naïve ES cells. H3K4me1 is lost and CpG increased as cells differentiate towards EpiSC (via an EpiLC state). It is not clear whether PGCLC enhancers are actively maintained in a 'primed' state or whether this simply represents a differentiation process which is interrupted by PGCLC induction. The language the authors employ on this topic does not help, and seems to be focussed on employing fashionable jargon rather than explaining their own observations. Statements like - 'they transiently acquire a primed state in EpiLC' (line 12) – do seem to suggest an active process, although the authors never really show this or give a clear indication of what they mean by 'priming' or 'acquisition of a primed state'. The statement 'Therefore, the transient priming of the PGCLC enhancers in formative epiblast cells could endow them with permissive chromatin features' on Page 18 is another example, and combines uncertainty regarding the mechanisms of priming with inaccurate description of the data – the authors have not undertaken studies in epiblast cells. Finally, it is not clear to what extent the authors think this is a PGCLC specific phenomenon?

3. ChiP-seq data

This is apparently generated in EBs rather than PGCLCs. Given that the majority of these EBs (based on the authors own data) are not PGCLCs this calls into question all of the interpretations based on these datasets.

4. 'faithfully recapitulates PGC specification' (Line 7)

As mentioned above (1d) this has not in fact been demonstrated. What has been shown is that PGCLCs can be induced which bear the hallmark of E9.5 PGCs. Whether the specification process is the same as PGC specification is not known. This has a major implications for the design of the study. It really is not clear what 'active PGCLC enhancers' designated using an 'H3K27ac ChIP-seq data generated in d2 and d6-sorted PGCLC' (Page 10) represent with regards in vivo development. Based on current evidence these might represent enhancers active in E9.5 PGCs. If this is the case, then these could readily be identified in actual E9.5 PGCs – of which thousands are readily obtained from a single litter of embryos. Whether these would be relevant to PGC specification is of course a separate question.

5. PGCLC enhancers

a) The decision to only focus on those enhancers near the pre-selected PGC genes also seems to restrict the analysis – especially as in the current version of the manuscript it is not immediately clear how these PGCLC genes are defined (i.e. Line 8: 'PGC markers (e.g. Nanog , Tfap2c , Prdm14,

Prdm1, Dppa3) (Fig. 1d)'. As these 'PGC markers' are discussed quite extensively, it would be helpful if in the text the authors indicate how many genes are in this gene-set and how they are chosen.

b) The large number of enhancers could be due to complex/redundant enhancer usage but also could be due to overcalling of enhancers – is this possible to interrogate in the data?

c) There are a number of genes that retain H3K27ac in both EpiLCs and EpiSCs – which genes are these?

d) Page 11 'All these active chromatin features decreased upon exit from naïve pluripotency, but, H3K4me1 was partly retained in EpiLC in comparison to EpiSC (Fig. 2d, Supplementary Fig. 2c)'. As this is major claim in the paper it seems important to properly defined what 'partially retained' means in the main text. The difference between H3K4me1 profiles in Figure 2D in EpiSC vs EpiLC appears very minor. Again, the most notable pattern in Figure 2f is the similarity in the CpG methylation between EpiLC and EpiSC (when compared with ESC) and this does not fit the description in the text which states 'intermediate' and 'hypermethylated'. Similarly, the description of Figure 3a as showing 'intermediate' levels in E5.5 does actually seem to accurately reflect the data. As such the conclusion on Pages 12-13, seems far too strong and therefore amounts to an overclaim based on the data presented. e) This is another example of the uncertainty created by the terminology of 'priming of PGCLC enhancers' – this implies an active and targeted process that is particular to this state. In fact, the authors show a transition from ESC to EpiSC, with the gradual loss of H3K4me1 and gain of CpG methylation (both of which appear largely complete by the EpiLC stage – rather than a distinctive intermediate state).

6. Epigenetic heterogeneity:

a) Figure 3e – the claim that 'CpG methylation heterogeneity of the E5.5 epiblast was more pronounced for PGCLC enhancers than for other enhancers or the whole genome' is not clearly supported by the data. The degree of epigenetic heterogeneity seems very similar at EpiLC enhancers, indicating this is not a specific feature of PGCLC enhancers. Similarly the data presented in Figure 3i is difficult to interpret without comparing with non-PGCLC enhancers.

b) Page 15 – again the conclusions here are far too strong based on the data presented. For instance, not all the data presented is from 'formative epiblast'. What is meant by H3K4me1 variability in this example? The authors cannot comment on heterogeneity between epiblast cells of the same embryo (or any cells for chromatin marks). Does higher H3K4me1 correlate with lower DNA methylation?

7. Enhancer deletion experiments.

The data for the enhancer deletions is difficult to interpret as:

a) only Stella-GFP is used to designate PGCLCs and this is not specific. Rather the authors could be measuring effects on Stella regulation quite independent of PGCLC derivation efficiency.

b) the reduction in expression of *Esrrb* does not impact PGCLC formation, and this is the one gene with a known role in PGCs (Mitsunaga et al., 2004).

c) The generalisations around regulation of enhancers 'PGCLC enhancers frequently control the expression of their target genes already in ESC, further supporting that a significant set of enhancers is functionally shared between naïve pluripotency and PGCLC' (Page 17) again does not seem to reflect the data presented. Enhancer regulation for both *PRDM14* and *Esrrb* seems quite different in PGCLCs versus ES cells and it is not clear how the authors can state 'frequently' or 'significant' here?

8. MLL3/4 catalytically deficient ES cell experiments.

The most interesting aspect of this study is the extension of their previous work, using the various cell lines generated. However, what is not clear is if the effects observed are specific to EpiLC and PGCLC generation, or if this is a general way in which enhancers are regulated during post-implantation development. In vivo data is sadly lacking from this manuscript, and would significantly improve it, as the regulation of the 'PGCLC state' is only really interesting in so far as it might correspond to elements of actual PGC biology. The approach in (Zhang et al., 2018) might well be feasible, in which mutant ES cells are used to make chimaeras, and PGC specification is compared between mutant and endogenous wildtype cells (although properly 'timing' any defect might still be challenging).

b) the connection between H3K4me1 and DNA methylation is not so clear – in particular despite an apparently clear reduction in H3K4me1 in EpiLC (Figure 6C) the change in methylation in EpiLC looks marginal at best (Figure 6d). Again the description of the data is unclear. On page 20, the authors state 'no longer acquire a primed state in EpiLC' – in fact, there is reduced H3K4me1 in ESCs and this state is then inherited in EpiLC rather than 'acquired'. As such, whether the subsequent reduction in PGCLC generating ability is an issue with PGCLC induction– or rather an epigenetic defect inherited from the ESC state is not clear.

c) Figure 6f would benefit from showing all of the histone marks, in all of the different cell types, as the current presentation is difficult to interpret.

d) The ChIP in d4 EBs is difficult to interpret as there are fewer PGCLCs in these EBs, and so the comparison is not fair. The same is true of the RNA-seq analysis, in which it is not surprising that PGCLC genes are reduced, the question is whether less PGCLCs are made, or whether there are defects in these PGCLCs – and unfortunately this is not answered by the analysis.

e) With these open questions the conclusion 'our results show that H3K4me1/2 is required for proper PGCLC specification and supports the importance of PGCLC enhancer priming for germline competence' is once more not supported by the data. The authors present no data that can convincingly be tied to PGC specification in vivo.

9. OTX2 data.

This is an interesting connection with previous work from the Chambers lab. However, again the data is not clearly described, which leads to inconsistencies in the story.

a) There seems to be little if any difference in PGCLC induction on day 8, despite apparent retention of H3K4me1 at PGCLC enhancers. This does not seem to fit with authors' model.

b) H3K4me1 levels in Day 2 WT EpiLC and Day 4 Otx2 KO looks comparable, and yet there is vastly different PGCLC induction efficiency, indicating that Otx2's action is independent of H3Kme1.

c) A further issue which is not addressed is whether the increase in H3K4me1 at PGCLC enhancers is already present in Otx2 null ESCs.

d) General claims regarding DNA methylation based on analysis of a single enhancer are not appropriate.

e) The reduction in PGCLC induction efficiency in dCD Otx2 null cells is an interesting result. However, this experiment does seem less clear-cut than in Figure 7a – in which PGCLC induction efficiency is much higher and more consistent in Otx2 null lines. This makes interpretation of the double knockout data more challenging. The relevance of the subsequent DNA methylation data is not very clear.

Other points:

1. Page 8 . 'Remarkably, these subclusters were similar to the extraembryonic tissues (i.e. extraembryonic ectoderm, extraembryonic mesoderm and endothelium) that surround PGCs in the proximo-posterior end of the mouse embryo following germline specification in vivo (Fig. 1c, Supplementary Fig. 1a)' – this is not at all clear from the data shown, which appears to rely on cherry-picked genes. Given that this is a surprising finding, could the authors present more clearly how similar non-PGCs are to the cell types indicated?

2. 'Therefore, the extinction of the naïve program seems to be necessary but not sufficient for the acquisition of germline competence, suggesting that differences, other than transcriptional, should exist between competent (EpiLC, E5.5 epiblast) and non-competent (EpiSC, >E6.5 epiblast) epiblast cells' – it is very difficult to ascertain the exact point the authors wish to make here. Of course, it is not sufficient simply to extinguish the naïve program to acquire germline competence – all other cell types (apart from the formative epiblast/EpiLC) have extinguished the naïve program and are not competent for germline induction. This statement and the paragraph in which it is embedded is very unclear, and seems to serve only to make the point that EpiSCs are less competent than EpiLC for PGC induction, which is widely appreciated.

3. When epigenomic datasets are compared between different studies are the cell lines cultured in similar conditions? (For instance, ChIP-seq data obtained in serum grown ESCs is not relevant for 2i grown in ESCs, which from an epigenomic point of view are essentially a different cell type).

4. Figure 3b – is 33% correct? It looks like there is 66% similarity depicted here?

5. Figure 5b – how many replicates were performed for the ChIP experiments? More generally it would be helpful if the number of replicates for all ChIP experiments was clearly shown.

6. Based on all the above issues, the title of the manuscript is clearly not appropriate.

Ficz, G., Hore, T. A., Santos, F., Lee, H. J., Dean, W., Arand, J., et al. (2013). Short Article. *Cell Stem Cell*, 1–9. <http://doi.org/10.1016/j.stem.2013.06.004>

Habibi, E., Brinkman, A. B., Arand, J., Kroeze, L. I., Kerstens, H. H. D., Matarese, F., et al. (2013). Short Article. *Cell Stem Cell*, 13(3), 360–369. <http://doi.org/10.1016/j.stem.2013.06.002>

Hajkova, P., Ancelin, K., Waldmann, T., Lacoste, N., Lange, U. C., Cesari, F., et al. (2008). Chromatin dynamics during epigenetic reprogramming in the mouse germ line. *Nature*, 452(7189), 877–881. <http://doi.org/10.1038/nature06714>

Hajkova, P., Jeffries, S. J., Lee, C., Miller, N., Jackson, S. P., & Surani, M. A. (2010). Genome-wide reprogramming in the mouse germ line entails the base excision repair pathway. *Science (New York, NY)*, 329(5987), 78–82. <http://doi.org/10.1126/science.1187945>

Hill, P. W. S., Leitch, H. G., Requena, C. E., Sun, Z., Amouroux, R., Roman-Trufero, M., et al. (2018). Epigenetic reprogramming enables the transition from primordial germ cell to gonocyte. *Nature*, 555(7696), 392–396. <http://doi.org/10.1038/nature25964>

Hill, R. J., & Crossan, G. P. (2019). DNA cross-link repair safeguards genomic stability during premeiotic germ cell development. *Nature Genetics*, 51(8), 1283–1294. <http://doi.org/10.1038/s41588-019-0471-2>

Kalkan, T., Olova, N., Roode, M., Mulas, C., Lee, H. J., Nett, I., et al. (2017). Tracking the embryonic stem cell transition from ground state pluripotency. *Development (Cambridge, England)*, 144(7), 1221–1234. <http://doi.org/10.1242/dev.142711>

Leitch, H. G., McEwen, K. R., Turp, A., Encheva, V., Carroll, T., Grabole, N., et al. (2013). Naive pluripotency is associated with global DNA hypomethylation. *Nature Structural & Molecular Biology*, 20(3), 311–316. <http://doi.org/10.1038/nsmb.2510>

Mitsunaga, K., Araki, K., Mizusaki, H., Morohashi, K.-I., Haruna, K., Nakagata, N., et al. (2004). Loss of PGC-specific expression of the orphan nuclear receptor ERR-beta results in reduction of germ cell number in mouse embryos. *Mechanisms of Development*, 121(3), 237–246. <http://doi.org/10.1016/j.mod.2004.01.006>

Mulas, C., Kalkan, T., & Smith, A. (2017). NODAL Secures Pluripotency upon Embryonic Stem Cell Progression from the Ground State. *Stem Cell Reports*, 9(1), 77–91. <http://doi.org/10.1016/j.stemcr.2017.05.033>

- Ohinata, Y., Ohta, H., Shigeta, M., Yamanaka, K., Wakayama, T., & Saitou, M. (2009). A signaling principle for the specification of the germ cell lineage in mice. *Cell*, 137(3), 571–584. <http://doi.org/10.1016/j.cell.2009.03.014>
- Ohinata, Y., Payer, B., O'carroll, D., Ancelin, K., Ono, Y., Sano, M., et al. (2005). Blimp1 is a critical determinant of the germ cell lineage in mice. *Nature*, 436(7048), 207–213. <http://doi.org/10.1038/nature03813>
- Seki, Y., Hayashi, K., Itoh, K., Mizugaki, M., Saitou, M., & Matsui, Y. (2005). Extensive and orderly reprogramming of genome-wide chromatin modifications associated with specification and early development of germ cells in mice. *Developmental Biology*, 278(2), 440–458. <http://doi.org/10.1016/j.ydbio.2004.11.025>
- Seki, Y., Yamaji, M., Yabuta, Y., Sano, M., Shigeta, M., Matsui, Y., et al. (2007). Cellular dynamics associated with the genome-wide epigenetic reprogramming in migrating primordial germ cells in mice. *Development (Cambridge, England)*, 134(14), 2627–2638. <http://doi.org/10.1242/dev.005611>
- Senft, A. D., Bikoff, E. K., Robertson, E. J., & Costello, I. (2019). Genetic dissection of Nodal and Bmp signalling requirements during primordial germ cell development in mouse. *Nature Communications*, 10(1), 1089–11. <http://doi.org/10.1038/s41467-019-09052-w>
- Smith, A. (2017). Formative pluripotency: the executive phase in a developmental continuum. *Development (Cambridge, England)*, 144(3), 365–373. <http://doi.org/10.1242/dev.142679>
- Weber, S., Eckert, D., Nettersheim, D., Gillis, A. J. M., Schäfer, S., Kuckenber, P., et al. (2010). Critical function of AP-2 gamma/TCFAP2C in mouse embryonic germ cell maintenance. *Biology of Reproduction*, 82(1), 214–223. <http://doi.org/10.1095/biolreprod.109.078717>
- Yamaji, M., Seki, Y., Kurimoto, K., Yabuta, Y., Yuasa, M., Shigeta, M., et al. (2008). Critical function of Prdm14 for the establishment of the germ cell lineage in mice. *Nature Genetics*, 40(8), 1016–1022. <http://doi.org/10.1038/ng.186>
- Zhang, J., Zhang, M., Acampora, D., Vojtek, M., Yuan, D., Simeone, A., & Chambers, I. (2018). OTX2 restricts entry to the mouse germline. *Nature*, 562(7728), 595–599. <http://doi.org/10.1038/s41586-018-0581-5>

Reviewer #2 (Remarks to the Author):

The work by Bleckwehl et al. describes the role of H3K4 methylation in primordial germ cells (PGC) enhancers priming. First they perform single cell RNA-seq to characterize their differentiation system. Then, they accomplish a deep and comprehensive characterization of histone posttranscriptional modifications, DNA methylation, and chromatin accessibility of a set of 511

enhancer regions in the proximity of germline-activated genes. They find that H3K4me1 is partly retained at these enhancers in formative epiblast cells (EpiLC, which can give rise to PGC) in comparison to primed Epiblast cells (EpiSC, which cannot give rise to PGC). They propose that this difference is important to determine germline competence. After that, they demonstrate an important role of Mll3 and Mll4 in controlling the epigenetic state of these enhancers, PGC gene expression and specification. My main concern with this manuscript is the relatively small difference observed between the levels of H3K4me1 in EpiLC and EpiSc, especially in the E14Tg2a ESC line which was selected to perform most of the experiments. Although statistically significant when the whole population of 511 enhancers was compared, differences between EpiLC and EpiSc in figure 2d are really hard to see, especially in the enhancers of the lower half of the heatmap. Same comment can be argued about figure 2e (ATAC-seq), and S2e (H3K9me2). Several comments about this:

- The 511 enhancers were selected based on their proximity to PGC-regulated genes. Since not all close enhancers may control PGC-dependent expression of the linked genes, an important number of the analyzed enhancers may not be related to PGC specification. In fact, when the authors perform 4C at the Prdm14 gene, only the E3 enhancer seems to be a region that strongly contact the Prdm14 promoter, but not E1 and E2.

- The authors should consider to use promoter-capture Hi-C data in order to better select the set of enhancers analyzed. Alternatively, they should use other bioinformatic tools, in addition to enhancer-gene proximity, to select the set of PGC enhancers. For example, the correlation between enhancers H3K27Ac signal and gene expression can be used in addition to proximity.

- A general small decrease of H3K4me signal in EpiSC versus EpiLC is observed in all genome browser views shown (Figure 4a, 4d, S4a), both in the selected enhancer regions but also in other close peaks around. Have the authors analyzed whether reduced H3K4me1 signal in EpiSC occurs in domains? Is H3K4me1 signal generally decreased in EpiSC versus EpiLC? The authors should show control regions where H3K4me1 signal does not change and boxplots or density plots comparing all accessible enhancers H3K4me signal in the two stages. Have the authors verified whether levels of Mll3, Mll4 and Lsd1 proteins are similar in EpiSC and EpiLC.

- Authors state that “differences –EpiLC versus EpiSC – were not observed around the transcription start sites (TSS) of the PGCLC genes”. This is not what I see in Supplementary figure S2h for H3K4me1. I see differences comparable to those of PGCLC enhancers.

- Authors should show levels of H3K4me1 by ChIP-PCR analysis of several enhancers in order to verify the ChIP-seq results.

Other concerns:

- In the Abstract the authors write: “we demonstrate that priming by H3K4me1/2 enables the robust activation of PGC enhancers”. The authors should be more cautious when they refer about the role of H3K4me1. Since PGC specification is not completely abolished and PGC gene expression is not strongly impaired in the dCD mutants they shouldn't say “enables”. In fact, PGC specification is possible in the absence of correct H3K4me1. Therefore, I find more appropriated the way that the authors use in the Discussion: “we propose that priming by H3K4me1/2 might facilitate, rather than

being essential for, enhancer activation and the robust induction of developmental gene expression programs". This facilitating but not essential role should be stated in the abstract.

- The model presented should express the results obtained in a more quantitative manner, other way the model may be misleading. For example, the model (Fig. 7f) shows H3K4me1/2 mark in Naïve pluripotency and wt formative pluripotency, but absence of mark in the primed pluripotency state. This is not what I see in Figure 2d. I see a drastic reduction between ESC and EpiLC and a small reduction, not an absence of the mark, between EpiLC and EpiSC. Same comment about H3K9me2/3. The model shows no H3K9me2/3 mark in the formative pluripotency state and presence of the mark in the primed pluripotency state, which it is not the experimental scenario. Authors should find a better way to show the experimental differences in their model.

Minor points.

- Page 9. Line 4 from the bottom. It should be "Supplementary Fig. 1g" instead of "Fig. 1g".

- Molecular data demonstrating the CRISPR-mediated enhancers deletion should be provided in supplementary figures.

- Are the R1 WT (EpiLC and EpiSC) panels in Figures 6c and S2f identical? If this is the case, this should be mentioned in the figure legend.

- Page 21. Line 3 from the bottom. It should be "Supplementary Fig. 6g" instead of "Supplementary Fig. 6f".

Reviewer #3 (Remarks to the Author):

The manuscript by Bleckwehl et al. shows that enhancer priming by H3K4me1/2 ensures gene expression involved in PGC differentiation. Using single cell analysis and in vitro differentiation system, the authors first confirmed homogenous downregulation of naïve pluripotent gene expression upon differentiation to the formative state. In contrast, the authors also found that enhancers controlling PGC specification had heterogenous DNA methylation, chromatin accessibility and H3K4me1. Among these heterogenous states, the authors demonstrated using MLL3/4 mutants that H3K4me1 plays a major role on priming enhancer for later gene expression in PGC specification. Furthermore, the authors confirmed the priming effect in Otx2-mutant; accelerated PGC

specification in Otx2-mutant was attributable to elevated level of H3K4me1 in the PGC gene enhancers and was abrogated by impairment of MLL3/4 function.

Following the previous report by the authors showing functional involvement of Foxd3 in exit of naïve pluripotent and germ cell specification, this study revealed molecular mechanisms of priming enhancers for PGC specification, which include sufficiently novel findings that have a significant impact on the research field. Specifically, it has been obscure how PGC competence is conferred to formative state pluripotent cells. This study provides a clear answer that H3K4me1(2) by MLL3/4 primes the enhancers of the PGC genes. The experiments are well designed, and the results largely support the author's conclusions. Although there are some comments below to be considered, this reviewer supports publication of this manuscript in Nature Communications.

Specific comments

1) Differences in H3K4me1 and chromatin accessibility between EpiLC and EpiSC look subtle (Figure 2d and e). To help readers understand, it would be better to explain more intensively difference in repressive mark such as DNA methylation and H3K9 methylation. In this context, showing heat maps for H3K9me2/3 on the PGCLC enhancers would be informative.

2) The authors showed a lower mCpG level in PGCLC enhancers in Esrrb and Lrrc31. However, only two loci are not sufficient to conduct the conclusion. How about the enhancers of other genes? Also are these alterations of the mCpG level in Esrrb and Lrrc31 statistically significant?

3) The deletion of the enhancers of Lrrc31 and Klf5 resulted in reduction of PGCLC differentiation. However, there is no report demonstrating that Lrrc31 or Klf5 is essential for PGC specification. The rationality of the functional outcome of the enhancer deletion should be shown.

4) FACS analyses of PGCLC induction are not convincing, especially in Figure S6d (therefore also Figure 6e) and the experiment using Dppa3-GFP. Refinement of the analysis is needed.

5) Following sentences sounds awkward: Regardless, Klf5 and Lrrc31 might represent PGCLC regulators shared between naïve and PGCLC, similarly to other naïve/PGCLC TFs with this dual regulatory role (e.g. Prdm14, Nanog). On the other hand, the Esrrb enhancer deletion moderately reduced the expression of Esrrb in ESC and did not significantly affect PGCLC differentiation (Fig. 4b,c). Nevertheless, the expression of Esrrb was severely diminished in d4 EB (Fig. 4c), demonstrating that this enhancer is needed for the proper induction of Esrrb in PGCLC.

Can the authors revise? The authors should discuss these observations, based on the functionality of these genes in PGC specification.

6) In 4C-seq analysis, the authors only showed Prdm14 locus. Showing several examples other than Prdm14 would be more informative.

Point-by-point Response to the reviewers' comments

Reviewer #1 (Remarks to the Author):

Bleckwehl et al investigate enhancer regulation during the derivation of primordial germ cell-like cells (PGCLCs) in vitro. They begin by detailing gene expression changes as cells exit naïve pluripotency and acquire competence for PGCLC induction. They then describe a set of PGCLC enhancers and detail the changes in some chromatin features at these during the process of PGCLC derivation. This leads to a description of 'enhancer priming' – an apparent relative maintenance of H3K4me1 and loss of CpG methylation – at these enhancers, albeit this concept is never particularly developed mechanistically. Similarly the concept of 'epigenetic heterogeneity' in PGC competent tissues is entertained but not really satisfactorily demonstrated. Next the authors detail changes at some specific PGCLC enhancers. Finally, they use their previously published MLL3/4 catalytically deficient ES cell line to interrogate the consequences of a reduction in H3K4me1 throughout the entire process of PGCLC induction from ES cells. This suggests that H3K4me1 likely plays an important (but non-essential) role during the transition from ES cells to PGCLCs (via EpiLC), and contributes to the enhanced PGCLC competence of Otx2 knockout cells. How PGC competence is regulated is an important topic of wide-ranging interest, and so any study which advances current understanding would be a strong candidate for publication. While some aspects of this study revealed intriguing data which could be the basis of just such a study, this was unfortunately overwhelmed by a long, verbose and ultimately flawed manuscript. I had major concerns regarding the authors background knowledge of the field, the conceptual underpinning of many of the experiments, as well as some technical aspects of the specific experiments performed. These concerns were amplified by the frequent overinterpretation of the data and the conclusions drawn, which were often not supported by the data. The many inaccuracies and the sheer length of this manuscript made it extremely challenging to review, and as such I suspect that I have not been able to detail all of the issues. I have tried to focus on the major flaws, in the hope that this will help the authors refashion this manuscript and perhaps undertake further experiments that build on its strengths – as I do believe they have some tools and approaches which would allow them to answer interesting questions. Unfortunately, as the manuscript currently stands it is not appropriate for publication and requires such a complete overhaul that a fresh review process would be best.

We would like to thank the reviewer for the very detailed and insightful revision of our work. We agree with the reviewer in that the previous version of the manuscript was too long and complex. Moreover, we also appreciate all the reviewer's suggestions regarding literature that we did not properly cited as well as certain statements that could be considered as over-interpretations of the

presented data. Therefore, we have taken in consideration all the comments and suggestions from all three reviewers in order to extensively revise our manuscript, including major changes to the text as well as new and relevant experiments and computational analyses. We think that, thanks to the reviewers' comments, the current manuscript version describes in a more accurate and simple manner our major findings, highlighting how the partial decommissioning of enhancers and the presence of H3K4me1 can be important for *in vitro* germline competence and the subsequent (re)activation of enhancers and their target genes.

Major issues:

1. Introduction

a) Referencing for the formative state. The authors do not appropriately reference the key publications that detail the formative state hypothesis (Kalkan et al., 2017; Mulas, Kalkan, & Smith, 2017; Smith, 2017).

We have added these missing references to the revised manuscript version:

Page 3: *“Furthermore, regardless of their position within the embryo, formative epiblast cells (~E5.5-6.25) (Kalkan et al. 2017; Mulas et al. 2017; Smith 2017) are germline competent when exposed to appropriate signals, but this ability gets lost as the epiblast progresses towards a primed pluripotency state (>E6.5) (Ohinata et al. 2009).”*

Page 6: *“Previous work indicates that the acquisition of germline competence in day 2 (d2) EpiLC entails the complete dismantling of the naïve gene expression program (Mulas et al. 2017; Kalkan et al. 2017),...”*

b) Page 3: ‘Nevertheless, only a fraction (typically <20%) of the formative epiblast cells can give rise to PGCs when exposed to the appropriate signals’. This statement is both wrong and incorrectly referenced. Ohinata et al., 2009 demonstrated that essentially the entire epiblast can be converted to PGCLCs, and that the time window for this efficient induction is E5.5 – E6.25. That essentially all cells are competent for PGCLC induction during this time window, seems to undermine the authors claims that ‘the formative epiblast is heterogeneous in terms of its intrinsic germline competence’ which appears to be one of the main assumptions underlying the current study.

We have to respectfully disagree with the reviewer regarding whether or not the entire epiblast can be converted to PGCLC. Ohinata et al. 2009 used a *Blimp1* reporter to show that around 50% of epiblast cells can be converted to PGCLC based on *Blimp1* reporter expression when exposed for 132h to high PGC inducing signals (Fig 5 C-D; see image below).

Moreover, in a subsequent study (Hayashi et al., 2011), the same group switched to a more robust *Blimp1*-*Stella* double reporter system (BV/SC). Using this system they showed that while around 40-50% of epiblast cells induced *Blimp1* after two days of exposure to PGCLC inducing signals, only around 10% of the cells were actually positive for both *Blimp1* and *Stella* reporters after 4-6 days of PGCLC induction (Fig S5c from Hayashi et al., see image below) and can be thus considered as PGCLC. Similar numbers of PGCLC were reported by Hayashi et al. using either *in vitro* derived EpiLC or epiblast cells isolated from embryos. When using different ESC strains to obtain EpiLC the authors observed some variability in the % of PGCLC that can be obtained, which nevertheless were typically lower than 20%.

On the other hand, there are reports indicating that *Blimp1* is not exclusively expressed in PGC *in vivo* and that the *Blimp1* reporter system might not only mark the PGC lineage (e.g. Mikedis et al., 2016; DOI: 10.1002/dvdy.24461). Accordingly, evaluation of scRNA-seq data generated in E8.25 mouse embryos (Ibarra-Soria et al., 2018; DOI: 10.1038/s41556-017-0013-z) shows that *Prdm1* expression can be observed in several embryonic and extraembryonic tissues (see image below).

Hence, although stating that <20% of epiblast cells are germline competent might be an underestimation, we still think that claiming that the formative epiblast is heterogeneous in terms of its intrinsic germline competence should not be considered incorrect. Nevertheless, our study is focused on the epigenetic differences within enhancers between EpiLC and EpiSC rather than on the heterogeneity of the formative epiblast in terms of its intrinsic germline competence. Therefore, the previous statements regarding the heterogeneity of the formative epiblast have been removed in the revised manuscript.

c) Page 4 'primordial germ cell-like cells (PGCLC). This system revealed transcription factors (TFs) and epigenomic reprogramming events involved in PGC specification and led to a better understanding of the mouse peri-implantation transitions in general'. This statement appears to reveal a fundamental misinterpretation of the literature. The critical transcription factors for PGC specification were all discovered *in vivo* (for instance, Ohinata et al., 2005; Weber et al., 2010; Yamaji et al., 2008) prior to the discovery of the PGCLC system. Epigenetic reprogramming again, has been best studied *in vivo*. The early stages of epigenetic reprogramming (in some nomenclatures 'reprogramming 1') have been characterised *in vivo* (Seki et al., 2005; 2007) and confirmed *in vitro*. The main epigenetic reprogramming event ('reprogramming 2') that occurs on colonisation of the gonads is particular difficult to study *in vitro* as there is no

well-defined model for this stage of PGC development – and again in vivo observations have been much more informative here (Hajkova et al., 2008; 2010; P. W. S. Hill et al., 2018).

We agree with the reviewer in that our previous text did not properly describe the literature and that the major PGC regulators as well as main epigenetic reprogramming events were originally described *in vivo*. Our intention, which was not properly explained, was to highlight how the *in vitro* PGCLC system has facilitated the mechanistic and genomic characterization of those TFs and reprogramming events. Therefore, in the revised version we have stated the following (Page 3): “*This system facilitated the mechanistic and genomic characterization of transcription factors (TFs) (Hackett et al. 2018; Murakami et al. 2016; Mitani et al. 2017) and epigenetic reprogramming events (Shirane et al. 2016; Kurimoto et al. 2015; von Meyenn et al. 2016) previously shown to be involved in PGC specification in vivo (Saitou et al. 2012; Sybirna et al. 2019)*”.

d) Naïve ESCs grown in 2i are not largely devoid of CpG methylation. Depending on the exact culture conditions there is an approximately 3-4 fold reduction compared to the levels observed in serum grown ESCs (Ficz et al., 2013; Habibi et al., 2013; Leitch et al., 2013).

We agree with the reviewer in that “*devoid of CpG methylation*” is an overstatement that does not properly describe CpG methylation levels in 2i ESC. Therefore, this sentence has been eliminated in the revised manuscript and we now only use the term “*devoid*” in the Discussion section to describe mCpG levels in *Drosophila* compared to mammalian cells (page 17): “*in comparison to mammalian cells, the D. melanogaster genome is largely devoid (< 0.03 %) of CpG methylation throughout its entire life cycle (Deshmukh et al. 2018)*”.

e) Page 6 ‘Germline specification is impaired’. This an overclaim and one of many examples in which the authors switch between the in vivo process of PGC specification (occurring from the competent epiblast), and the in vitro observations made during PGCLC induction from EpiLCs. There is no clear evidence that PGCLC induction follows the same trajectory as PGC specification. It certainly does result in PGCLCs that have an identity similar to E9.5 PGCLCs, some of which are functional to make gametes. While the PGCLC system is very useful for studying many aspects of PGC biology, it is not clear that observations in this system are always an accurate reflection of the in vivo mechanisms, in fact there are clear examples in which this is not the case (R. J. Hill & Crossan, 2019; Senft, Bikoff, Robertson, & Costello, 2019).

We agree with the reviewer in that, since our work is based on the use of the PGCLC *in vitro* system, we should be more careful in implying or suggesting that

our findings can be relevant during *in vivo* PGC specification. Therefore, in the revised manuscript we have tried to clearly distinguish between *in vitro* and *in vivo* observations. For example, the sentence highlighted by the reviewer has now been changed to the following (page 5): ***“Most importantly, the persistence of H3K4me1 within PGCLC enhancers seems to contribute to in vitro germline competence, as in the absence of this histone mark, PGCLC differentiation efficiency is reduced.”***

f) Page 7: ‘Previous work indicated that the acquisition of germline competence by the formative epiblast (i.e. E5.5-6.5 epiblast; EpiLC)...another possibility that has not been thoroughly investigated is that germline competence is associated with a few cells of the formative epiblast in which the naïve expression program is totally or partially retained’. In fact, the reference cited makes quite clear that the naïve network is dismantled at the single cell level. In addition, there is a huge amount of evidence that argues against a small subset of naïve cells maintaining expression of the naïve network – including observations *in vivo* (staining of these markers at relevant time points, the various reporter mice used in the field, scRNA-seq papers) and *in vitro* (the PGCLC system). As such the authors create a false dichotomy. In the end their analysis supports the prevailing view and adds no novelty. I would remove from any subsequent manuscript.

The reference that the reviewer mentions (Mulas et al., 2017) uses a *Rex1-GFP* reporter system as a readout for the naïve expression program. Using this reporter, the authors showed that *Rex1* negative cells have dismantled the naïve program upon ESC differentiation as compared to *Rex1* positive cells. However, in the study by Mulas et al. the global dismantling of the naïve expression program was investigated by single-cell RT-qPCR analysis of selected genes and by bulk RNA-seq comparing *Rex1-low* and *Rex1-high* cells 24 hours after withdrawal from 2i conditions (Kalkan et al. 2017). We think that our work provides additional support to this previous study by performing a global assessment of the naïve gene expression program at the single-cell level. Moreover, in the particular case of the mouse PGCLC system, we are not aware of any previous characterization using single-cell RNA-seq across the different stages that this *in vitro* system entails (i.e. 2i ESC, Day 1 EpiLC, Day 2 EpiLC, D2 EBs, D4 EBs). Therefore, although we agree with the reviewer in that our data confirms the already prevailing model, we think that it is important to show that such model is also supported by data generated with novel, global and more sensitive technologies (i.e. single-cell RNA-seq). We have now incorporated new text acknowledging the previous and prevailing view in the field (page 6): ***“Previous work indicates that the acquisition of germline competence in day 2 (d2) EpiLC entails the complete dismantling of the naïve gene expression program (Mulas et al. 2017; Kalkan et al. 2017)”***.

On the other hand, our single-cell RNA-seq characterization of the PGCLC system has also provided some other interesting insights. For example, as stated in page 5, it revealed that in Day 4 EBs there were cell clusters “similar to the extraembryonic tissues (*i.e.* extraembryonic ectoderm, extraembryonic mesoderm and endothelium) that surround PGCs in the proximo-posterior end of the mouse embryo following germline specification *in vivo*”. We have further confirmed this observation, which the reviewer also found surprising, by performing additional analyses that are presented in the revised Fig. S1A. In order to facilitate the evaluation of these and other observations, in the revised manuscript we have added the following note (page 5): “(the scRNA-seq data can be easily explored with the cloupe file available through GEO: GSE155088).”

Overall, we think that the scRNA-seq presented in our work and generated at various time points during PGCLC differentiation should represent a valuable resource for the scientific community, especially if, as we intend, it can be easily accessed, visualized and analyzed.

2. Definition of priming. This leads to much confusion throughout the paper. PGCLC enhancers are marked with H3K4me1 and have low CpG methylation in naïve ES cells. H3K4me1 is lost and CpG increased as cells differentiate towards EpiSC (via an EpiLC state). It is not clear whether PGCLC enhancers are actively maintained in a ‘primed’ state or whether this simply represents a differentiation process, which is interrupted by PGCLC induction. The language the authors employ on this topic does not help, and seems to be focussed on employing fashionable jargon rather than explaining their own observations. Statements like - ‘they transiently acquire a primed state in EpiLC’ (line 12) – do seem to suggest an active process, although the authors never really show this or give a clear indication of what they mean by ‘priming’ or ‘acquisition of a primed state’. The statement ‘Therefore, the transient priming of the PGCLC enhancers in formative epiblast cells could endow them with permissive chromatin features’ on Page 18 is another example, and combines uncertainty regarding the mechanisms of priming with inaccurate description of the data – the authors have not undertaken studies in epiblast cells. Finally, it is not clear to what extent the authors think this is a PGCLC specific phenomenon?.

Following the suggestions by the reviewers, we have re-defined the PGCLC enhancers and the way we assign them to genes using a more streamlined and robust approach (see pages 7 and 30 in the Results and Methods sections, respectively). Briefly, PGCLC enhancers were defined as those distal d6 PGCLC H3K27ac peaks that could be physically linked to the PGCLC gene set. Moreover, both in the previous as well as in the current analyses of the PGCLC enhancers, it seems clear that the partial retention of H3K4me1 in EpiLC in comparison to EpiSC occurs preferentially at a subset rather than at all the PGCLC enhancers. We

have now more quantitatively defined the subset of PGCLC enhancers in which H3K4me1 signals are higher in EpiLC than in EpiSC using three H3K4me1 ChIP-seq biological replicates for each cell type (Fig 3c+d, Supplementary Fig. 3b). For those same PGCLC enhancers showing higher H3K4me1 in EpiLC than in EpiSC, which we now refer to as Group I enhancers, we also observed lower levels of heterochromatin features (i.e. mCpG, H3K9me3) in EpiLC than in EpiSC (Fig 3e). However, chromatin accessibility, as measured by ATAC-seq, seems to be already very low in EpiLC and similar to the levels measured in EpiSC (Fig S3d). The major loss of ATAC-seq signals already in EpiLC argues against an active priming mechanism in which “pioneer” TF and MLL3/4 remain bound to the PGCLC enhancers but rather supports, as mentioned by the reviewer, more passive mechanisms in which H3K4me1 is progressively lost at PGCLC enhancers from ESC to EpiSC, but with slower dynamics in comparison to the silencing of PGCLC genes. Consequently, we have made major changes in the manuscript text to more accurately describe our observations, eliminating confusing statements like the ones mentioned by the reviewer. Some examples below:

Title: *Enhancer-associated H3K4 methylation safeguards in vitro germline competence*

Page 5, Introduction:

„During the transition from 2i ESC to EpiLC, FOXD3-bound enhancers lose TF and co-activator binding as well as H3K27ac but partly retain H3K4me1. This suggests that these enhancers do not become fully decommissioned, but transiently display a chromatin state similar, but not identical, to that of primed enhancers (Calo and Wysocka 2013; Creighton et al. 2010). Enhancer priming typically involves binding of pioneer TFs and pre-marking by H3K4me1 that can precede and facilitate subsequent enhancer activation (i.e. marking by H3K27ac, recruitment of RNA Pol II, production of eRNAs) (Lara-Astiaso et al. 2014; Lee et al. 2019; Wang et al. 2015; Lai et al. 2017). Interestingly, in differentiated macrophages, enhancers activated upon stimulation rapidly lose H3K27ac and TF binding, while retaining H3K4me1 for considerably longer. It was proposed that H3K4me1 persistence could facilitate a faster and stronger enhancer induction upon restimulation (Ostuni et al. 2013). It is currently unknown whether, during development, H3K4me1 persistence once enhancers become decommissioned can similarly facilitate their eventual re-activation (Calo and Wysocka 2013).”

Page 16-17, Discussion:

“In the case of in vitro germline competence, here we report that a subset of PGCLC enhancers gets partly decommissioned in EpiLC and retains permissive chromatin features, including H3K4me1, already present in a preceding active state (i.e. in 2i ESC) (Fig 7g). This resembles the so called latent enhancers previously described in

differentiated macrophages, in which, following an initial round of activation and silencing, the persistence of H3K4me1 was proposed to facilitate subsequent enhancer induction upon restimulation (Ostuni et al. 2013). The mechanisms involved in the persistence of H3K4me1 and other permissive chromatin features are still unknown, although we can envision at least two non-mutually exclusive possibilities: (i) a passive mechanism whereby MLL3/4 binding to PGCLC enhancers is already lost in EpiLC, but H3K4me1 can still be transiently retained due to the slow dynamics of H3K4 demethylation (AlAbdi et al. 2020; Maltby et al. 2012); (ii) an active maintenance mechanism similar to the one reported for enhancer priming (Lee et al. 2019; Wang et al. 2015), whereby the binding of certain TFs might enable the persistent recruitment of MLL3/4 and the retention of H3K4me1 within PGCLC enhancers. Since PGCLC enhancers display low and similar ATAC-seq signals in EpiLC and EpiSC (Fig. 3), this would argue in favor of passive mechanisms rather than an active retention of TFs and co-activators (e.g. MLL3/4) in EpiLC.”

Finally, our data suggests that the partial decommissioning of PGCLC enhancers, including the persistence of H3K4me1, can facilitate their future re-activation during PGCLC induction. We believe that this is conceptually different from previously proposed “priming” mechanisms, but could resemble what has been previously described for the so called „latent“ enhancers in differentiated macrophages, in which, following an initial round of activation and silencing, the persistence of H3K4me1 was proposed to facilitate subsequent enhancer induction upon restimulation (Ostuni et al. 2013). We agree with the reviewer that it would be important to determine whether similar mechanisms can operate in other developmental contexts. However, we feel that this is out of the scope and time-frame of our study and, thus, have added the following sentence in the Discussion (Page 18):

“Future work will elucidate the prevalence and regulatory mechanisms by which the epigenetic state of enhancers can contribute to cellular competence and the robust deployment of developmental gene expression programs. In this regard, it would be important to evaluate whether, as reported here for PGCLC induction, the partial decommissioning of enhancers can be involved in their subsequent reactivation and, thus, in the induction of gene expression programs in other developmental contexts. Similar mechanisms might be also important in other physiological (Ostuni et al. 2013) and pathological (Kaufman et al. 2016; Pomerantz et al. 2020) contexts in which a previously used but already dismantled gene expression program gets re-activated.”

3. ChiP-seq data

This is apparently generated in EBs rather than PGCLCs. Given that the majority of these EBs (based on the authors own data) are not PGCLCs this calls into question all of the interpretations based on these datasets.

We assume that the referee refers to the H3K27ac ChIP-seq generated in WT and dCD day 4 EBs, as all other presented ChIP-seq datasets were generated in ESC, EpiLC and EpiSC.

As the reviewer points out, only a relatively small fraction of cells within the EBs are PGCLC, which actually makes it technically challenging to obtain sufficient PGCLC to generate high quality ChIP-seq profiles. We have estimated that through FACS sorting with the two surface markers we typically use, we could obtain around 10^4 WT PGCLC and 3×10^3 dCD PGCLC for each 96-well plate. Since even with highly sensitive methods, such as CUT&RUN or ChIPmentation, it is necessary to use around 10^5 cells to obtain high quality ChIP-seq profiles, this would imply pooling PGCLC from 10 plates for WT cells and 30 plates for dCD, which we feel is technically and economically very challenging. On the other hand, despite the low numbers of PGCLC present within EBs, we can still observe clear H3K27ac signals in the WT EBs that are strongly reduced in the dCD EBs (Fig 6b). For many PGCLC enhancers this reduction in H3K27ac is stronger than the ~2-fold reduction in PGCLC numbers observed between WT and dCD EBs (Fig 6a). This suggests that the loss of H3K27ac is not simply explained by the loss of PGCLC in dCD EBs.

Nevertheless, we agree with the reviewer that a more direct comparison between WT and dCD PGCLC could be insightful. Therefore, we have now generated scRNA-seq data in WT and dCD EBs to compare the different cellular identities present within the EBs and to directly compare the expression profiles of WT and dCD PGCLC. Clustering analysis of this new scRNA-seq revealed that the subcluster corresponding to PGCLC contained considerably more WT than dCD cells (Fig. 6e, Supplementary Fig. 6c+d), thus in agreement with our FACS-based quantifications. In addition, we also noticed that within the dCD EBs there were cells expressing major PGC markers (i.e. *Prdm1* or *Dppa3*) but not naïve pluripotency ones (i.e. *Klf4*) (Fig 6d-e). The proportion of these *Prdm1**Dppa3*⁺/*Klf4*⁻ cells was actually quite similar among WT and dCD EBs (Fig 6e). However, while in the WT EBs, these cells were mostly found within the PGCLC subcluster, in the dCD EBs they were part of a larger subcluster with poorly defined identity (Supplementary Fig. 6c). Notably, the expression of PGCLC genes, especially those associated with PGCLC enhancers (e.g. *Tfap2c*, *Prdm14*, *Utf1*, *Esrrb*), was reduced in dCD *Prdm1**Dppa3*⁺/*Klf4*⁻ cells in comparison to their WT counterparts (Fig 6f). Overall, these scRNA-seq analyses suggest that dCD cells are capable of differentiating into PGCLC, which, nevertheless, display an abnormal induction of the PGCLC expression program, particularly of those genes linked to PGCLC enhancers.

Together with previous results obtained by bulk RNA-seq analyses of ESC/EpiLC/EpiSC, we conclude (Page 14): “*Altogether, our data shows that*

H3K4me1 is required for in vitro germline competence and proper PGCLC induction. Although we cannot rule out that gene expression and epigenetic changes in ESC and/or extraembryonic-like cell types (Xie et al. 2020) might also contribute to the PGCLC differentiation defects observed in dCD/dCT cells, our data suggests that the persistence of H3K4me1 within the PGCLC enhancers might facilitate their reactivation during PGCLC induction.

4. 'faithfully recapitulates PGC specification' (Line 7)

As mentioned above (1d) this has not in fact been demonstrated. What has been shown is that PGCLCs can be induced which bear the hallmark of E9.5 PGCs. Whether the specification process is the same as PGC specification is not known. This has a major implications for the design of the study. It really is not clear what 'active PGCLC enhancers' designated using an 'H3K27ac ChIP-seq data generated in d2 and d6-sorted PGCLC' (Page 10) represent with regards in vivo development. Based on current evidence these might represent enhancers active in E9.5 PGCs. If this is the case, then these could readily be identified in actual E9.5 PGCs – of which thousands are readily obtained from a single litter of embryos. Whether these would be relevant to PGC specification is of course a separate question.

We agree with the reviewer in that our previous statement about the PGCLC system was too strong and we have changed it accordingly (page 3): “These limitations were mitigated by a robust in vitro differentiation system whereby mouse embryonic stem cells (ESC) grown under 2i conditions (naïve pluripotency) can be sequentially differentiated into EpiLC and PGCLC that resemble the formative epiblast and E9.5 PGC, respectively (Hayashi et al. 2011).”

In the revised manuscript, we have only considered H3K27ac ChIP-seq data generated from d6-sorted PGCLC in order to call PGCLC enhancers, as those cells are considered to resemble E9.5 PGCs (Hayashi et al., 2011, and, thus have a more established in vivo counterpart than d2-sorted PGCLC. In addition, we have also analyzed DNase I data generated in PGCs isolated from E9.5 and E10.5 mouse embryos and found that our PGCLC enhancers show high chromatin accessibility in PGCs in vivo (Fig 7A), thus supporting their relevance.

Regarding the possibility of obtaining high-quality H3K27ac ChIP-seq data from E9.5 PGCs, based on the literature it seems that it could be possible to isolate around 1000 cells from each embryo. So, as stated above, in order to isolate 10⁵ PGCs and thus generate high-quality ChIP-seq profiles, it would be necessary to use around 100 embryos, which we find technically quite challenging. Moreover, such profiles would need to be compared to profiles generated in vivo for the formative and primed epiblast not only for H3K27ac but also for the additional histone modifications investigated in our study. Considering this and

acknowledging that we can not assume or imply that our *in vitro* findings are relevant *in vivo* (page 16; last Results section paragraph), we still think that the *in vitro* PGCLC system offers a powerful approach to generate genomic profiles and address mechanistic questions that are relevant not only from a germline point of view but also for enhancer biology in general. In this regard, we think that some of the major findings in our work (e.g. (i) the description of how the partial decommissioning of enhancers can be relevant for subsequent enhancer re-activation and (ii) how the persistence of H3K4me1 can actually facilitate the (re)activation of such enhancers) can be considered of general interest in the enhancer field.

5. PGCLC enhancers

a) The decision to only focus on those enhancers near the pre-selected PGC genes also seems to restrict the analysis – especially as in the current version of the manuscript it is not immediately clear how these PGCLC genes are defined (i.e. Line 8: ‘PGC markers (e.g. Nanog , Tfap2c , Prdm14, Prdm1, Dppa3) (Fig. 1d)’). As these ‘PGC markers’ are discussed quite extensively, it would be helpful if in the text the authors indicate how many genes are in this gene-set and how they are chosen.

In the previous manuscript version, the full list of PGCLC genes was provided in Supplementary Data 2 (151 genes in total) and a detailed explanation of how that gene set was defined was provided in the Methods section: *“The EpiLC and PGCLC gene sets were defined by differential expression using Seurat and the “negbinom” option for differential expression testing.....PGCLC genes were determined by differential expression between the d2+d4 PGCLC clusters and the remaining clusters from d2 and d4 EB as well as d2 EpiLC. Again, from this analysis only the genes upregulated in PGCLC (adjusted p-value < 0.001) and with a low expression distribution in the other analyzed cells (expressed in less than 40% of d2 EpiLC and non-PGCLC EB cells) were considered. In the case of PGCLC genes this resulted in 151 PGCLC genes (Supplementary Data 2).”*. Nevertheless, to be more inclusive, in the revised version we have slightly changed the definition of the PGCLC genes by considering genes upregulated in d2+d4 PGCLC clusters with respect to the remaining clusters in d2 and d4 EBs but without including in this analysis the d2 EpiLC (see Methods for more details). This has resulted in 389 PGCLC genes that are now presented in the revised Supplementary Data 2. Moreover, as we agree with the reviewer in that this information should be more accessible to the reader, we have now added the following text to the results section (page 5): *“Furthermore, differential expression analysis between the PGCLC cluster and the remaining cells of the d2 and d4 EBs (see Methods) led to the identification of a set of 389 PGCLC genes (Supplementary Data 2), which included the PGC markers mentioned above as well as major naïve pluripotency regulators (e.g. Nanog, Esrrb) (Fig. 1d-e).”*.

On the other hand, we also acknowledge that in the previous manuscript version the epigenomic comparisons were largely focused on the PGCLC enhancers. Therefore, it was not clear whether the observed epigenomic differences were specific to these enhancers or simply reflect genome-wide changes. To avoid this, in the current manuscript version we have defined two additional enhancer sets (EpiLC (n=312) and EpiSC (n=223) enhancers; Supplemental Data 3; Fig 2a) using similar criteria to the ones used to define PGCLC enhancers (see Methods for details). Moreover, through the revised manuscript, epigenomic comparisons are presented for these three enhancer sets (i.e. PGCLC, EpiLC and EpiSC enhancers) as well as for the TSS of the PGCLC genes.

b) The large number of enhancers could be due to complex/redundant enhancer usage but also could be due to overcalling of enhancers – is this possible to interrogate in the data?

In the previous manuscript version, PGCLC and EpiLC enhancers were called using rather similar criteria yet the number of enhancers/gene was higher for the PGCLC genes. This argues against the complex/redundant enhancer usage observed for PGCLC genes being caused by overcalling of enhancers.

Nevertheless, we agree with the reviewer in that the observed number of enhancers/gene seems to be too high for many EpiLC and PGCLC genes (former Fig S2A). One potential reason for this could be that we previously used ATAC-seq peaks together with H3K27ac data to call enhancers, which are very abundant and are often spatially clustered and embedded within single H3K27ac peaks. Rather than individual enhancers, these nearby ATAC-seq peaks could represent distinct TF binding sites found within the same enhancers. This possibility has been minimized in the revised manuscript, since following the reviewers suggestions, we have now changed the strategy to identify enhancers in PGCLC using a more streamlined and robust approach. Briefly, we now used H3K27ac data from d6-sorted PGCLC and then assigned the identified H3K27ac peaks to PGCLC genes based on physical contacts (using Capture-C data; see Methods for further details). Using this new strategy, we have identified 415 PGCLC enhancers associated with 216 PGCLC genes (1.9 enhancers/gene). The PGCLC enhancers are all listed in Supplemental Data2.

c) There are a number of genes that retain H3K27ac in both EpiLCs and EpiSCs – which genes are these?

We assume that the reviewer means PGCLC enhancers rather than genes (former Fig 2b). It is true that in both the previous as well as in the revised PGCLC enhancer list (new Fig2a and Fig S3d), there is a set of regions that shows high

levels of H3K27ac in ESC, EpiLC and EpiSC. These PGCLC enhancers are linked to genes that are expressed in the three in vitro pluripotent stages (i.e. ESC, EpiLC, EpiSC), but specifically in PGCLC within the d4 EBs (cloupe browser, Supplemental data 1+2). We indicate below some examples of these PGCLC enhancers (genomic coordinates) and their putative target genes:

- chr10 95261730 95263512 Socs2
- chr9 58264938 58281605 Pml
- chr6 122711567 122717799 Nanog
- chr1 136565235 136571848 Platr22

d) Page 11 'All these active chromatin features decreased upon exit from naïve pluripotency, but, H3K4me1 was partly retained in EpiLC in comparison to EpiSC (Fig. 2d, Supplementary Fig. 2c)'. As this is major claim in the paper it seems important to properly defined what 'partially retained' means in the main text. The difference between H3K4me1 profiles in Figure 2D in EpiSC vs EpiLC appears very minor. Again, the most notable pattern in Figure 2f is the similarity in the CpG methylation between EpiLC and EpiSC (when compared with ESC) and this does not fit the description in the text which states 'intermediate' and 'hypermethylated'. Similarly, the description of Figure 3a as showing 'intermediate' levels in E5.5 does actually seem to accurately reflect the data. As such the conclusion on Pages 12-13, seems far too strong and therefore amounts to an overclaim based on the data presented.

Both reviewer 1 and 2 have similar concerns about the mild differences in H3K4me1 between EpiLC and EpiSC. We think that part of the problem is that we previously referred to all PGCLC enhancers to describe the retention of H3K4me1 in EpiLC, while this was only obvious for a subset of enhancers, particularly those that are initially highly active in ESC (see new Fig 3d and Fig S3b). In the revised text, we have clearly stated that the partial decommissioning of enhancers in EpiLC compared to EpiSC is observed for a subset of PGCLC enhancers. In addition, to more quantitatively defined the "partial retention" of H3K4me1 in EpiLC compared to EpiSC, we have now used three H3K4me1 ChIP-seq replicates in each cell type to classify PGCLC enhancers in two major groups (Fig 3c): (i) Group I: PGCLC enhancers showing a H3K4me1 EpiLC/EpiSC ratio higher than 1.2-fold in at least two of the three ChIP-seq replicates; (ii) Group II: all other PGCLC enhancers showing either similar or higher H3K4me1 signals in EpiSC compared to EpiLC. Using these criteria, 71% of the PGCLC enhancers were assigned to Group I and thus show higher H3K4me1 levels in EpiLC than in EpiSC (Fig 3d, Fig S3b-c).

Using this new classification of PGCLC enhancers into Group I and II, it becomes more obvious that those PGCLC enhancers showing higher H3K4me1 levels in EpiLC than in EpiSC (i.e. Group I) are also more protected from CpG methylation

(Fig 3e-f). Importantly, this seems to be also true when evaluating mCpG levels within PGCLC enhancers in the E5.5 and E6.5 epiblast (Fig 7b). However, we agree with the reviewer in that the mCpG levels within Group I PGCLC enhancers are still considerably higher in EpiLC than in ESC and that the terms “Intermediate” and “hypermethylated” are not appropriate. Therefore, these terms have been eliminated in the revised text:

- Abstract: *“Namely, a subset of these enhancers partly retain H3K4me1, accumulate less heterochromatic marks and remain accessible and responsive to transcriptional activators.”*

- Page 15: *“In agreement with our in vitro observations, PGCLC enhancers showing incomplete decommissioning in EpiLC (i.e. Group I enhancers) displayed lower CpG methylation levels in germline competent E5.5 epiblast cells than in the E6.5 epiblast (Fig. 7b), in which germline competence is already reduced (Ohinata et al., 2009).”*

e) This is another example of the uncertainty created by the terminology of ‘priming of PGCLC enhancers’ – this implies an active and targeted process that is particular to this state. In fact, the authors show a transition from ESC to EpiSC, with the gradual loss of H3K4me1 and gain of CpG methylation (both of which appear largely complete by the EpiLC stage – rather than a distinctive intermediate state).

As stated in a previous response (Comment #2), chromatin accessibility, as measured by ATAC-seq, seems to be already low in EpiLC and similar to the levels measured in EpiSC for most PGCLC enhancers (new Fig S3d). The major loss of ATAC-seq signals already in EpiLC argues against an active priming mechanism in which “pioneer” TF and MLL3/4 remain bound to the PGCLC enhancers but rather supports, as mentioned by the reviewer, more passive mechanisms in which H3K4me1 is progressively lost at PGCLC enhancers from ESC to EpiSC, but with slower dynamics in comparison to the silencing of PGCLC genes. As a result, there are a group of PGCLC enhancers (i.e. Group I) that show higher H3K4me1 levels and lower levels of heterochromatin features in EpiLC than in EpiSC. The potential mechanisms leading to the partial decommissioning of a subset of PGCLC enhancers in EpiLC are briefly discussed in pages 16-17:

“The mechanisms involved in the persistence of H3K4me1 and other permissive chromatin features are still unknown, although we can envision at least two non-mutually exclusive possibilities: (i) a passive mechanism whereby MLL3/4 binding to PGCLC enhancers is already lost in EpiLC, but H3K4me1 can still be transiently retained due to the slow dynamics of H3K4 demethylation (AlAbdi et al., 2020; Maltby et al., 2012); (ii) an active maintenance mechanism similar to the one reported for enhancer priming (Lee et al., 2019; Wang et al., 2015), whereby the binding of certain TFs might enable the persistent recruitment of MLL3/4 and the retention of H3K4me1 within PGCLC enhancers. Since PGCLC enhancers display low and similar ATAC-seq signals in EpiLC and EpiSC (Fig. 3), this would argue in favor

of passive mechanisms rather than an active retention of TFs and co-activators (e.g. MLL3/4) in EpiLC.”

On the other hand, although the epigenetic differences between EpiLC and EpiSC within PGCLC enhancers might not be dramatic, they can still be meaningful and biologically relevant. In agreement with this, the Group I PGCLC enhancers are considerably more accessible and responsive to transcriptional activators in EpiLC than in EpiSC (Fig 4f-g).

6. Epigenetic heterogeneity:

a) Figure 3e – the claim that ‘CpG methylation heterogeneity of the E5.5 epiblast was more pronounced for PGCLC enhancers than for other enhancers or the whole genome’ is not clearly supported by the data. The degree of epigenetic heterogeneity seems very similar at EpiLC enhancers, indicating this is not a specific feature of PGCLC enhancers. Similarly the data presented in Figure 3i is difficult to interpret without comparing with non-PGCLC enhancers.

The reviewer is right and the epigenetic heterogeneity for PGCLC enhancers and EpiLC enhancers was quite similar and higher than for the rest of the genome. Moreover, these observations are in agreement with previous results from Wolf Reik lab (Rulands et al., 2018) showing that the formative epiblast is epigenetically heterogeneous, particularly within enhancers. Nevertheless, since in the revised manuscript we have modified the criteria to call the PGCLC enhancers as well as the EpiLC and EpiSC enhancers, we have measured again epigenetic heterogeneity for these three enhancers sets. As can be seen in the new Fig 7e, the Group I PGCLC enhancers appear to be more heterogeneous than the other enhancer groups, although the differences are rather moderate. We have made changes to the text to better describe our data (page 15):

“When comparing different enhancer sets across epiblast stages, the highest epigenetic heterogeneity (~30 %) was observed for the Group I PGCLC enhancers in the E5.5 epiblast (Fig. 7e).”

The purpose of the ChIP-bisulfite sequencing experiments shown in the former Fig 3i was to investigate whether, as suggested by *in vitro* biochemical assays (Ooi et al., 2007; Zhang et al., 2010; Guo et al., 2015), H3K4me1/2 bound chromatin could be protected from DNA methylation and thus, display lower mCpG levels than bulk chromatin. This property should not be specific to PGCLC enhancers but rather a general feature of H3K4me1/2 bound chromatin. Moreover, as discussed in the response to the next point raised by the reviewer, in the revised manuscript we have decided to show the ChIP-bisulfite data (former Fig 3i) now in Fig. 3g, as they support that the presence of H3K4me1 within these enhancers is correlated with lower DNA methylation levels.

b) Page 15 – again the conclusions here are far too strong based on the data presented. For instance, not all the data presented is from ‘formative epiblast’. What is meant by H3K4me1 variability in this example? The authors cannot comment on heterogeneity between epiblast cells of the same embryo (or any cells for chromatin marks). Does higher H3K4me1 correlate with lower DNA methylation?.

As mentioned in previous responses and following the reviewer’s advice, in the revised manuscript we have tried to avoid referring to *in vivo* stages (e.g. formative epiblast) whenever presenting data from *in vitro* cell types (i.e. EpiLC). Therefore, in the section dedicated to epigenetic heterogeneity (new Fig 7), we are now only showing results for mCpG data generated *in vivo* and removed the ChIP-bisulfite sequencing experiments performed in EpiLC.

On the other hand, as the reviewer points out, the ChIP-bisulfite sequencing experiments (former Fig 3i, new Fig 3g) do not address whether there is variability for H3K4me1 among EpiLC, but rather evaluate whether, as suggested by previous reports (Ooi et al., 2007; Zhang et al., 2010; Guo et al., 2015), the presence of H3K4me1/2 within PGCLC enhancers could provide protection from DNA methylation. Therefore, the ChIP-bisulfite sequencing experiments are now presented in the new Fig 3g. Together with new panels presented in Fig 3d-f, the ChIP-bisulfite data suggest that H3K4me1 and mCpG levels are anti-correlated within PGCLC enhancers.

7. Enhancer deletion experiments.

The data for the enhancer deletions is difficult to interpret as:

a) only Stella-GFP is used to designate PGCLCs and this is not specific. Rather the authors could be measuring effects on Stella regulation quite independent of PGCLC derivation efficiency.

We agree that the identification of PGCLC based on single reporters might not be as robust as when using a double reporter system (Hayashi et al., 2011). Therefore, the Stella-GFP signals might not be specific enough to identify PGCLC. Considering that the main goal of the enhancer deletion experiments was to validate our PGCLC enhancer calling strategy and evaluate whether those enhancers were important for the expression of their predicted target genes (new Fig 2 and new Supplementary Fig 2), we have decided to eliminate the PGCLC quantifications based on the DPPA3-GFP reporter.

b) the reduction in expression of *Esrrb* does not impact PGCLC formation, and this is the one gene with a known role in PGCs (Mitsunaga et al., 2004).

As mentioned in the previous response, we have removed the PGCLC quantifications based on the DPPA3-GFP system, including those for the cell lines with the *Esrrb* enhancer deletion, due to the potential limitations of using a single reporter. However, we would like to point out that according to Mitsunaga et al.: *“The ERR- β null embryo of 9.5 dpc showed normal morphology and migrating PGCs were observed in the mesentery as in those of the normal littermates. It was expected because ERR- β expression was not observed in the PGC of E9.5 embryos. Morphology of the gonads of the *Estrrb*-/- embryos at later stages was also very similar to that of the wild-type gonads. However, number of the PGCs appeared to be lower in the null mutants than in the wild-type littermate (Fig. 4). At E13.5 and E15.5, less number of GFP-positive PGCs was seen in the testis cord of the *Estrrb*-/- male gonad (Fig. 4 A,B; left). As shown in Fig. 4C, in this particular -/- female embryo of E13.5, much fewer GFP-positive PGCs were observed only in the middle part of the gonad. Reductions of the PGCs in the *Estrrb*-/- mutants of both sexes were also observed at E15.5 (Fig. 4B,D). Immunostaining with anti-OCT3/4 antibody clearly revealed that number of the GFP-positive PGCs were two-to-five fold lower in the mutants in comparison with the wild-type littermates”*. Therefore, according to Mitsunaga et al. , ESRRB does not seem to be required for early PGC specification and PGC defects are not observed at E9.5 but at later stages once PGC have reached the gonads. Since the PGCLC system, as used in our study, recapitulates the initial phase of PGC development , we think that our *in vitro* findings were not in major disagreement with the *in vivo* observations.

c) The generalisations around regulation of enhancers ‘PGCLC enhancers frequently control the expression of their target genes already in ESC, further supporting that a significant set of enhancers is functionally shared between naïve pluripotency and PGCLC’ (Page 17) again does not seem to reflect the data presented. Enhancer regulation for both PRDM14 and *Esrrb* seems quite different in PGCLCs versus ES cells and it is not clear how the authors can state ‘frequently’ or ‘significant’ here?.

We agree with the reviewer that generalisations about the regulatory role of PGCLC enhancers should be avoided, as we deleted a limited number of enhancers to evaluate their functional relevance. Therefore, the sentence mentioned by the reviewer has been removed from the revised manuscript. Instead, the results of the enhancer deletions are summarized as follows (page 8): *“Altogether, the previous deletions support the relevance of the identified PGCLC enhancers and suggest that some of them (e.g. *Esrrb* and *Prdm14* E1 enhancers) are particularly relevant during PGCLC induction, while others might be important in both ESC and PGCLC.”*

8. MLL3/4 catalytically deficient ES cell experiments.

The most interesting aspect of this study is the extension of their previous work, using the various cell lines generated. However, what is not clear is if the effects observed are

specific to EpiLC and PGCLC generation, or if this is a general way in which enhancers are regulated during post-implantation development. In vivo data is sadly lacking from this manuscript, and would significantly improve it, as the regulation of the 'PGCLC state' is only really interesting in so far as it might correspond to elements of actual PGC biology. The approach in (Zhang et al., 2018) might well be feasible, in which mutant ES cells are used to make chimaeras, and PGC specification is compared between mutant and endogenous wildtype cells (although properly 'timing' any defect might still be challenging).

In collaboration with Miguel Manzanares laboratory, we were planning to perform chimaera experiments similar to the ones mentioned by the reviewer (Zhang et al., 2018). However, due to the working restrictions caused by the COVID pandemic, such experiments had to be postponed several times and only a few preliminary and inconclusive injections could be performed so far. Moreover, while working in the revision of our manuscript, a recent preprint (Xie et al., BioRxiv, 2020) reported that Mll3/4 catalytic mutant mouse embryos die around E8.5. This strongly suggests that H3K4me1 is essential for early mouse embryogenesis, but also indicates that it would be very challenging to quantify PGCs in these H3K4me1-defective embryos. Therefore, although we fully agree with the reviewer in that the relevance of our work would be strengthened if *in vivo* experiments could be incorporated, we feel that this is currently not feasible within a reasonable time frame. Instead, as already mentioned in a previous response, in the revised manuscript we have tried to state very clearly that our work is based on *in vitro* differentiation systems and that the *in vivo* relevance remains to be evaluated. In addition, we think that our work provides important insights into how enhancer function can be regulated during certain cellular transitions, including (i) how the partial decommissioning of enhancers and (ii) the persistence of H3K4me1 can facilitate enhancer (re)activation. Nevertheless, we acknowledge that it will be important to evaluate whether the persistence of H3K4me1 and the partial decommissioning of enhancers can be important for enhancer activity in additional developmental and cellular contexts. Therefore, we have added the following sentences at the end of the Discussion section (Page 18): *"It would be important to evaluate whether, as reported here for PGCLC induction, the partial decommissioning of enhancers can be involved in their subsequent reactivation and, thus, in the induction of gene expression programs in other developmental contexts. Similar mechanisms might be also important in other physiological (Ostuni et al. 2013) and pathological (Kaufman et al. 2016; Pomerantz et al. 2020) contexts in which a previously used but already dismantled gene expression program gets re-activated."*

b) the connection between H3K4me1 and DNA methylation is not so clear – in particular despite an apparently clear reduction in H3K4me1 in EpiLC (Figure 6C) the

change in methylation in EpiLC looks marginal at best (Figure 6d). Again the description of the data is unclear. On page 20, the authors state 'no longer acquire a primed state in EpiLC' – in fact, there is reduced H3K4me1 in ESCs and this state is then inherited in EpiLC rather than 'acquired'. As such, whether the subsequent reduction in PGCLC generating ability is an issue with PGCLC induction– or rather an epigenetic defect inherited from the ESC state is not clear.

We agree with the reviewer in that the changes in DNA methylation in the dCD EpiLC are rather moderate compared to the major losses of H3K4me1/2 (see new Figure 5b and 5d). However, as stated in previous responses, in the revised manuscript we have classified the PGCLC enhancers as either Group I and Group II, depending on whether they showed significantly higher H3K4me1 levels in EpiLC than in EpiSC. As it can be seen in the new Fig 5d, DNA methylation at some Group I enhancers is lower in WT EpiLC than in dCD EpiLC, while such differences are not observed for Group II enhancers. Moreover, and as stated in previous responses, we have eliminated statements, like the one previously appearing on page 20, regarding the acquisition of a primed state in EpiLC. Accordingly, on page xxx of the revised manuscript we state the following: “Overall, our analyses indicate that the decommissioning of a subset of PGCLC enhancers gets exacerbated in dCD EpiLC compared to their WT counterparts, resulting in a chromatin state similar to the one observed in WT EpiSC (i.e. lower H3K4me1 and higher mCpG; Fig. 3). However, these epigenetic changes do not result in major gene expression changes in any of the investigated in vitro pluripotent cell types.”

c) Figure 6f would benefit from showing all of the histone marks, in all of the different cell types, as the current presentation is difficult to interpret.

We have made changes to the former Fig 6f (Fig 5c and Supplementary Fig. 5g in the revised manuscript) following the reviewer's advice. In the new Fig 5c, we now show all the investigated histone marks in WT and dCD/dCT cells not only for a representative PGCLC enhancer (i.e. *Esrrb* enhancer) but also for representative EpiLC (i.e. *Grhl2* enhancer) and EpiSC (i.e. *Wnt3* enhancer) enhancers. In addition, in the new Fig S5g we show similar histone profiles for another representative enhancer of each enhancer group.

d) The ChIP in d4 EBs is difficult to interpret as there are fewer PGCLCs in these EBs, and so the comparison is not fair. The same is true of the RNA-seq analysis, in which it is not surprising that PGCLC genes are reduced, the question is whether less PGCLCs are made, or whether there are defects in these PGCLCs – and unfortunately this is not answered by the analysis.

As mentioned in a previous response, due to the small fraction of PGCLC present within the EBs, it is technically challenging to obtain sufficient PGCLC to generate

high quality ChIP-seq profiles. We have estimated that through FACS sorting with the two surface markers we typically use, we could obtain around 10^4 WT PGCLC and 3×10^3 dCD PGCLC per 96-well plate. Since even with highly sensitive methods, such as CUT&RUN or ChIPmentation, it is necessary to use around 10^5 cells to obtain high quality ChIP-seq profiles, this would imply pooling PGCLC from 10 plates for WT cells and 30 plates for dCD, which we feel is technically and economically challenging. On the other hand, despite the low numbers of PGCLC present within EBs, we can still observe clear H3K27ac signals in the WT EBs that are strongly reduced in the dCD EBs (new Fig 6b). Moreover, for many PGCLC enhancers, especially those in Group I, the reduction in H3K27ac levels is stronger than the ~2-fold reduction in PGCLC numbers observed between WT and dCD EBs (Fig 6a). This suggests that the loss of H3K27ac is not simply explained by the loss of PGCLC in dCD EBs. However, we acknowledge that performing ChIP-seq experiments in EBs has certain limitations and this is now clearly stated in the revised text (page 13): *“Since the previous ChIP-seq experiments were performed in EBs and not in sorted PGCLC, the low H3K27ac levels in dCD cells could be caused by either a defect in the activation of PGCLC enhancers and their associated genes or by an overall reduction in the number of PGCLC present within d4 EBs. To distinguish between these two possibilities, we performed scRNA-seq analyses of WT (1416 cells) and dCD (1699 cells) d4 EBs (Supplementary Data 5).”*

In the revised manuscript we have generated scRNA-seq data in WT and dCD EBs to compare the different cellular identities present within the EBs and to directly compare the expression profiles of WT and dCD PGCLC. As can be seen in the updated Fig 6 and Supplementary Fig 6, clustering analysis of this new scRNA-seq revealed that the subcluster corresponding to PGCLC contained considerably more WT than dCD cells (Fig 6c-e), thus in agreement with our FACS-based quantifications. In addition, we also noticed that within the dCD EBs there were cells expressing major PGC markers (i.e. *Prdm1* or *Dppa3*) but not naïve pluripotency ones (i.e. *Klf4*) (Fig 6d, Fig 6e). The proportion of these *Prdm1* or *Dppa3*+/ *Klf4*- cells was actually quite similar among WT and dCD EBs (Fig 6e). However, while in the WT EBs, these cells were mostly found within the PGCLC subcluster, in the dCD EBs they were part of a larger subcluster with poorly defined identity (Supplementary Fig 6c). Notably, the expression of PGCLC genes, especially those associated with PGCLC enhancers (e.g. *Tfap2c*, *Prdm14*, *Utf1*, *Esrrb*), was significantly reduced ($p=7.3 \times 10^{-7}$) in dCD *Prdm1* or *Dppa3*+/ *Klf4*- cells in comparison to their WT counterparts (Fig 6f-g). Overall, these scRNA-seq analyses suggest that dCD cells are capable of differentiating into PGCLC, which, nevertheless, display an abnormal induction of the PGCLC expression program, particularly of those genes linked to PGCLC enhancers.

e) With these open questions the conclusion 'our results show that H3K4me1/2 is required for proper PGCLC specification and supports the importance of PGCLC enhancer priming for germline competence' is once more not supported by the data. The authors present no data that can convincingly be tied to PGC specification *in vivo*.

As already stated in previous responses, in the revised manuscript we have tried to avoid implying or suggesting that our findings can be relevant during *in vivo* germline competence or PGC specification. Moreover, we have also tried to clearly distinguish between *in vitro* and *in vivo* observations throughout the manuscript. Therefore, the sentence mentioned by the reviewer as well as similar ones have been eliminated in the revised manuscript. Instead, the new scRNA-seq data generated in WT and dCD EBs together with the results obtained by bulk RNA-seq analyses of WT and dCD ESC/EpiLC/EpiSC, led us to conclude the following (Page 14): "Altogether, our data shows that H3K4me1 is required for *in vitro* germline competence and proper PGCLC induction. Although we cannot rule out that gene expression and epigenetic changes in ESC and/or extraembryonic-like cell types ⁶² might also contribute to the PGCLC differentiation defects observed in dCD/dCT cells, our data suggests that the persistence of H3K4me1 within the PGCLC enhancers might facilitate their reactivation during PGCLC induction."

9. OTX2 data.

This is an interesting connection with previous work from the Chambers lab. However, again the data is not clearly described, which leads to inconsistencies in the story.

a) There seems to be little if any difference in PGCLC induction on day 8, despite apparent retention of H3K4me1 at PGCLC enhancers. This does not seem to fit with authors' model.

We agree with the reviewer in that in the former Fig 7b, the *Otx2*^{-/-} d8 EpiSC show H3K4me1 levels within PGCLC enhancers comparable to those observed in WT EpiLC. However, as described in previous responses, in the revised manuscript we have classified the PGCLC enhancers as Group I and Group II depending on whether or not they show higher H3K4me1 levels in EpiLC than in EpiSC. As can be seen in the new Fig 4c, the H3K4me1 signals for the Group I enhancers progressively decrease in *Otx2*^{-/-} cells from EpiLC to d8 EpiSC. Moreover, the H3K4me1 signals within these Group I enhancers are considerably lower in *Otx2*^{-/-} d8 EpiSC than in WT EpiLC and are actually similar to the signals observed in WT d4 EpiSC, which already have low *in vitro* germline competence. In contrast, the H3K4me1 levels for the Group II enhancers remain quite constant in both WT and *Otx2*^{-/-} cells (Fig 4c).

On the other hand, it is unlikely that the H3K4me1 levels within Group I PGCLC enhancers are the only chromatin feature contributing to *in vitro* germline competence. Instead, the regulation of the epigenetic state of the PGCLC

enhancers (Group I) is likely to be complex and to involve multiple regulatory layers. To acknowledge this complexity, in the revised manuscript we have added the following sentence (page 10): *“Therefore, in addition to H3K4me1, other chromatin features within PGCLC enhancers might also contribute to the extended germline competence of Otx2^{-/-} cells.”*

b) H3K4me1 levels in Day 2 WT EpiLC and Day 4 Otx2 KO looks comparable, and yet there is vastly different PGCLC induction efficiency, indicating that Otx2's action is independent of H3Kme1.

As stated in the previous response, the regulation of the epigenetic state of the Group I PGCLC enhancers is likely to be complex and to involve not only H3K4me1 but also other regulatory layers. In this regard, in the revised manuscript we show that Group I enhancers display slightly higher H3K4me2 signals in d4 Otx2^{-/-} EpiSC than in WT EpiLC (new Fig 4c). Moreover, bisulfite sequencing of two representative Group I enhancers (*i.e.* *Esrrb* and *Prdm1* enhancers) shows that CpG methylation levels within these enhancers is lower in Otx2^{-/-} d4 EpiSC than in WT EpiLC (new Supplementary Fig. 4d). These results are presented in page 10 of the revised manuscript: *“Nevertheless, the correlation between germline competence and H3K4me1 levels within PGCLC enhancers was not perfect, since Otx2^{-/-} d4 EpiSC displayed higher germline competence than WT EpiLC, yet slightly lower H3K4me1 levels within Group I enhancers (Fig. 4c). Therefore, in addition to H3K4me1, other chromatin features within PGCLC enhancers might also contribute to the extended germline competence of Otx2^{-/-} cells. In agreement with this possibility, Group I enhancers showed slightly higher H3K4me2 in Otx2^{-/-} d4 EpiSC than in WT EpiLC (Fig. 4c). Furthermore, genome-wide as well as detailed analysis of representative enhancers (*i.e.* *Esrrb* and *Prdm1* enhancers) showed that the increased competence of Otx2^{-/-} EpiLC and d4 EpiSC was also reflected in reduced CpG methylation levels within Group I PGCLC enhancers (Fig. 4d, Supplementary Fig. 4d). Overall, as the PGCLC genes get properly silenced in Otx2^{-/-} EpiLC (Supplementary Fig. 1g), these results suggest that the extended germline competence of Otx2^{-/-} cells could be linked to the impaired and delayed decommissioning of a subset of PGCLC enhancers.”*

On the other hand, the mechanisms through which OTX2 controls gene expression either in general or more specifically during the establishment of *in vitro* germline competence remain largely unknown. Therefore, we do not claim that OTX2 regulatory function is dependent on H3K4me1, but rather that the extended germline competence of Otx2^{-/-} cells seems to be dependent on H3K4me1/2, since dCD and dCD-Otx2^{-/-} EpiLC showed a strong and similar reduction in their PGCLC differentiation capacity (new Fig 6h).

c) A further issue which is not addressed is whether the increase in H3K4me1 at PGCLC enhancers is already present in *Otx2* null ESCs.

To address this point we have analyzed H3K4me1 and H3K4me2 levels within several Group I PGCLC enhancers by ChIP-qPCR in both ESC and EpiLC (new Supplementary Fig. 4c). For most of the investigated enhancers, H3K4me1 and H3K4me2 levels are actually quite similar in WT and *Otx2*^{-/-} ESC. In contrast, in EpiLC the signals for both histone marks tend to be higher in the *Otx2*^{-/-} cells (Supplementary Fig. 4c).

d) General claims regarding DNA methylation based on analysis of a single enhancer are not appropriate.

The DNA methylation levels in *Otx2*^{-/-} EpiLC have been also investigated globally by whole-genome sequencing experiments. As shown in new Fig 4d, CpG methylation levels within Group I PGCLC enhancers are lower in *Otx2*^{-/-} EpiLC than in WT EpiLC, while such differences are rather minor for Group II enhancers. Moreover, the previous bisulfite sequencing analyses performed for the *Esrrb* enhancer have been extended to another representative Group I enhancer associated with *Prdm1* (new Supplementary Fig. 4d). All together, we think that the presented data in the revised manuscript supports our claims more appropriately (page 10): *“Furthermore, genome-wide as well as detailed analysis of representative enhancers (i.e. Esrrb and Prdm1 enhancers) showed that the increased competence of Otx2^{-/-} EpiLC and d4 EpiSC was also reflected in reduced CpG methylation levels within Group I PGCLC enhancers (Fig. 4d, Supplementary Fig. 4d).”*

e) The reduction in PGCLC induction efficiency in dCD *Otx2* null cells is an interesting result. However, this experiment does seem less clearcut than in Figure 7a – in which PGCLC induction efficiency is much higher and more consistent in *Otx2* null lines. This makes interpretation of the double knockout data more challenging. The relevance of the subsequent DNA methylation data is not very clear.

It is important to mention that in the former Fig 7a (new Fig 4a) and Fig 7d (new Fig 6h), *Otx2*^{-/-} cells were generated using ESC lines derived from different mouse strains: the *Otx2*^{-/-} cells shown in new Fig 4a were previously generated by Acampora et al. (Acampora et al., 2013) using E14 ESC; the *Otx2*^{-/-} cells shown in new Fig 6h were generated by us in this study using R1 ESC. In this regard, it has been reported that different ESC lines show variable germline competence *in vitro* (Hayashi et al., 2011). This has now been explained in the revised manuscript on page 14: *“As expected, the deletion of Otx2 in the R1 ESC resulted in increased germline competence (Fig 6h), although not as pronounced as in E14 ESC (Fig 4a) (Zhang et al., 2018a), which could be attributed to the variable germline*

competence observed among different ESC lines (Hayashi et al, 2011)." We think that observing increased germline competence for *Otx2*^{-/-} cells (albeit not as pronouncedly) when using a different parental ESC line adds further support to the previous results reported by Zhang et al. Moreover, the differences in PGCLC induction efficiency between *Otx2*^{-/-} and dCD-*Otx2*^{-/-} are still quite obvious (Fig 6h) and, thus, we feel that the results obtained with the dCD-*Otx2*^{-/-} cells are still quite relevant.

On the other hand, the DNA methylation analyses, which are now presented in Fig 6i, show that DNA methylation levels within Group I enhancers are higher in dCD-*Otx2*^{-/-} EpiLC than in *Otx2*^{-/-} EpiLC, while such differences are not observed for Group II enhancers. We think that these results further support the protective role of H3K4me1/2 against DNA methylation within an important subset of PGCLC enhancers.

Other points:

1. Page 8 . 'Remarkably, these subclusters were similar to the extraembryonic tissues (i.e. extraembryonic ectoderm, extraembryonic mesoderm and endothelium) that surround PGCs in the proximo-posterior end of the mouse embryo following germline specification in vivo (Fig. 1c, Supplementary Fig. 1a)' – this is not at all clear from the data shown, which appears to rely on cherry-picked genes. Given that this is a surprising finding, could the authors present more clearly how similar non-PGCs are to the cell types indicated?

The annotation of the main cell clusters found within Day 4 EBs was not based on a few selected genes but rather on the combined expression of the major cell identity markers identified by single-cell transcriptional profiling of E8.25 mouse embryos (Ibarra-Soria et al., 2018). This is described in more detail in the Methods section (pages 26-27):

"Briefly, monocle2 (Trapnell et al. 2014) was used to evaluate the in vitro scRNAseq data generated across the different PGCLC differentiation stages. Therefore, k-means clustering was performed on the t-SNE plots (with k=3 for d2 EB and k=4 for d4 EB). From the resulting clusters, those containing PGCLC were identified by the enrichment of previously defined core PGC genes from d4/d6 PGCLC and E9.5 PGCs (Nakaki et al. 2013). To determine the cellular identity of the remaining clusters found within the EB, the expression of lineage specific markers identified in E8.25 mouse embryos (Ibarra-Soria et al. 2018) was used. To this end, all markers with a log2FoldChange >2.5 were considered. Each EB cluster was annotated as equivalent to the mouse embryonic tissue for which we observed the most significant enrichment in the expression of the corresponding marker genes."

Using the criteria mentioned above, the number of genes used as specific markers for each E8.25 mouse tissue was:

49 Amnion
73 Cardiac
75 CorePGC
65 EmbryonicBlood
74 Endothelial
69 ExtraEmbryonicEctoderm
140 ExtraEmbryonicEndoderm
25 ExtraEmbryonicMesoderm
19 ForeBrain
30 ForeGut
7 MidHindGut
10 NeuralCrest
10 NeuralTube
100 Notochord
12 Placodes

Furthermore, in the revised version we have included a new figure (Fig S1a) in which the average expression of the specific markers defining the main embryonic and extraembryonic tissues found within E8.25 mouse embryos (Ibarra-Soria *et al.* 2018) is shown for the cell clusters identified within d4 EBs. This new figure complements the previous one in which the expression of a few selected markers was shown for each of the main d4 EB cell clusters (new Fig S1b).

2. 'Therefore, the extinction of the naïve program seems to be necessary but not sufficient for the acquisition of germline competence, suggesting that differences, other than transcriptional, should exist between competent (EpiLC, E5.5 epiblast) and non-competent (EpiSC, >E6.5 epiblast) epiblast cells' – it is very difficult to ascertain the exact point the authors wish to make here. Of course, it is not sufficient simply to extinguish the naïve program to acquire germline competence – all other cell types (apart from the formative epiblast/EpiLC) have extinguished the naïve program and are not competent for germline induction. This statement and the paragraph in which it is embedded is very unclear, and seems to serve only to make the point that EpiSCs are less competent than EpiLC for PGC induction, which is widely appreciated.

We acknowledge that the previous statement about the competence of EpiSC and EpiLC for PGCLC induction was too complex. Therefore, it has been eliminated in the revised manuscript and substituted by the following sentences, which we believe give a more clear and concise message (pages 6-7): “Many PGCLC genes, especially those active in ESC, are lowly and similarly expressed in EpiLC and EpiSC (Fig. 1d,e, Supplemental Fig. 1d), yet only EpiLC display high germline competence.

Taking previous observations into account (Zylicz et al. 2015; Respuela et al. 2016; Tischler et al. 2019), we hypothesized that enhancers involved in the induction of PGCLC genes might display epigenetic differences between EpiLC and EpiSC that could explain their distinct germline competence.”.

3. When epigenomic datasets are compared between different studies are the cell lines cultured in similar conditions? (For instance, ChIPseq data obtained in serum grown ESCs is not relevant for 2i grown in ESCs, which from an epigenomic point of view are essentially a different cell type).

Yes, when comparing datasets from different studies, we always made sure that cell lines were cultured under similar conditions. In the particular case of ESC, all the presented data was generated in cells grown under 2i conditions.

4. Figure 3b – is 33% correct? It looks like there is 66% similarity depicted here?

The 33% is correct since what we are depicting is dissimilarity rather than similarity to quantify mCpG heterogeneity according to the original method described in (Hui et al., 2018). In Fig 3b (Fig 7c in the revised manuscript), when cell 1 and cell 2 are compared, there is one out of three CpG (i.e. 33%) for which the two cells show different methylation status. More details about how mCpG heterogeneity was quantified are provided in the Methods section (page 32): “CpG methylation heterogeneity was estimated with the PDclust package (Hui et al., 2018). The number of CpGs covered in each pair of cells resulted in approximately 150 CpGs for each pairwise comparison when PGCLC enhancers were considered. Then, the average of the absolute difference in the methylation values for all the CpGs covered for each pairwise comparison were computed as a dissimilarity matrix.”

5. Figure 5b – how many replicates were performed for the ChIP experiments? More generally it would be helpful if the number of replicates for all ChIP experiments was clearly shown.

The ChIP-seq experiments shown in Fig 5b (Fig 4f in the revised manuscript) were performed as single replicates, except for NANOG-HA in EpiSC, which were performed as biological duplicates. As the NANOG-HA binding signal within PGCLC enhancers was considerably lower in EpiSC compared to EpiLC, the ChIP-seq experiments in EpiSC were performed twice to ensure that the weak binding signal was not due to technical reasons.

In general, all the ChIP-seq experiments performed in our study, including the number of replicates, are provided in the “*Reporting-Summary*” file that *Nature Communications* asked us to fill in upon submission and that we believe is accessible to the reviewers. Nevertheless, this information has now been incorporated into Supplemental Data 6 and is also mentioned in the Methods section (page 29):

- *H3K4me1 ChIP-seq experiments in ESC (n = 2), EpiLC (n = 4) and EpiSC (n = 2) performed in R1 and E14Tg2a cell lines.*
- *H3K4me2 ChIP-seq experiments in ESC (n = 2), EpiLC (n = 3) and EpiSC (n = 3) performed in R1 and E14Tg2a cell lines.*
- *H3K4me3 ChIP-seq experiments in ESC (n = 2), EpiLC (n = 2) and EpiSC (n = 2) performed in R1 and E14Tg2a cell lines.*
- *H3K27ac ChIP-seq experiments in ESC (n = 2), EpiLC (n = 4) and EpiSC (n=2) performed in R1 and E14Tg2a cell lines.*
- *NANOG-HA ChIP-seq experiments in EpiSC were performed as two biological replicates in E14Tg2a.*
- *Additional ChIP-seq experiments were performed as single replicates (Supplementary Data 6).”*

6. Based on all the above issues, the title of the manuscript is clearly not appropriate.

The title of the revised manuscript has been modified: “*Enhancer-associated H3K4 methylation safeguards in vitro germline competence*”.

Reviewer #2 (Remarks to the Author):

The work by Bleckwehl et al. describes the role of H3K4 methylation in primordial germ cells (PGC) enhancers priming. First they perform single cell RNA-seq to characterize their differentiation system. Then, they accomplish a deep and comprehensive characterization of histone posttranscriptional modifications, DNA methylation, and chromatin accessibility of a set of 511 enhancer regions in the proximity of germline-activated genes. They find that H3K4me1 is partly retained at these enhancers in formative epiblast cells (EpiLC, which can give rise to PGC) in comparison to primed Epiblast cells (EpiSC, which cannot give rise to PGC). They propose that this difference is important to determine germline competence. After that, they demonstrate an important role of Mll3 and Mll4 in controlling the epigenetic state of these enhancers, PGC gene expression and specification. My main concern with this manuscript is the relatively small difference observed between the levels of H3K4me1 in EpiLC and EpiSC, especially in the E14Tg2a ESC line which was selected to perform most of the experiments. Although statistically significant when the whole population of 511 enhancers was compared, differences between EpiLC and EpiSC in figure 2d are really hard to see, especially in the enhancers of the lower half of the heatmap. Same comment can be argued about figure 2e (ATAC-seq), and S2e (H3K9me2). Several comments about this:

We would like to thank the reviewer for the overall positive impression about our work and for the insightful and constructive suggestions to improve it. Taking into consideration the suggestions from the three reviewers, we have now made extensive changes to the manuscript, including a new strategy to call PGCLC enhancers (see details in the following responses) and to identify those showing obvious differences in H3K4me1 levels between EpiLC and EpiSC.

All three reviewers have similar concerns about the mild differences in H3K4me1 between EpiLC and EpiSC. We think that part of the problem is that we previously referred to all PGCLC enhancers to describe the retention of H3K4me1 in EpiLC, while this was only obvious for a subset of enhancers, particularly those that are initially highly active in ESC (see new Fig 3c-d and Supplementary Fig 3b). In the revised text, we have clearly stated that the partial decommissioning of enhancers in EpiLC compared to EpiSC is observed for a subset of PGCLC enhancers. In addition, to more quantitatively defined the “partial retention” of H3K4me1 in EpiLC compared to EpiSC, we have now used three H3K4me1 ChIP-seq replicates in each cell type to classify PGCLC enhancers in two major groups (Fig 3c): (i) Group I: PGCLC enhancers showing a H3K4me1 EpiLC/EpiSC ratio higher than 1.2-fold in at least two of the three ChIP-seq replicates; (ii) Group II: all other PGCLC enhancers showing either similar or higher H3K4me1 signals in EpiSC compared to EpiLC. Using these criteria, 71% of the PGCLC enhancers were assigned to Group I and thus show higher H3K4me1 levels in EpiLC than in EpiSC (Fig 3d, Fig S3b-c). Using this new classification of PGCLC enhancers into Group I

and II, it becomes more obvious that those PGCLC enhancers showing higher levels of H3K4me1 in EpiLC than in EpiSC (i.e. Group I) are also more protected from DNA methylation and H3K9me3 (Fig 3e-f). Importantly, this seems to be also true when evaluating mCpG levels within PGCLC enhancers *in vivo* in the E5.5 and E6.5 epiblast (Fig 7b). Furthermore, the Group I enhancers also show higher H3K4me1 levels and lower CpG methylation in *Otx2*^{-/-} cells compared to their WT counterparts (Fig 4c-d), while such differences are not observed for Group II enhancers. Similarly, Group I enhancers seem to be particularly accessible and responsive to transcriptional activators (i.e. PRDM14 and NANOG) in EpiLC but not in EpiSC (Fig 4f-g).

On the other hand, chromatin accessibility, as measured by ATAC-seq, seems to be already very low in EpiLC and similar to the levels measured in EpiSC (Fig S3d). The major loss of ATAC-seq signals already in EpiLC argues against an active priming mechanism in which “pioneer” TF and MLL3/4 remain bound to the PGCLC enhancers but rather support more passive mechanisms in which H3K4me1 is progressively lost at PGCLC enhancers from ESC to EpiSC, but with slower dynamics in comparison to the silencing of PGCLC genes (see Discussion section, pages 16-17).

- The 511 enhancers were selected based on their proximity to PGC-regulated genes. Since not all close enhancers may control PGC dependent expression of the linked genes, an important number of the analyzed enhancers may not be related to PGC specification. In fact, when the authors perform 4C at the *Prdm14* gene, only the E3 enhancer seems to be a region that strongly contact the *Prdm14* promoter, but not E1 and E2.

The reviewer is right and linking enhancers and genes solely based on proximity can lead to frequent misassignments. Therefore, following the reviewer’s advice, we have now linked PGCLC enhancers to their putative target genes using publically available capture Hi-C data. More details about the new PGCLC enhancer calling strategy can be found in the following response.

- The authors should consider to use promoter-capture Hi-C data in order to better select the set of enhancers analyzed. Alternatively, they should use other bioinformatic tools, in addition to enhancer-gene proximity, to select the set of PGC enhancers. For example, the correlation between enhancers H3K27Ac signal and gene expression can be used in addition to proximity.

In the revised manuscript we have modified quite extensively our strategy to call PGCLC enhancers. Firstly, we have slightly changed the definition of the PGCLC genes by considering genes upregulated in d2+d4 PGCLC clusters with respect to

the remaining clusters in d2 and d4 EBs but without including in this analysis the d2 EpiLC (see Methods for more details). This has resulted in 389 PGCLC genes that are now presented in the revised Supplementary Data 2. Moreover, in response to some of the concerns from Reviewer#1, we have decided to call enhancers in PGCLC using H3K27ac data from d6-sorted PGCLC alone rather than by combining H3K27ac data from d2 and d6 PGCLC, since only d6-sorted PGCLC have an established *in vivo* counterpart (Hayashi et al., 2011). Lastly, following the reviewer's suggestions we have used Capture Hi-C data previously generated in 2i ESC by Atlasi et al., 2019 in order to call PGCLC enhancers based on their physical interaction with PGCLC genes. Our strategy to identify PGCLC enhancers is extensively described in the Methods section (page 30) and more briefly in the Results section (page 7):

"We first identified distal H3K27ac peaks in d6-sorted PGCLC using publically available data (Kurimoto et al. 2015). In agreement with our previous observations (Respuela et al. 2016), a large fraction of the d6 PGCLC H3K27ac peaks were initially active in ESC, lost H3K27ac in EpiLC and became progressively reactivated in d2 and d6 PGCLC (Supplementary Fig. 2a). Since most of the d6 PGCLC H3K27ac peaks were also active in ESC, we then used Capture Hi-C data generated in ESC (Atlasi et al. 2019) to systematically associate these distal peaks to their putative target genes. Finally, we defined PGCLC enhancers as those distal d6 PGCLC H3K27ac peaks that could be physically linked to our PGCLC gene set (Supplementary Data 2). This resulted in 415 PGCLC enhancers linked to 216 of the 389 PGCLC genes (Fig. 2a). Furthermore, to compare epigenetic changes between different enhancer groups, EpiLC and EpiSC enhancers were defined using similar criteria (Methods; Supplementary Data 3)."

- A general small decrease of H3K4me signal in EpiSC versus EpiLC is observed in all genome browser views shown (Figure 4a, 4d, S4a), both in the selected enhancer regions but also in other close peaks around. Have the authors analyzed whether reduced H3K4me1 signal in EpiSC occurs in domains? Is H3K4me1 signal generally decreased in EpiSC versus EpiLC? The authors should show control regions where H3K4me1 signal does not change and boxplots or density plots comparing all accessible enhancers H3K4me signal in the two stages. Have the authors verified whether levels of Mll3, Mll4 and Lsd1 proteins are similar in EpiSC and EpiLC.

As stated in a previous response, we have now classified PGCLC enhancers in two major groups (Fig 3c-d, Supplementary Fig. 3b): (i) Group I (71%): PGCLC enhancers showing a H3K4me1 EpiLC/EpiSC ratio higher than 1.2-fold in at least two of the three ChIP-seq replicates; (ii) Group II (29%): all other PGCLC enhancers showing either similar or higher H3K4me1 signals in EpiSC compared to EpiLC. Moreover, in order to compare epigenetic changes between different enhancer groups, in the revised manuscript we have also defined EpiLC (n=312) and EpiSC (n=223) enhancers using similar criteria to the ones used to call PGCLC

enhancers (see Methods; Supplementary Data 3). These EpiLC and EpiSC enhancers are now included as control regions in all the epigenomic quantifications shown throughout the manuscript (e.g. Fig 3a, Fig 4b, Fig S5d) to illustrate that the epigenetic changes observed for PGCLC enhancers do not simply reflect global differences between EpiLC and EpiSC. In addition, epigenomic profiles for representative PGCLC, EpiLC and EpiSC enhancers are also presented in Fig 5c and Fig S5g. Group II PGCLC enhancers (Fig 3d, Fig. S3b) and especially EpiSC enhancers (Fig 3a) show similar or higher H3K4me1 levels in EpiSC than in EpiLC, clearly illustrating that H3K4me1 signals are not generally decreased in EpiSC versus EpiLC.

On the other hand, analysis of public proteomic data (Yang et al., 2019) showed that the protein levels of LSD1/KDM1A, MLL3/KMT2C and MLL4/KMT2D were only slightly reduced for the histone methyltransferases and slightly increased for the histone demethylase upon EpiLC differentiation, respectively (see Reviewer Figure below). Such small changes in MLL3/4 levels are unlikely to have any profound effect on PGCLC differentiation, since, as shown in Fig. 6a, the ESC line with catalytic dead MLL4 but WT MLL3 (i.e. 4CT) showed normal PGCLC differentiation efficiency. Furthermore, the RNA-seq data we generated in d2 EpiLC and EpiSC did not show any significant differences for *Lsd1*, *Mll3* or *Mll4* between EpiLC and EpiSC.

- Authors state that “differences –EpiLC versus EpiSC – were not observed around the transcription start sites (TSS) of the PGCLC genes”. This is not what I see in Supplementary figure S2h for H3K4me1. I see differences comparable to those of PGCLC enhancers.

The reviewer is right and in the previous version of the manuscript we did not properly describe the results observed around the TSS of the PGCLC genes. This has now been modified in the revised manuscript (page 8): *“Moreover, when analyzing the transcription start sites (TSS) of the PGCLC genes we found that, although H3K4me1 was higher in EpiLC than in EpiSC, its overall levels were low compared to PGCLC enhancers (Fig. 3a). Similarly, constitutive heterochromatin*

marks (e.g. H3K9me3, mCpG) around TSS increased in EpiSC, but their levels were lower than within PGCLC enhancers. Other chromatin features typically found at promoter regions (e.g. H3K4me2/3, high chromatin accessibility) were similar around PGCLC TSS in EpiLC and EpiSC (Fig. 3a). Therefore, subsequent analyses were focused on PGCLC enhancers rather than promoters.”

- Authors should show levels of H3K4me1 by CHIP-PCR analysis of several enhancers in order to verify the CHIP-seq results.

We have performed CHIP-qPCR analyses for several Group I PGCLC enhancers in EpiLC and EpiSC (Supplementary Fig. 3c), which agree with our CHIP-seq results and further support that H3K4me1 levels within these enhancers are higher in EpiLC than in EpiSC.

Other concerns:

- In the Abstract the authors write: “we demonstrate that priming by H3K4me1/2 enables the robust activation of PGC enhancers”. The authors should be more cautious when they refer about the role of H3K4me1. Since PGC specification is not completely abolished and PGC gene expression is not strongly impaired in the dCD mutants they shouldn't say “enables”. In fact, PGC specification is possible in the absence of correct H3K4me1. Therefore, I find more appropriated the way that the authors use in the Discussion: “we propose that priming by H3K4me1/2 might facilitate, rather than being essential for, enhancer activation and the robust induction of developmental gene expression programs”. This facilitating but not essential role should be stated in the abstract.

We fully agree with the reviewer's suggestion and we have changed the abstract accordingly: “Our work suggests that, although H3K4me1 might not be essential for enhancer function, it can facilitate the (re)activation of enhancers and the establishment of gene expression programs during specific developmental transitions.”

- The model presented should express the results obtained in a more quantitative manner, other way the model may be misleading. For example, the model (Fig. 7f) shows H3K4me1/2 mark in Naïve pluripotency and wt formative pluripotency, but absence of mark in the primed pluripotency state. This is not what I see in Figure 2d. I see a drastic reduction between ESC and EpiLC and a small reduction, not an absence of the mark, between EpiLC and EpiSC. Same comment about H3K9me2/3. The model shows no H3K9me2/3 mark in the formative pluripotency state and presence of the mark in the primed pluripotency state, which it is not the experimental scenario. Authors should find a better way to show the experimental differences in their model.

We thank the reviewer for the suggestions. We have made changes to Fig 7f to summarize our results in a more quantitative and accurate manner.

Minor points.

- Page 9. Line 4 from the bottom. It should be “Supplementary Fig. 1g” instead of “Fig. 1g”.

This mistake has been corrected.

- Molecular data demonstrating the CRISPR-mediated enhancers deletion should be provided in supplementary figures.

The presence of the enhancer deletions in all the clonal ESC lines was confirmed by PCR genotyping followed by Sanger sequencing. Representative chromatograms showing the presence of the intended enhancer deletions are now shown in Supplementary Fig. 2c.

- Are the R1 WT (EpiLC and EpiSC) panels in Figures 6c and S2f identical?. If this is the case, this should be mentioned in the figure legend.

Yes, the same ChIP-seq data from R1 WT cells was used in the previous Fig 6c and S2f. In the revised manuscript, whenever the same ChIP-seq data is being shown in more than one figure this has been mentioned in the corresponding figure legends.

- Page 21. Line 3 from the bottom. It should be “Supplementary Fig. 6g” instead of “Supplementary Fig. 6f”.

This mistake has been corrected.

Reviewer #3 (Remarks to the Author):

The manuscript by Bleckwehl et al. shows that enhancer priming by H3K4me1/2 ensures gene expression involved in PGC differentiation. Using single cell analysis and in vitro differentiation system, the authors first confirmed homogenous downregulation of naïve pluripotent gene expression upon differentiation to the formative state. In contrast, the authors also found that enhancers controlling PGC specification had heterogenous DNA methylation, chromatin accessibility and H3K4me1. Among these heterogenous states, the authors demonstrated using MLL3/4 mutants that H3K4me1 plays a major role on priming enhancer for later gene expression in PGC specification. Furthermore, the authors confirmed the priming effect in Otx2-mutant; accelerated PGC specification in Otx2-mutant was attributable to elevated level of H3K4me1 in the PGC gene enhancers and was abrogated by impairment of MLL3/4 function. Following the previous report by the authors showing functional involvement of Foxd3 in exit of naïve pluripotent and germ cell specification, this study revealed molecular mechanisms of priming enhancers for PGC specification, which include sufficiently novel findings that have a significant impact on the research field. Specifically, it has been obscure how PGC competence is conferred to formative state pluripotent cells. This study provides a clear answer that H3K4me1(2) by MLL3/4 primes the enhancers of the PGC genes. The experiments are well designed, and the results largely support the author's conclusions. Although there are some comments below to be considered, this reviewer supports publication of this manuscript in Nature Communications.

We appreciate the reviewer's overall positive impression about our work. Taking into consideration the suggestions from the three reviewers, we have now made extensive changes to the manuscript, including a new strategy to call PGCLC enhancers and to identify those showing obvious epigenetic differences (especially in H3K4me1) between EpiLC and EpiSC (see responses below).

Specific comments

1) Differences in H3K4me1 and chromatin accessibility between EpiLC and EpiSC look subtle (Figure 2d and e). To help readers understand, it would be better to explain more intensively difference in repressive mark such as DNA methylation and H3K9 methylation. In this context, showing heat maps for H3K9me2/3 on the PGCLC enhancers would be informative.

All three reviewers have concerns about the subtle differences in H3K4me1 signals between EpiLC and EpiSC. We think that part of the problem is that we previously referred to all PGCLC enhancers to describe the retention of H3K4me1 in EpiLC, while this was only obvious for a subset of enhancers, particularly those that are initially highly active in ESC (see new Fig 3d and Supplementary Fig 3d). In the revised text, we have clearly stated that the partial decommissioning (rather than priming) of enhancers in EpiLC compared to EpiSC is observed for a subset of PGCLC enhancers. In addition, to more quantitatively defined the

“partial retention” of H3K4me1 in EpiLC compared to EpiSC, we have now used three H3K4me1 ChIP-seq replicates in each cell type to classify PGCLC enhancers in two major groups (Fig 3c): (i) Group I: PGCLC enhancers showing a H3K4me1 EpiLC/EpiSC ratio higher than 1.2-fold in at least two of the three ChIP-seq replicates; (ii) Group II: all other PGCLC enhancers showing either similar or higher H3K4me1 signals in EpiSC compared to EpiLC. Using these criteria, 71% of the PGCLC enhancers were assigned to Group I and, thus show higher H3K4me1 levels in EpiLC than in EpiSC (Fig 3d, Supplementary Fig 3b-c). Using this new classification of PGCLC enhancers into Group I and II, it becomes more obvious that those PGCLC enhancers showing higher levels of H3K4me1 in EpiLC than in EpiSC (i.e. Group I) are also more protected from DNA methylation and H3K9me3 (Fig 3e-f; H3K9me3 signals for PGCLC enhancers are now presented as a heat map in Fig 3e). Importantly, this seems to be also true when evaluating mCpG levels within PGCLC enhancers *in vivo* in the E5.5 and E6.5 epiblast (Fig 7b). Furthermore, the Group I enhancers also show higher H3K4me1 levels and lower CpG methylation in *Otx2*^{-/-} cells compared to their WT counterparts (Fig 4c-d), while such differences are not observed for Group II enhancers. Similarly, Group I enhancers seem to be particularly accessible and responsive to transcriptional activators (i.e. PRDM14 and NANOG) in EpiLC but not in EpiSC (Fig 4f-g).

On the other hand, chromatin accessibility, as measured by ATAC-seq, seems to be already very low in EpiLC and similar to the levels measured in EpiSC (Fig S3d). The major loss of ATAC-seq signals already in EpiLC argues against an active priming mechanism in which “pioneer” TF and MLL3/4 remain bound to the PGCLC enhancers but rather support more passive mechanisms in which H3K4me1 is progressively lost at PGCLC enhancers from ESC to EpiSC, but with slower dynamics in comparison to the silencing of PGCLC genes. Consequently, we have made changes in the manuscript text to more accurately describe our observations, which indicate that the partial decommissioning of PGCLC enhancers, including the persistence of H3K4me1, rather than a priming mechanism is important for *in vitro* germline competence. Some examples below:

Title: *Enhancer-associated H3K4 methylation safeguards in vitro germline competence*

Abstract: *„...In contrast, the decommissioning of enhancers associated with these germline genes is incomplete. Namely, a subset of these enhancers partly retain H3K4me1, accumulate less heterochromatic marks and remain accessible and responsive to transcriptional activators. Subsequently, as in vitro germline competence is lost, these enhancers get further decommissioned and lose their responsiveness to transcriptional activators...“*

Page 4, Introduction:

„Enhancer priming typically involves binding of pioneer TFs and pre-marking by H3K4me1 that can precede and facilitate subsequent enhancer activation (i.e. marking by H3K27ac, recruitment of RNA Pol II, production of eRNAs) (Lara-Astiaso et al. 2014; Lee et al. 2019; Wang et al. 2015; Lai et al. 2017). Interestingly, in differentiated macrophages, enhancers activated upon stimulation rapidly lose H3K27ac and TF binding, while retaining H3K4me1 for considerably longer. It was proposed that H3K4me1 persistence could facilitate a faster and stronger enhancer induction upon restimulation (Ostuni et al. 2013). It is currently unknown whether, during development, H3K4me1 persistence once enhancers become decommissioned can similarly facilitate their eventual re-activation(Calo and Wysocka 2013)“.

Page 16, Discussion:

“In the case of in vitro germline competence, here we report that a subset of PGCLC enhancers gets partly decommissioned in EpiLC and retains permissive chromatin features, including H3K4me1, already present in a preceding active state (i.e. in 2i ESC) (Fig 7g). This resembles the so-called latent enhancers previously described in differentiated macrophages, in which, following an initial round of activation and silencing, the persistence of H3K4me1 was proposed to facilitate subsequent enhancer induction upon restimulation (Ostuni et al. 2013). The mechanisms involved in the persistence of H3K4me1 and other permissive chromatin features are still unknown, although we can envision at least two non-mutually exclusive possibilities: (i) a passive mechanism whereby MLL3/4 binding to PGCLC enhancers is already lost in EpiLC, but H3K4me1 can still be transiently retained due to the slow dynamics of H3K4 demethylation(AlAbdi et al. 2020; Maltby et al. 2012); (ii) an active maintenance mechanism similar to the one reported for enhancer priming (Lee et al. 2019; Wang et al. 2015), whereby the binding of certain TFs might enable the persistent recruitment of MLL3/4 and the retention of H3K4me1 within PGCLC enhancers. Since PGCLC enhancers display low and similar ATAC-seq signals in EpiLC and EpiSC (Fig. 3), this would argue in favor of passive mechanisms rather than an active retention of TFs and co-activators (e.g. MLL3/4) in EpiLC.“

All together, the new analyses and data provided in the revised manuscript led us to suggest that the partial decommissioning of PGCLC enhancers, including the persistence of H3K4me1, can facilitate their future re-activation during PGCLC induction. We believe that this is conceptually different from previously proposed “priming” mechanisms, but could resemble what has been previously described for the so called „latent“ enhancers in differentiated macrophages, in which, following an initial round of activation and silencing, the persistence of H3K4me1 was proposed to facilitate subsequent enhancer induction upon restimulation (Ostuni et al. 2013).

2) The authors showed a lower mCpG level in PGCLC enhancers in *Esrrb* and *Lrrc31*. However, only two loci are not sufficient to conduct the conclusion. How about the enhancers of other genes? Also are these alterations of the mCpG level in *Esrrb* and *Lrrc31* statistically significant?

We assume that the referee refers to the ChIP-bisulfite sequencing experiments previously shown in Fig 3i (now in Fig 3g). However, we think that the negative correlation between H3K4me1 and mCpG levels within PGCLC enhancers is additionally supported by several data presented in the revised manuscript (new Fig. 3f):

- Genome-wide CpG methylation analysis in EpiLC and EpiSC shows that PGCLC enhancers globally show lower DNA methylation levels in EpiLC and EpiSC. This is particularly clear for Group I enhancers, which, as described above, display higher H3K4me1 signals in EpiLC than in EpiSC (new Fig 3d-e). The lower mCpG levels in EpiLC compared to EpiSC have been confirmed by locus-specific bisulfite sequencing of a couple of representative Group I PGCLC enhancers associated with *Esrrb* and *Prdm1*, respectively (new Fig S4d).

- CpG methylation and H3K4me1 levels are negatively correlated with each other across Group I PGCLC enhancers, particularly in EpiLC (new Fig 3f).

- The negative correlation between H3K4me1 and mCpG is further supported by analyses performed in *Otx2*^{-/-} cells, which are characterized by their extended germline competence. Global analyses show that *Otx2*^{-/-} EpiLC display higher H3K4me1 and lower mCpG within Group I PGCLC enhancers than WT EpiLC, while such differences are less obvious for Group II enhancers (new Fig 4b-d).

3) The deletion of the enhancers of *Lrrc31* and *Klf5* resulted in reduction of PGCLC differentiation. However, there is no report demonstrating that *Lrrc31* or *Klf5* is essential for PGC specification. The rationality of the functional outcome of the enhancer deletion should be shown.

Reviewer#1 has expressed some concerns regarding whether DPPA3-GFP signals alone are specific enough to identify PGCLC. After careful evaluation of previous reports in which either single or double reporter systems were used to quantify PGCLC (Ohinata et al. 2009; Hayashi et al., 2011), we share the concerns of Reviewer#1. Most importantly, the main goal of the enhancer deletion experiments was to validate our PGCLC enhancer calling strategy and evaluate whether those enhancers were important for the expression of their predicted target genes (new Fig 2 and Supplementary Fig 2) rather than for PGCLC induction. Therefore, taking all this into consideration, we have decided to eliminate the PGCLC quantifications based on the DPPA3-GFP reporter system from the revised manuscript.

4) FACS analyses of PGCLC induction are not convincing, especially in Figure S6d (therefore also Figure 6e) and the experiment using Dppa3-GFP. Refinement of the analysis is needed.

As mentioned in the previous response, the FACS analyses of the DPPA3-GFP reporter lines have been eliminated in the revised manuscript.

Regarding the FACS analyses originally shown in Fig S6d (now shown in Supplementary Fig. 6a in the revised version), we would like to mention that, in order to be consistent, all the PGCLC quantifications using surface markers presented throughout our manuscript have been performed using almost identical cut-offs. Moreover, the reproducibility of the results presented in Fig 6e (now Fig 6a in the revised manuscript) is supported by the fact that PGCLC quantifications were performed in at least four biological replicates for each of the investigated cell lines. On the other hand, we think that the reviewer's concern might be caused, at least partly, by the examples originally shown in Fig S6a, which were not the most representative of all the PGCLC quantifications performed in that set of experiments. Therefore, we are now showing new examples of the FACS analyses in Supplementary Fig. 6a. In addition, we have also added examples of the FACS analyses performed in WT and dCD cells after six days of PGCLC differentiation (new Supplementary Fig. 6b).

Furthermore, in the revised manuscript we have performed single-cell RNA-seq profiling of WT and dCD d4 EBs (new Fig 6c-e). In agreement with the results obtained using FACS (new Fig 6a), these new scRNA-seq analyses show that among the cell clusters identified in the d4 EBs, the one corresponding to PGCLC consist mostly of WT cells (new Fig. 6c,e). Interestingly, we also noticed that within the dCD EBs there were cells expressing major PGC markers (i.e. *Prdm1* or *Dppa3*) but not naïve pluripotency ones (i.e. *Klf4*) (new Fig. 6d), suggesting a cellular identity similar to PGCLC. The proportion of these *Prdm1orDppa3+/Klf4-* cells was similar among WT and dCD EBs (new Fig 6e). However, while in the WT EBs these cells were mostly found within the PGCLC subcluster, in the dCD EBs they were part of the subclusters with poorly defined identity (new Fig 6e), suggesting important transcriptional differences between WT and dCD *Prdm1+/Dppa3+/Klf4-* cells. Congruently, the expression of the PGCLC genes associated with PGCLC enhancers (e.g. *Tfap2c*, *Prdm14*, *Utf1*, *Esrrb*) was significantly reduced in dCD *Prdm1+/Dppa3+/Klf4-* cells in comparison to their WT counterparts, while the differences were less pronounced for the PGCLC genes without associated enhancers (new Fig. 6f-g, Supplemental Fig. 6f). Overall, these scRNA-seq analyses suggest that the induction of the PGCLC expression program, particularly of those genes linked to PGCLC enhancers, is compromised in dCD cells. Most importantly, together with our FACS analyses, these new results

further support the importance of H3K4me1 for *in vitro* germline competence and proper PGCLC induction.

5) Following sentences sounds awkward: Regardless, Klf5 and Lrrc31 might represent PGCLC regulators shared between naïve and PGCLC, similarly to other naïve/PGCLC TFs with this dual regulatory role (e.g. Prdm14 , Nanog). On the other hand, the Esrrb enhancer deletion moderately reduced the expression of Esrrb in ESC and did not significantly affect PGCLC differentiation (Fig. 4b,c). Nevertheless, the expression of Esrrb was severely diminished in d4 EB (Fig. 4c), demonstrating that this enhancer is needed for the proper induction of Esrrb in PGCLC. Can the authors revise? The authors should discuss these observations, based on the functionality of these genes in PGC specification.

These sentences were also unclear for other reviewers and we have made changes to the text accordingly. Moreover, as mentioned in a previous response, in the revised manuscript we have decided to focus on the effects that the enhancer deletions had on the expression of their predicted target genes rather than on PGCLC induction.

Page 7: “The deletion of the enhancers associated with Lrrc31/Lrrc34 and Klf5 significantly reduced the expression of the corresponding target genes in ESC and d4 EB (Supplementary Fig. 2d-e). The Esrrb enhancer deletion had a moderate effect in ESC and severely diminished Esrrb expression in d4 EB (Fig. 2c).”

6) In 4C-seq analysis, the authors only showed Prdm14 locus. Showing several examples other than Prdm14 would be more informative.

We thank the reviewer for this important comment. Since Reviewer#2 also expressed some concerns about the strength of the enhancer-promoter contacts observed within the *Prdm14* locus, we decided to more globally explore the chromatin interactions established by PGCLC genes using publically available Hi-C data previously generated in 2i ESC, EpiLC and EpiSC (Miura *et al.* 2019, DOI: 10.1038/s41588-019-0474-z; GEO: GSE113981). Briefly, we considered chromatin interactions previously identified in 2i ESC by *Atlasi et al.* 2019 using Capture Hi-C. Then, we used the Hi-C data from ESC, EpiLC and EpiSC to generate pile-up aggregate plots between either all the pairwise interactions reported by *Atlasi et al.* 2019 (n=135593) or the subset of those interactions in which one of the anchors was located within 3 Kb of the TSS of a PGCLC gene (n=2945). As it can be seen in the image below, chromatin interactions appear rather stable across pluripotent states when considering either all interactions or only the subset involving PGCLC genes.

The previous results suggest that the epigenetic differences observed for PGCLC enhancers between EpiLC and EpiSC do not seem to have a major impact on the 3D chromatin organization of the PGCLC gene loci. In contrast, in the revised manuscript we show that PGCLC enhancers displaying the most pronounced epigenetic differences between EpiLC and EpiSC (i.e. Group I enhancers) are highly accessible and responsive to transcriptional activators in EpiLC but not in EpiSC (new Fig 4f-g). Altogether, these results suggest that the partial decommissioning of PGCLC enhancers in EpiLC, including the persistence of H3K4me1, can facilitate their subsequent reactivation by germline TFs rather than the interaction with their target genes. Given these results and since Reviewer#1 suggested us to simplify and streamline the manuscript, in the revised version we have decided to exclude data related to 3D chromatin organization across *in vitro* pluripotent cell types (i.e. 4C-seq experiments within the *Prdm14* locus). However, if the reviewer considers that such 4C-seq experiments and/or the pile-up aggregate Hi-C plots shown above are worth reporting, we will be happy to include them in our manuscript.

REVIEWER COMMENTS

Reviewer #1 (Remarks to the Author):

I read with interest the rebuttal document and revised manuscript by Bleckwehl and colleagues. Given the extensive revisions I have reviewed the manuscript afresh, and have some remaining comments. Overall the manuscript is much improved, with a clearer narrative and message. What emerges is a large body of work that makes some interesting contributions to the field, as well as including a number of datasets that will be useful to others.

Major issues:

1. Figure 1b – PGCLC cluster. The appearance of this cluster is unusual. The majority of the d2 EB PGCLCs seem indistinguishable from other EB cells in the t-SNE plot. Whereas the d4 PGCLCs seem very similar to 2i ESCs. Are the authors confident that these d4 PGCLCs have not reverted to a naïve pluripotent state? Can the authors present alternative analysis that gives greater confidence that the cells selected in this plot are in fact PGCLCs that are clearly distinguishable from 2i ESC and d2 EB cells. Would unsupervised hierarchical clustering be appropriate for instance? Also do these PGCLC populations (particularly those clustering close to ESCs) display features that clearly distinguish them from ESCs – downregulation of *Klf4* and high levels of PGC genes (*Prdm1*, *Tfap2c*, *Dnd1*). On a related point, the 389 PGCLC genes identified by the authors appear to be more highly expressed in ESCs vs PGCLCs (Figure 1D). Are the authors able to identify a PGCLC signature that is actually higher in PGCLCs? Or a PGCLC gene set specific for PGCLCs – as they do for the three pluripotent states (what happens if they include PGCLCs in such analysis? i.e. alongside ESC, d2 EpiLC, EpiSC?). How would using such a specific gene set influence downstream analysis?

2. PGCLC enhancers. Related to the previous point, how many of the 415 PGCLC enhancers are also physically linked in ESCs? i.e. how specific is this physical linkage? Can truly PGCLC specific enhancers be identified, and if so how does this impact downstream analysis? More specifically, for the *Esrrb* enhancer deleted (Fig 2c) – is this physically linked in ESCs as well as in PGCLCs? If so, how does one interpret the data – is there a different ESC specific *Esrrb* enhancer? This relevant as the authors claim that this enhancer is particularly relevant in PGCLC induction – however, if it is similarly linked in ESCs, this rather suggests that it is the lack of other regulatory elements that is important.

The E3 *Pdm14* enhancer appears to have the smallest impact on gene expression, but the biggest impact on PGCLC induction. How should we interpret this? Finally I would alter this sentence: ‘the E1-E3 elements differentially contribute to *Prdm14* expression in either ESC (i.e. E2) or PGCLC (i.e. E1)’ to make clear the contribution is in early PGCLCs – as the authors point out the impact in later PGCLCs is fairly minimal.

3. Page 10: 'Otx2 -/- cells was correlated with the retention of H3K4me1, H3K4me2 and H3K27ac in PGCLC enhancers, particularly within those displaying partial decommissioning in WT EpiLC (i.e Group 1 enhancers)'. The language here is unhelpful, particularly 'partial decommissioning'. What defines Group1 enhancers is higher levels of H3K4me1 in EpiLC vs EpiSCs, so it would be much clearer to simply state this (i.e. 'those displaying higher levels of H3K4me1 in WT EpiLC').

I do not agree with the authors interpretation of Supplementary Figure 4c. In all genes examined H3K4me1 is higher or at an equivalent level in OTX2-/- ES cells compared with OTX2-/- EpiLC. For *Esrp1*, *Prdm1*, *Zfp42*, this is entirely consistent with the idea that higher levels of H3K4me1 are due to an increase in OTX2-/- ESCs and subsequent maintenance in EpiLC. This is the exact opposite of the authors description of the data, and impacts the conclusions drawn.

4. H3K4me1 is necessary for in vitro germline competence. The authors argue that the number of PGCLCs they can induce makes performing H3K27ac profiling in sorted PGCLCs unrealistic. As referenced by the authors, others have done this previously (Kurimoto et al. 2015). Although their calculations do suggest a very high numbers of PGCLC plates required, it does seem there is room for improvement in their PGCLC induction efficiency (which is low) and if this was combined with optimised low input profiling methods, then the experiment would be more feasible. This would undoubtedly improve the manuscript. However, given their later observations regarding dCD derived PGCLCs it may be that this whole section needs a rethink (and might render this experiment uninformative anyway).

The new scRNA-seq dataset used in place of performing CHIP here, does not really answer the same question and as such is introduced in a slightly unusual manner. However, it is certainly interesting. This suggests that very few, if any(?), PGCLCs are actually made from the mutant – less than suggested by the PGCLC FACS experiments. I am surprised the authors do not draw greater attention to this or offer further analysis. The description of *Prdm1orDppa3+/Klf4-* cells is confusing. In WT are these simply the same as PGCLCs or are there distinctive populations within the 'PGCLC cluster' – i.e. are these aberrant cells with less convincing naïve gene expression even in the WT setting? How many mutant PGCLCs are actually found in the PGCLC cluster and how does their transcriptome compare to WT. Are any mutant PGCLCs normal? If not, can further analysis of the difference give an indication as to what is going wrong? This is quite a striking worthy of further analysis. If no normal PGCLCs are made in the mutant, then this is an impressive phenotype. If there is an aberrant population that could be identified and sorted then it would be very interesting to perform H3K27ac CHIP/CUT&RUN in these cells – as this would really be the experiment that proves whether enhancer dysregulation leads to the phenotype (i.e. if H3K27ac is deposition is relatively normal despite the gene dysregulation, then this would argue strongly against this conclusion).

The conclusions in the Discussion might need to be revisited, in particular, 'Similarly, we found that the induction of PGCLC genes linked to enhancers was disrupted in dCD cells. Therefore, we propose that H3K4me1 might facilitate, rather than being essential for, enhancer (re)activation and the robust induction of developmental gene expression programs'. H3K4me1 may be doing substantially more than 'facilitating' if there really is such an impressive defect in PGCLC generation.

Minor points

1. Group 1 enhancers. I would define the groups in the text, as it will make the paper easier to follow.
2. First line, Page 6. Do the authors mean 'more highly expressed in ESC'? The data does seem to indicate this (see Major issue 1 above).
3. Especially regarding regulation of H3K9me3, Zyllicz et al. 2015 (PMID: 26551560), made an important contribution here (including in vivo data), and I think this study should be referenced/discussed more prominently (relevant to Pages 8/9 in particular).
4. Line numbers do help reviewers. Please include next time.

Reviewer #2 (Remarks to the Author):

The current version of the manuscript by Bleckwehl et al. has improved very much respect to the previous version I reviewed. In fact, most of my concerns have been addressed. However, I still have an important point that need to be improved. Statistical significance of differences has to be quantified all throughout the manuscript. The authors often use expression such as: "slightly lower H3K4me1 levels", "slightly higher H3K4me2", "reduced CpG methylation levels" without statistical test that support these statements. It is not sufficient to show the heatmaps. Different signal intensities at the selected enhancer sets should be quantified and statistical tests that support, or not, the existence of differences should be performed and provided at least in the following figures: 3d, 3e, 4c, 4d, 4g, 5b, 5d, 6i, 7a, 7b, and also in several similar supplementary figures.

Minor point:

- Supplementary figure 3d. It should be "E14Tg2a-H3K4me2" instead of "E14Tg2a-H34me2"

Reviewer #3 (Remarks to the Author):

In the revised manuscript, the authors made two main revisions in response to the comments of the reviewer. One is that PGC enhancers are classified into Group1 and 2 on the basis of enrichment of H3K4me1 in EpiLCs compared to EpiSCs. Under this condition, it becomes clear that Group1 enhancers are more protected from DNA methylation and H3K9me3. Although there is still a concern whether the 1.2-fold change in H3K4me1 enrichment is appropriate for the classification, this may be acceptable, because the protective effect is temporary during decommission of epigenetic marks in the PGC enhancers. The other is that the authors use FACS data with CD15 and CD61 antibodies instead of that using Dppa3-GFP. The results using CD15 and CD61 antibodies (Figure S6) are not as clear as previous reports (for example, Hayashi et al., 2011 Cell, Zhang et al. 2018 Nature). This may be due to difference in cells used and/or FACS apparatus. Nevertheless, the authors showed specific PGCLC differentiation in response to cytokines and more importantly demonstrated involvement of PGC enhancers in the expression of the predicted target genes. Other revision and responses made by author are overall acceptable.

REVIEWER COMMENTS

Reviewer #1 (Remarks to the Author):

I read with interest the rebuttal document and revised manuscript by Bleckwehl and colleagues. Given the extensive revisions I have reviewed the manuscript afresh, and have some remaining comments. Overall the manuscript is much improved, with a clearer narrative and message. What emerges is a large body of work that makes some interesting contributions to the field, as well as including a number of datasets that will be useful to others.

We would like to thank the reviewer for the constructive feedback and the overall positive impression about our work.

Major issues:

1. Figure 1b – PGCLC cluster. The appearance of this cluster is unusual. The majority of the d2 EB PGCLCs seem indistinguishable from other EB cells in the t-SNE plot. Whereas the d4 PGCLCs seem very similar to 2i ESCs. Are the authors confident that these d4 PGCLCs have not reverted to a naïve pluripotent state? Can the authors present alternative analysis that gives greater confidence that the cells selected in this plot are in fact PGCLCs that are clearly distinguishable from 2i ESC and d2 EB cells. Would unsupervised hierarchical clustering be appropriate for instance? Also do these PGCLC populations (particularly those clustering close to ESCs) display features that clearly distinguish them from ESCs – downregulation of *Klf4* and high levels of PGC genes (*Prdm1*, *Tfap2c*, *Dnd1*). On a related point, the 389 PGCLC genes identified by the authors appear to be more highly expressed in ESCs vs PGCLCs (Figure 1D). Are the authors able to identify a PGCLC signature that is actually higher in PGCLCs? Or a PGCLC gene set specific for PGCLCs – as they do for the three pluripotent states (what happens if they include PGCLCs in such analysis? i.e. alongside ESC, d2 EpiLC, EpiSC?). How would using such a specific gene set influence downstream analysis?

It was challenging to identify the PGCLC cluster from the complete data set containing all differentiation stages (ESC, EpiLC, EpiSC, d2 EB, d4 EB). Therefore, and as stated in the methods (page 25; line 791-803), we used unsupervised clustering (i.e. k-means clustering) of each stage separately (with k=3 for d2 EB and k=4 for d4 EB; Fig 1c) and identified d2 and d4 PGCLC clusters, which we then combined to annotate the PGCLC cluster. We are confident that the PGCLC cluster contains PGCLCs, as the cells of this cluster specifically express high levels of major PGCLC markers, including the ones mentioned by the reviewer (i.e. *Tfap2c*, *Prdm1*, *Dnd1*) (see Reviewer Fig 1; Fig 1e. Supplementary Fig. 1d).

Reviewer Fig 1.

Furthermore, the PGCLC cluster showed low levels of ESC markers (*Klf4*, *Tbx3*, *Zfp57*) (Fig. 1d, Supplementary Fig. 1d). This is also now stated in the text (page 6, lines 153-160). In addition, we have now performed a differential expression analysis between the cells of the PGCLC cluster and the 2i ESC (added to Supplementary Data 2), which shows that *Dppa3* and *Dnd1* are among the most highly up regulated genes (average log₂FC ~2 for both) in the PGCLC cluster, while *Klf4* and *Tfcp2l1* are among the most down regulated genes (average log₂FC -2,2 and -1,7, respectively).

On the other hand and as suggested by the reviewer, we have also performed a differential expression analysis of the PGCLC cluster vs. (EB / ESC / EpiLC and EpiSC, Supplementary Fig. 1e), which revealed a subset of 100 PGCLC specific genes, including *Dppa3*, *Dnd1*, *Prdm1* and *Tfap2c*. We think that this gene subset is too small to perform the downstream epigenomic comparisons of associated PGCLC enhancers. Nevertheless, as shown in Reviewer Fig 2, this subset of 100 PGCLC specific PGCLC genes were also strongly reduced in the *Dppa3+orPrdm1+andKlf4*- dCD cells of the d4 EB compared to their WT counterparts.

Overall, these new analyses show that the PGCLC cluster is formed by PGCLC that, despite their similarities with 2i ESC, are transcriptionally distinct.

Reviewer Fig.2

2. PGCLC enhancers. Related to the previous point, how many of the 415 PGCLC enhancers are also physically linked in ESCs? i.e. how specific is this physical linkage? Can truly PGCLC specific enhancers be identified, and if so how does this impact downstream analysis? More specifically, for the Esrrb enhancer deleted (Fig 2c) – is this physically linked in ESCs as well as in PGCLCs? If so, how does one interpret the data – is there a different ESC specific Esrrb enhancer? This relevant as the authors claim that this enhancer is particularly relevant in PGCLC induction – however, if it is similarly linked in ESCs, this rather suggests that it is the lack of other regulatory elements that is important.

The E3 Pdm14 enhancer appears to have the smallest impact on gene expression, but the biggest impact on PGCLC induction. How should we interpret this? Finally I would alter this sentence: ‘the E1-E3 elements differentially contribute to Prdm14 expression in either ESC (i.e. E2) or PGCLC (i.e. E1)’ to make clear the contribution is in early PGCLCs – as the authors point out the impact in later PGCLCs is fairly minimal.

All of the 415 enhancers are physically linked to at least one of the 389 PGCLC genes in ESC according to the capture Hi-C data from Atlasi et al. 2019, which, as stated in (page 7, line 193-198), was generated in ESC. Although the capture Hi-C data has high sequencing depth and resolution, we cannot rule out that some PGCLC enhancers might not be captured by using data generated in ESC. Moreover, this approach is also limited to enhancers with a physical interaction with their target genes, which might not be a universal property of all enhancers (PMID: 31494034). However, Hi-C methods in general and capture Hi-C in particular, typically requires even more cells than other genomic methods, thus currently precluding its application to PGCLC (PMID: 30010637). Nevertheless, using this capture Hi-C data, we were able to identify interactions between

enhancers and PGCLC genes that display low H3K27ac and gene expression levels in ESC, respectively (e.g. *Espr1*; Reviewer Fig 3), in agreement with the prevalence of pre-formed enhancer-gene contacts preceding gene activation (PMID: 25043061).

Reviewer Fig.3

Regarding the *Esrrb* locus, it is definitely possible that, as suggested by the reviewer, the enhancer landscape in ESC is more complex than in PGCLC (see updated Figure 2b). In agreement with this there are additional H3K27ac positive regions in ESC that are not present in PGCLC and that could represent ESC-specific enhancers, including a region immediately downstream of the deleted enhancer (see regions highlighted in blue in Fig 2b). Therefore, while in PGCLC the deleted enhancer seems to be the major *Esrrb* regulatory element, in ESC multiple enhancers might control *Esrrb* expression in a partially redundant manner (PMID: 29420474) and thus compensate for the absence of the deleted enhancer. This possibility is now briefly mentioned in the results section (page 7, line 209-213).

Regarding the *Prdm14* enhancers, as can be seen in Fig. 2e-f, the three different enhancers seem to have a rather similar impact on *Prdm14* expression (at day 4) and PGCLC induction. Nevertheless, the deletion of both E2 and E3 already affects

***Prdm14* expression in ESC, which could compromise the naïve pluripotent state by increasing the levels of DNMT3A and DNMT3B and of CpG methylation (Gretarsson and Hackett, 2020; Sim et al., 2017; Yamaji et al., 2013) and, thus, indirectly affect PGCLC induction. This possibility has been added to the main text, together with the correction suggested by the reviewer (page 8; line 224 and 227-231).**

3. Page 10: 'Otx2 ^{-/-} cells was correlated with the retention of H3K4me1, H3K4me2 and H3K27ac in PGCLC enhancers, particularly within those displaying partial decommissioning in WT EpiLC (i.e Group 1 enhancers)'. The language here is unhelpful, particularly 'partial decommissioning'. What defines Group1 enhancers is higher levels of H3K4me1 in EpiLC vs EpiSCs, so it would be much clearer to simply state this (i.e. 'those displaying higher levels of H3K4me1 in WT EpiLC').

I do not agree with the authors interpretation of Supplementary Figure 4c. In all genes examined H3K4me1 is higher or at an equivalent level in OTX2^{-/-} ES cells compared with OTX2^{-/-} EpiLC. For *Esrp1*, *Prdm1*, *Zfp42*, this is entirely consistent with the idea that higher levels of H3K4me1 are due to an increase in OTX2^{-/-} ESCs and subsequent maintenance in EpiLC. This is the exact opposite of the authors description of the data, and impacts the conclusions drawn.

It is true that in some cases (*Esrp1*, *Prdm1*, *Zfp42*) the H3K4me1 levels are higher in the *Otx2*^{-/-} ESC than in the WT ESC, although this is not the case for all enhancers. Therefore, we have changed the sentences according to the reviewer's suggestions, as we agree that this can improve the understanding of our results (page 10; line 296-302):

"ChIP-seq experiments revealed that the increased germline competence of Otx2^{-/-} cells was correlated with the retention of H3K4me1, H3K4me2 and H3K27ac in PGCLC enhancers, particularly within those displaying higher levels of H3K4me1 in WT EpiLC (i.e. Group 1 enhancers) (Fig. 4b-c, Supplementary Fig. 4b). Moreover, H3K4me1/2 levels were higher in Otx2^{-/-} ESC than in WT ESC in some but not all the investigated Group 1 enhancers (Supplementary Fig. 4c)."

4. H3K4me1 is necessary for in vitro germline competence. The authors argue that the number of PGCLCs they can induce makes performing H3K27ac profiling in sorted PGCLCs unrealistic. As referenced by the authors, others have done this previously (Kurimoto et al. 2015). Although their calculations do suggest a very high numbers of PGCLC plates required, it does seem there is room for improvement in their PGCLC induction efficiency (which is low) and if this was combined with optimised low input profiling methods, then the experiment would be more feasible. This would undoubtedly improve the manuscript. However, given their later observations regarding dCD derived PGCLCs it may be that this whole section needs a rethink (and might render this experiment uninformative anyway).

The new scRNA-seq dataset used in place of performing ChIP here, does not really answer the same question and as such is introduced in a slightly unusual manner. However, it is certainly interesting. This suggests that very few, if any(?), PGCLCs are actually made from the mutant – less than suggested by the PGCLC FACS experiments. I am surprised the authors do not draw greater attention to this or offer further analysis. The description of *Prdm1orDppa3+/Klf4-* cells is confusing. In WT are these simply the same as PGCLCs or are there distinctive populations within the ‘PGCLC cluster’ – i.e. are these aberrant cells with less convincing naïve gene expression even in the WT setting? How many mutant PGCLCs are actually found in the PGCLC cluster and how does their transcriptome compare to WT. Are any mutant PGCLCs normal? If not, can further analysis of the difference give an indication as to what is going wrong? This is quite a striking worthy of further analysis. If no normal PGCLCs are made in the mutant, then this is an impressive phenotype. If there is an aberrant population that could be identified and sorted then it would be very interesting to perform H3K27ac ChIP/CUT&RUN in these cells – as this would really be the experiment that proves whether enhancer dysregulation leads to the phenotype (i.e. if H3K27ac is deposition is relatively normal despite the gene dysregulation, then this would argue strongly against this conclusion). The conclusions in the Discussion might need to be revisited, in particular, ‘Similarly, we found that the induction of PGCLC genes linked to enhancers was disrupted in dCD cells. Therefore, we propose that H3K4me1 might facilitate, rather than being essential for, enhancer (re)activation and the robust induction of developmental gene expression programs’. H3K4me1 may be doing substantially more than ‘facilitating’ if there really is such an impressive defect in PGCLC generation.

Following the reviewer’s comments and suggestions, we have extended the analyses of the d4 EB single-cell RNA-seq data in WT and dCD and also have better described how the *Prdm1orDppa3+/Klf4-* were identified and analyzed (page 13-14, lines 395-435):

- **In order to more clearly show how many dCD and WT cells are found within the PGCLC cluster, we now provide a pie chart that shows that among all the cells found within this cluster, 95% are WT (n=162) and 5% are dCD (n=8) (Fig. 6d). Nevertheless, the expression of PGCLC genes in the 8 dCD cells was high and similar to the one of the 162 WT cells (Reviewer Fig. 4).**
- **To better describe how and why the *Dppa3+or Prdm1+andKlf4-* cells were investigated, we have now included a new Supplementary figure panel (Supplementary Fig. 6e) showing the expression of PGCLC genes in several of the subclusters found within d4 EBs. These analyses show that within some of the non-PGCLC clusters (e.g. “undefined”dCD2) there were cells displaying high expression of PGCLC genes and, thus, a cellular identity somehow similar to PGCLC. Based on these observations, we decided to further explore the PGCLC differentiation defects of the dCD cells using an alternative approach, whereby we identified all cells within the WT and dCD d4 EBs expressing of**

- major PGC markers (i.e. *Prdm1* or *Dppa3*) but not naïve pluripotency ones (i.e. *Klf4*). As shown in the new Figures 6e-f, although the overall abundance of these *Prdm1orDppa3+/Klf4-* cells was similar in WT and dCD EBs, their distribution among the different cell clusters was quite different. Namely, while the vast majority of WT *Prdm1orDppa3+/Klf4-* cells were found within the PGCLC cluster (91%), only a few dCD cells were part of the PGCLC cluster (5%) and many were found within the “undefined” clusters (61%) (Fig. 6e-f).
- Overall, these new analyses suggest that while there are few dCD cells that normally induce the PGCLC expression program (Supplementary Fig. 6f, Fig 6f), in the majority of dCD cells the induction of the PGCLC genes, particularly of those linked to PGCLC enhancers (Fig 6g), is reduced but not fully abrogated. Consequently, many dCD cells display a poorly defined identity (i.e. part of “undefined” clusters) in which the PGCLC expression program is only partially established. Therefore, we think that our previous statement regarding the “facilitating” role of H3K4me1 properly describes our results, since although the induction of PGCLC genes linked to enhancers is reduced in dCD cells, the magnitude of the observed gene expression changes is moderate (Fig. 6f-g). The term “facilitating” was recommended by another reviewer in light of the magnitude of the observed gene expression changes.
 - As stated above, the majority (61%) of the *Prdm1orDppa3+/Klf4-* dCD cells are found within the “undefined” clusters, which show poorly defined cellular identity (i.e. lack of differentially expressed specific markers, high transcriptional heterogeneity (Supplementary Fig. 6e). Therefore, we think it would be challenging to sort the dCD cells for a H3K27ac CUT&RUN approach as well as to properly interpret the resulting H3K27ac profiles. Moreover, based on our analyses and taking into consideration the comments made by the reviewer, we have decided to remove the d4 EB H3K27ac ChIP-seq data from the manuscript as we also found it uninformative at this point.

Reviewer Fig.4

Minor points

1. Group 1 enhancers. I would define the groups in the text, as it will make the paper easier to follow.

The Group I and Group II enhancers are now defined in the main text and not only in the Figure 3 legend (page 9, line 257-262).

2. First line, Page 6. Do the authors mean 'more highly expressed in ESC'? The data does seem to indicate this (see Major issue 1 above).

The suggested change has now been incorporated (page 6, line 151).

3. Especially regarding regulation of H3K9me3, Zyllicz et al. 2015 (PMID: 26551560), made an important contribution here (including in vivo data), and I think this study should be referenced/discussed more prominently (relevant to Pages 8/9 in particular).

We agree with the reviewer's suggestion and the indicated reference has now been included and briefly discussed in the main text (page 9, line 269-272).

4. Line numbers do help reviewers. Please include next time.

Line numbers have now been included in the main manuscript file.

Reviewer #2 (Remarks to the Author):

The current version of the manuscript by Bleckwehl et al. has improved very much respect to the previous version I reviewed. In fact, most of my concerns have been addressed. However, I still have an important point that need to be improved. Statistical significance of differences has to be quantified all throughout the manuscript. The authors often use expression such as: "slightly lower H3K4me1 levels", "slightly higher H3K4me2", "reduced CpG methylation levels" without statistical test that support these statements. It is not sufficient to show the heatmaps. Different signal intensities at the selected enhancer sets should be quantified and statistical tests that support, or not, the existence of differences should be performed and provided at least in the following figures: 3d, 3e, 4c, 4d, 4g, 5b, 5d, 6i, 7a, 7b, and also in several similar supplementary figures.

We would like to thank the reviewer for the constructive feedback and the overall positive impression about our work. Following the reviewer's suggestion, we have now added statistical tests results to all the indicated main figures as well as similar supplementary figures.

Minor point:

- Supplementary figure 3d. It should be “E14Tg2a-H3K4me2” instead of “E14Tg2a-H34me2”

This typo has now been corrected.

Reviewer #3 (Remarks to the Author):

In the revised manuscript, the authors made two main revisions in response to the comments of the reviewer. One is that PGC enhancers are classified into Group1 and 2 on the basis of enrichment of H3K4me1 in EpiLCs compared to EpiSCs. Under this condition, it becomes clear that Group1 enhancers are more protected from DNA methylation and H3K9me3. Although there is still a concern whether the 1.2-fold change in H3K4me1 enrichment is appropriate for the classification, this may be acceptable, because the protective effect is temporary during decommission of epigenetic marks in the PGC enhances. The other is that the authors use FACS data with CD15 and CD61 antibodies instead of that using Dppa3-GFP. The results using CD15 and CD61 antibodies (Figure S6) are not as clear as previous reports (for example, Hayashi et al., 2011 Cell, Zhang et al. 2018 Nature). This may be due to difference in cells used and/or FACS apparatus. Nevertheless, the authors showed specific PGCLC differentiation in response to cytokines and more importantly demonstrated involvement of PGC enhancers in the expression of the predicted target genes. Other revision and responses made by author are overall acceptable.

We would like to thank the reviewer for the constructive feedback and the overall positive assessment of our work.

REVIEWERS' COMMENTS

Reviewer #1 (Remarks to the Author):

The authors have dealt with my major concerns and the manuscript is now ready for publication.

There is one remaining overclaim:

Line 451: 'Altogether, our data shows that H3K4me1 is required for in vitro germline competence and proper PGCLC induction.'

The authors have not shown that H3K4me1 is required for either competence or induction - as some bona fide PGCLCs were derived from dCD cells. They have certainly shown strong evidence that this histone modification plays an important role.

Please correct this and any other similar overclaims that I might have overlooked.

Otherwise, I congratulate the authors on a job well done.

Reviewer #2 (Remarks to the Author):

All my concerns have been addressed. I have no additional criticisms about this manuscript.

REVIEWERS' COMMENTS

Reviewer #1 (Remarks to the Author):

The authors have dealt with my major concerns and the manuscript is now ready for publication.

There is one remaining overclaim:

Line 451: 'Altogether, our data shows that H3K4me1 is required for in vitro germline competence and proper PGCLC induction.'

The authors have not shown that H3K4me1 is required for either competence or induction - as some bona fide PGCLCs were derived from dCD cells. They have certainly shown strong evidence that this histone modification plays an important role.

Please correct this and any other similar overclaims that I might have overlooked.

Otherwise, I congratulate the authors on a job well done.

We would like to sincerely thank the reviewer for all the insightful comments and suggestions, which have truly helped us to improve our manuscript and avoid making unnecessary and unjustified overclaims.

We have made changes to the indicated sentence, which now reads as follows:

"Altogether, our data strongly suggest that H3K4me1 is important for in vitro germline competence and proper PGCLC induction."

Reviewer #2 (Remarks to the Author):

All my concerns have been addressed. I have no additional criticisms about this manuscript.

We thank the reviewer for the positive assessment of our work.